# SUBGRADIENT DESCENT LEARNS ORTHOGONAL DICTIONARIES

**Yu Bai, Qijia Jiang & Ju Sun**
Stanford University
{yub,qjiang2,sunju}@stanford.edu

## ABSTRACT

This paper concerns dictionary learning, i.e., sparse coding, a fundamental representation learning problem. We show that a subgradient descent algorithm, with random initialization, can recover orthogonal dictionaries on a natural nonsmooth, nonconvex $\ell_1$ minimization formulation of the problem, under mild statistical assumption on the data. This is in contrast to previous provable methods that require either expensive computation or delicate initialization schemes. Our analysis develops several tools for characterizing landscapes of nonsmooth functions, which might be of independent interest for provable training of deep networks with nonsmooth activations (e.g., ReLU), among other applications. Preliminary synthetic and real experiments corroborate our analysis and show that our algorithm works well empirically in recovering orthogonal dictionaries.

## 1 INTRODUCTION

Dictionary learning (DL), i.e. , sparse coding, concerns the problem of learning compact representations, i.e., given data $Y$, one tries to find a representation basis $A$ and coefficients $X$, so that $Y \approx AX$ where $X$ is most sparse. DL has numerous applications especially in image processing and computer vision (Mairal et al., 2014). When posed in analytical form, DL seeks a transformation $Q$ such that $QY$ is sparse; in this sense DL can be considered as an (extremely!) primitive "deep" network (Ravishankar & Bresler, 2013).

Many heuristic algorithms have been proposed to solve DL since the seminal work of Olshausen & Field (1996), most of them surprisingly effective in practice (Mairal et al., 2014; Sun et al., 2015). However, understandings on when and how DL is solvable have only recently started to emerge. Under appropriate generating models on $A$ and $X$, Spielman et al. (2012) showed that complete (i.e., square, invertible) $A$ can be recovered from $Y$, provided that $X$ is ultra-sparse. Subsequent works (Agarwal et al., 2017; Arora et al., 2014; 2015; Chatterji & Bartlett, 2017; Awasthi & Vijayaraghavan, 2018) provided similar guarantees for overcomplete (i.e. fat) $A$, again in the ultra-sparse regime. The latter methods are invariably based on nonconvex optimization with model-dependent initialization, rendering their practicality on real data questionable.

The ensuing developments have focused on breaking the sparsity barrier and addressing the practicality issue. Convex relaxations based on the sum-of-squares (SOS) SDP hierarchy can recover overcomplete $A$ when $X$ has linear sparsity (Barak et al., 2015; Ma et al., 2016; Schramm & Steurer, 2017), while incurring expensive computation (solving large-scale SDP's or large-scale tensor decomposition). By contrast, Sun et al. (2015) showed that complete $A$ can be recovered in the linear sparsity regime by solving a certain nonconvex problem with arbitrary initialization. However, the second-order optimization method proposed there is still expensive. This problem is partially addressed by (Gilboa et al., 2018) which proved that the first-order gradient descent with random initialization enjoys a similar performance guarantee.

A standing barrier toward practicality is dealing with nonsmooth functions. To promote sparsity in the coefficients, the $\ell_1$ norm is the function of choice in practical DL, as is common in modern signal processing and machine learning (Candès, 2014): despite its nonsmoothness, this choice often admits highly scalable numerical methods, such as proximal gradient method and alternating direction

---

The reader is welcome to refer to our arXiv version for future updates.

method (Mairal et al., 2014). The analyses in Sun et al. (2015); Gilboa et al. (2018), however, focused on characterizing the algorithm-independent function landscape of a certain nonconvex formulation of DL, which takes a smooth surrogate to $\ell_1$ to get around the nonsmoothness. The tactic smoothing there introduced substantial analysis difficulty, and broke the practical advantage of computing with the simple $\ell_1$ function.

In this paper, we show that working directly with a natural $\ell_1$ norm formulation results in neat analysis and a practical algorithm. We focus on the problem of learning orthogonal dictionaries: given data $\{\boldsymbol{y}_i\}_{i\in[m]}$ generated as $\boldsymbol{y}_i = \boldsymbol{A}\boldsymbol{x}_i$, where $\boldsymbol{A} \in \mathbb{R}^{n\times n}$ is a fixed unknown orthogonal matrix and each $\boldsymbol{x}_i \in \mathbb{R}^n$ is an iid Bernoulli-Gaussian random vector with parameter $\theta \in (0,1)$, recover $\boldsymbol{A}$. This statistical model is the same as in previous works (Spielman et al., 2012; Sun et al., 2015).

Write $\boldsymbol{Y} \doteq [\boldsymbol{y}_1,\ldots,\boldsymbol{y}_m]$ and similarly $\boldsymbol{X} \doteq [\boldsymbol{x}_1,\ldots,\boldsymbol{x}_m]$. We propose to recover $\boldsymbol{A}$ by solving the following nonconvex (due to the constraint), nonsmooth (due to the objective) optimization problem:

$$\text{minimize}_{\boldsymbol{q}\in\mathbb{R}^n} \ f(\boldsymbol{q}) \doteq \frac{1}{m}\left\|\boldsymbol{q}^\top\boldsymbol{Y}\right\|_1 = \frac{1}{m}\sum_{i=1}^{m}|\boldsymbol{q}^\top\boldsymbol{y}_i| \quad \text{subject to } \|\boldsymbol{q}\|_2 = 1. \qquad (1.1)$$

Based on the statistical model, $\boldsymbol{q}^\top\boldsymbol{Y} = \boldsymbol{q}^\top\boldsymbol{A}\boldsymbol{X}$ has the highest sparsity when $\boldsymbol{q}$ is a column of $\boldsymbol{A}$ (up to sign) so that $\boldsymbol{q}^\top\boldsymbol{A}$ is 1-sparse. Spielman et al. (2012) formalized this intuition and optimized the same objective as Eq. (1.1) with a $\|\boldsymbol{q}\|_\infty = 1$ constraint, which only works when $\theta \sim O(1/\sqrt{n})$. Sun et al. (2015) worked with the sphere constraint but replaced the $\ell_1$ objective with a smooth surrogate, introducing substantial analytical and computational deficiencies as alluded above.

In constrast, we show that with sufficiently many samples, the optimization landscape of formulation (1.1) is benign with high probability (over the randomness of $\boldsymbol{X}$), and a simple Riemannian subgradient descent algorithm can provably recover $\boldsymbol{A}$ in polynomial time.

**Theorem 1.1** (Main result, informal version of Theorem 3.1). *Assume $\theta \in [1/n, 1/2]$. For $m \geq \Omega(\theta^{-2}n^4\log^4 n)$, the following holds with high probability: there exists a $\mathrm{poly}(m, \epsilon^{-1})$-time algorithm, which runs Riemannian subgradient descent on formulation (1.1) from at most $O(n\log n)$ independent, uniformly random initial points, and outputs a set of vectors $\{\widehat{\boldsymbol{a}}_1,\ldots,\widehat{\boldsymbol{a}}_n\}$ such that up to permutation and sign change, $\|\widehat{\boldsymbol{a}}_i - \boldsymbol{a}_i\|_2 \leq \epsilon$ for all $i \in [n]$.*

In words, our algorithm works also in the linear sparsity regime, the same as established in Sun et al. (2015); Gilboa et al. (2018), at a lower sample complexity $O(n^4)$ in contrast to the existing $O(n^{5.5})$ in Sun et al. (2015). [1] As for the landscape, we show that (Theorems 3.4 and 3.6) each of the desired solutions $\{\pm\boldsymbol{a}_i\}_{i\in[n]}$ is a local minimizer of formulation (1.1) with a sufficiently large basin of attraction so that a random initialization will land into one of the basins with at least constant probability. To obtain the result, we integrate and develop elements from nonsmooth analysis (on Riemannian manifolds), set-valued analysis, and random set theory, which might be valuable to studying other nonconvex, nonsmooth optimization problems.

## 1.1 RELATED WORK

**Dictionary learning**  Besides the many results sampled above, we highlight similarities of our result to Gilboa et al. (2018). Both propose first-order optimization methods with random initialization, and several quantities we work with in the proofs are the same. A defining difference is we work with the nonsmooth $\ell_1$ objective directly, while Gilboa et al. (2018) built on the smoothed objective from Sun et al. (2015). We put considerable emphasis on practicality: the subgradient of the nonsmooth objective is considerably cheaper to evaluate than that of the smooth objective in Sun et al. (2015), and in the algorithm we use Euclidean projection rather than exponential mapping to remain feasible—again, the former is much lighter for computation.

**General nonsmooth analysis**  While nonsmooth analytic tools such as subdifferential for convex functions are now well received in machine learning and relevant communities, that for general functions are much less so. The Clarke subdifferential and relevant calculus developed for the family of locally Lipschitz functions seem to be particularly relevant, and cover several families of functions of interest, such as convex functions, differentiable functions, and many forms of composition

---

[1]The sample complexity in Gilboa et al. (2018) is not explicitly stated.

(Clarke, 1990; Aubin, 1998; Bagirov et al., 2014). Remarkably, majority of the tools and results can be generalized to locally Lipschitz functions on Riemannnian manifolds (Ledyaev & Zhu, 2007; Hosseini & Pouryayevali, 2011). Our formulation (1.1) is exactly optimization of a locally Lipschitz function (as it is convex) on a Riemannian manifold (the sphere). For simplicity, we try to avoid the full manifold language, nonetheless.

**Nonsmooth optimization on Riemannian manifolds or with constraints**  Equally remarkable is many of the smooth optimization techniques and convergence results can be naturally adapted to optimization of locally Lipschitz functions on Riemannian manifolds (Grohs & Hosseini, 2015; Hosseini, 2015; Hosseini & Uschmajew, 2017; Grohs & Hosseini, 2016). New optimization methods such as gradient sampling and variants have been invented to solve general nonsmooth problems (Burke et al., 2005; 2018; Bagirov et al., 2014; Curtis & Que, 2015; Curtis et al., 2017). Almost all available convergence results pertain to only global convergence, which is too weak for our purpose. Our specific convergence analysis gives us a local convergence result (Theorem 3.8).

**Nonsmooth landscape characterization**  Nonsmoothness is not a big optimization barrier if the problem is convex; here we review some recent work on analyzing nonconvex nonsmooth problems. Loh & Wainwright (2015) study the regularized empirical risk minimization problem with nonsmooth regularizers and show results of the type "all stationary points are within statistical error of ground truth" under certain restricted strong convexity of the smooth risk. Duchi & Ruan (2017); Davis et al. (2017) study the phase retrieval problem with $\ell_1$ loss, characterizing its nonconvex nonsmooth landscape and providing efficient algorithms.

There is a recent surge of work on analyzing one-hidden-layer ReLU networks, which are nonconvex and nonsmooth. Algorithm-independent characterizations of the landscape are mostly local and require strong initialization procedures (Zhong et al., 2017), whereas stronger global results can be established via designing new loss functions (Ge et al., 2017), relating to PDEs (Mei et al., 2018), or problem-dependent analysis of the SGD (Li & Yuan, 2017; Li & Liang, 2018). Our result provides an algorithm-independent chacaterization of the landscape of non-smooth dictionary learning, and is "almost global" in the sense that the initialization condition is satisifed by random initialization with high probability.

**Other nonsmooth problems in application**  Prevalence of nonsmooth problems in optimal control and economics is evident from all monographs on nonsmooth analysis (Clarke, 1990; Aubin, 1998; Bagirov et al., 2014). In modern machine learning and data analysis, nonsmooth functions are often taken to encode structural information (e.g., sparsity, low-rankness, quantization), or whenever robust estimation is desired. In deep learning, the optimization problem is nonsmooth when nonsmooth activations are in use, e.g., the popular ReLU. The technical ideas around nonsmooth analysis, set-valued analysis, and random set theory that we gather and develop here are particularly relevant to these applications.

## 2   PRELIMINARIES

**Problem setup**  Given an unknown orthogonal dictionary $\boldsymbol{A} = [\boldsymbol{a}_1, \ldots, \boldsymbol{a}_n] \in \mathbb{R}^{n \times n}$, we wish to recover $\boldsymbol{A}$ through $m$ observations of the form

$$\boldsymbol{y}_i = \boldsymbol{A}\boldsymbol{x}_i, \tag{2.1}$$

or $\boldsymbol{Y} = \boldsymbol{A}\boldsymbol{X}$ in matrix form, where $\boldsymbol{X} = [\boldsymbol{x}_1, \ldots, \boldsymbol{x}_m]$ and $\boldsymbol{Y} = [\boldsymbol{y}_1, \ldots, \boldsymbol{y}_m]$.

The coefficient vectors $\boldsymbol{x}_i$ are sampled from the Bernoulli-Gaussian distribution with parameter $\theta \in (0, 1)$, denoted as $\mathrm{BG}(\theta)$: each entry $x_{ij}$ is independently drawn from a standard Gaussian with probability $\theta$ and zero otherwise. The Bernoulli-Gaussian is a good prototype distribution for sparse vectors, as $\boldsymbol{x}_i$ will be on average $\theta$-sparse. For any $\boldsymbol{z} \sim_{iid} \mathrm{Ber}(\theta)$, we let $\Omega$ denote the set of non-zero indices, which is a random set itself.

We assume that $n \geq 3$ and $\theta \in [1/n, 1/2]$. In particular, $\theta \geq 1/n$ is to require that each $\boldsymbol{x}_i$ has at least one non-zero entry on average.

**First-order geometry**  We will focus on the first-order geometry of the non-smooth objective Eq. (1.1): $f(\boldsymbol{q}) = \frac{1}{m} \sum_{i=1}^{m} |\boldsymbol{q}^\top \boldsymbol{y}_i|$. In the whole Euclidean space $\mathbb{R}^n$, $f$ is convex with

sub-differential set

$$\partial f(\boldsymbol{q}) = \frac{1}{m} \sum_{i=1}^{m} \text{sign}(\boldsymbol{q}^\top \boldsymbol{y}_i) \boldsymbol{y}_i, \tag{2.2}$$

where $\text{sign}(\cdot)$ is the set-valued sign function (i.e. $\text{sign}(0) = [-1, 1]$). As we minimize $f$ subject to the constraint $\|\boldsymbol{q}\|_2 = 1$, our problem is no longer convex. The Riemannian sub-differential of $f$ on $\mathbb{S}^{n-1}$ is defined as (Hosseini & Uschmajew, 2017):

$$\partial_R f(\boldsymbol{q}) \doteq (\boldsymbol{I} - \boldsymbol{q}\boldsymbol{q}^\top)\partial f(\boldsymbol{q}). \tag{2.3}$$

A point $\boldsymbol{q}$ is stationary for problem Eq. (1.1) if $\boldsymbol{0} \in \partial_R f(\boldsymbol{q})$. We will not distinguish between local maxima and saddle points—we call a stationary point $\boldsymbol{q}$ a saddle point if there is a descent direction (i.e. direction along which the function is locally maximized at $\boldsymbol{q}$).

**Set-valued analysis** As the subdifferential is a set-valued mapping, analyzing it requires some set-valued analysis, which we briefly present here. The addition of two sets is defined as the Minkowski summation: $X + Y = \{x + y : x \in X, y \in Y\}$. The expectation of random sets is a straightforward extension of the Minkowski sum allowing any measurable "selection" procedure; for the concrete definition see (Molchanov, 2013). The Hausdorff distance between two sets is defined as

$$\text{d}_\text{H}(X_1, X_2) \doteq \sup \left\{ \sup_{\boldsymbol{x}_1 \in X_1} \text{d}(\boldsymbol{x}_1, X_2), \sup_{\boldsymbol{x}_2 \in X_2} \text{d}(\boldsymbol{x}_2, X_1) \right\}. \tag{2.4}$$

Basic properties about the Hausdorff distance are provided in Appendix A.1.

**Notations** Bold small letters (e.g., $\boldsymbol{x}$) are vectors and bold capitals are matrices (e.g., $\boldsymbol{X}$). The dotted equality $\doteq$ is for definition. For any positive integer $k$, $[k] \doteq \{1, \dots, k\}$. By default, $\|\cdot\|$ is the $\ell_2$ norm if applied to a vector, and the operator norm if applied to a matrix. $C$ and $c$ or any indexed versions are reserved for universal constants that may change from place to place.

## 3 MAIN RESULT

We now state our main result, the recovery guarantee for learning orthogonal dictionary by solving formulation (1.1).

**Theorem 3.1** (Recovering orthogonal dictionary via subgradient descent). *Suppose we observe*

$$m \geq C n^4 \theta^{-2} \log^4 n \tag{3.1}$$

*samples in the dictionary learning problem and we desire an accuracy $\epsilon \in (0, 1)$ for recovering the dictionary. With probability at least $1 - \exp\left(-cm\theta^3 n^{-3} \log^{-3} m\right) - \exp\left(-c'R/n\right)$, an algorithm which runs Riemannian subgradient descent $R = C'n \log n$ times with independent random initializations on $\mathbb{S}^{n-1}$ outputs a set of vectors $\{\hat{\boldsymbol{a}}_1, \dots, \hat{\boldsymbol{a}}_n\}$ such that up to permutation and sign change, $\|\hat{\boldsymbol{a}}_i - \boldsymbol{a}_i\|_2 \leq \epsilon$ for all $i \in [n]$. The total number of subgradient descent iterations is bounded by*

$$C'' R \theta^{-16/3} \epsilon^{-8/3} n^4 \log^{8/3} n. \tag{3.2}$$

*Here $C, C', C'', c, c' > 0$ are universal constants.*

At a high level, the proof of Theorem 3.1 consists of the following steps, which we elaborate throughout the rest of this section.

1. Partition the sphere into $2n$ symmetric "good sets" and show certain directional gradient is strong on population objective $\mathbb{E}[f]$ inside the good sets (Section 3.1).

2. Show that the same geometric properties carry over to the empirical objective $f$ with high probability. This involves proving the uniform convergence of the subdifferential set $\partial f$ to $\mathbb{E}[\partial f]$ (Section 3.2).

3. Under the benign geometry, establish the convergence of Riemannian subgradient descent to one of $\{\pm \boldsymbol{a}_i : i \in [n]\}$ when initialized in the corresponding "good set" (Section 3.3).

4. Calling the randomly initialized optimization procedure $O(n \log n)$ times will recover all of $\{\boldsymbol{a}_1, \dots, \boldsymbol{a}_n\}$ with high probability, by a coupon collector's argument (Section 3.4).

**Scaling and rotating to identity** Throughout the rest of this paper, we are going to assume WLOG that the dictionary is the identity matrix, i.e. $\boldsymbol{A} = \boldsymbol{I}_n$, so that $\boldsymbol{Y} = \boldsymbol{X}$, $f(\boldsymbol{q}) = \left\| \boldsymbol{q}^\top \boldsymbol{X} \right\|_1$, and the goal is to find the standard basis vectors $\{\pm \boldsymbol{e}_1, \ldots, \pm \boldsymbol{e}_n\}$. The case of a general orthogonal $\boldsymbol{A}$ can be reduced to this special case via rotating by $\boldsymbol{A}^\top$: $\boldsymbol{q}^\top \boldsymbol{Y} = \boldsymbol{q}^\top \boldsymbol{A} \boldsymbol{X} = (\boldsymbol{q}')^\top \boldsymbol{X}$ where $\boldsymbol{q}' = \boldsymbol{A}^\top \boldsymbol{q}$ and applying the result on $\boldsymbol{q}'$. We also scale the objective by $\sqrt{\pi/2}$ for convenience of later analysis.

## 3.1 PROPERTIES OF THE POPULATION OBJECTIVE

We begin by characterizing the geometry of the expected objective $\mathbb{E}[f]$. Recall that we have rotated $A$ to be identity, so that we have

$$f(\boldsymbol{q}) = \sqrt{\frac{\pi}{2}} \cdot \frac{1}{m} \left\| \boldsymbol{q}^\top \boldsymbol{X} \right\|_1 = \sqrt{\frac{\pi}{2}} \cdot \frac{1}{m} \sum_{i=1}^m \left| \boldsymbol{q}^\top \boldsymbol{x}_i \right|, \ \ \partial f(\boldsymbol{q}) = \sqrt{\frac{\pi}{2}} \cdot \frac{1}{m} \sum_{i=1}^m \operatorname{sign}\left( \boldsymbol{q}^\top \boldsymbol{x}_i \right) \boldsymbol{x}_i. \ \ (3.3)$$

**Minimizers and saddles of the population objective**  We begin by computing the function value and subdifferential set of the population objective and giving a complete characterization of its stationary points, i.e. local minimizers and saddles.

**Proposition 3.2** (Population objective value and gradient). *We have*

$$\mathbb{E}[f](\boldsymbol{q}) = \sqrt{\frac{\pi}{2}} \cdot \mathbb{E}\left[ \left| \boldsymbol{q}^\top \boldsymbol{x} \right| \right] = \mathbb{E}_\Omega \| \boldsymbol{q}_\Omega \| \tag{3.4}$$

$$\partial \mathbb{E}[f](\boldsymbol{q}) = \mathbb{E}[\partial f](\boldsymbol{q}) = \sqrt{\frac{\pi}{2}} \cdot \mathbb{E}\left[ \operatorname{sign}\left( \boldsymbol{q}^\top \boldsymbol{x} \right) \boldsymbol{x} \right] = \mathbb{E}_\Omega \begin{cases} \boldsymbol{q}_\Omega / \| \boldsymbol{q}_\Omega \|, & \boldsymbol{q}_\Omega \neq 0, \\ \{\boldsymbol{v}_\Omega : \| \boldsymbol{v}_\Omega \| \leq 1\}, & \boldsymbol{q}_\Omega = 0. \end{cases} \tag{3.5}$$

**Proposition 3.3** (Stationary points). *The stationary points of $\mathbb{E}[f]$ on the sphere are*

$$\mathcal{S} = \left\{ \frac{1}{\sqrt{k}} \boldsymbol{q} : \boldsymbol{q} \in \{-1, 0, 1\}^n, \|\boldsymbol{q}\|_0 = k, \ k \in [n] \right\}. \tag{3.6}$$

*The case $k = 1$ corresponds to the $2n$ global minimizers $\boldsymbol{q} = \pm \boldsymbol{e}_i$, and all other values of $k$ correspond to saddle points.*

A consequence of Proposition 3.3 is that the population objective has no "spurious local minima": each stationary point is either a global minimizer or a saddle point, though the problem itself is non-convex due to the constraint.

**Identifying $2n$ "good" subsets** We now define $2n$ subsets on the sphere, each containing one of the global minimizers $\{\pm \boldsymbol{e}_i\}$ and possessing benign geometry for both the population and empirical objective, following (Gilboa et al., 2018). For any $\zeta \in [0, \infty)$ and $i \in [n]$ define

$$\mathcal{S}_\zeta^{(i+)} \doteq \left\{ \boldsymbol{q} : q_i > 0, \frac{q_i^2}{\| \boldsymbol{q}_{-i} \|_\infty^2} \geq 1 + \zeta \right\}, \ \ \mathcal{S}_\zeta^{(i-)} \doteq \left\{ \boldsymbol{q} : q_i < 0, \frac{q_i^2}{\| \boldsymbol{q}_{-i} \|_\infty^2} \geq 1 + \zeta \right\}. \ \ (3.7)$$

For points in $\mathcal{S}_\zeta^{(i+)} \cup \mathcal{S}_\zeta^{(i-)}$, the $i$-th index is larger than all other indices (in absolute value) by a multiplicative factor of $\zeta$. In particular, for any point in these subsets, the largest index is unique, so by Proposition 3.3 all population saddle points are excluded from these $2n$ subsets.

Intuitively, this partition can serve as a "tiebreaker": points in $\mathcal{S}_{\zeta_0}^{(i+)}$ is closer to $\boldsymbol{e}_i$ than all the other $2n - 1$ signed basis vectors. Therefore, we hope that optimization algorithms initialized in this region could favor $\boldsymbol{e}_i$ over the other standard basis vectors, which we are going to show is indeed the case. For simplicity, we are going to state our geometry results in $\mathcal{S}_\zeta^{(n+)}$; by symmetry the results will automatically carry over to all the other $2n - 1$ subsets.

**Theorem 3.4** (Lower bound on directional subgradients). *Fix any $\zeta_0 \in (0, 1)$. We have*

*(a) For all $\boldsymbol{q} \in \mathcal{S}_{\zeta_0}^{(n+)}$ and all indices $j \neq n$ such that $q_j \neq 0$,*

$$\inf \left\langle \mathbb{E}[\partial_R f](\boldsymbol{q}), \frac{1}{q_j} \boldsymbol{e}_j - \frac{1}{q_n} \boldsymbol{e}_n \right\rangle \geq \frac{1}{2n} \theta (1 - \theta) \frac{\zeta_0}{1 + \zeta_0}. \tag{3.8}$$

*(b) For all $q \in \mathcal{S}_{\zeta_0}^{(n+)}$, we have that*

$$\inf \langle \mathbb{E} \left[ \partial_R f \right] (q), q_n q - e_n \rangle \geq \frac{1}{8} \theta (1 - \theta) \zeta_0 n^{-3/2} \| q_{-n} \|. \tag{3.9}$$

These lower bounds verify our intuition: points inside $\mathcal{S}_{\zeta_0}^{(n+)}$ have subgradients pointing towards $e_n$, both in a coordinate-wise sense and a combined sense: the direction $e_n - q_n q$ is exactly the tangent direction of the sphere at $q$ that points towards $e_n$.

## 3.2 BENIGN GEOMETRY OF THE EMPIRICAL OBJECTIVE

We now show that the benign geometry in Theorem 3.4 is carried onto the empirical objective $f$ given sufficiently many samples, using a concentration argument. The key result behind is the concentration of the empirical subdifferential set to the population subdifferential, where concentration is measured in the Hausdorff distance between sets.

**Proposition 3.5** (Uniform convergence of subdifferential). *For any $t \in (0, 1]$, when*

$$m \geq C t^{-2} n \log^2(n/t), \tag{3.10}$$

*with probability at least $1 - \exp \left( -cm\theta t^2 / \log m \right)$, we have*

$$\mathrm{d}_{\mathrm{H}} \left( \partial f (q), \mathbb{E} \left[ \partial f \right] (q) \right) \leq t \quad \text{for all } q \in \mathbb{S}^{n-1}. \tag{3.11}$$

*Here $C, c \geq 0$ are universal constants.*

The concentration result guarantees that the sub-differential set is close to its expectation given sufficiently many samples with high probability. Choosing an appropriate concentration level $t$, the lower bounds on the directional subgradients carry over to the empirical objective $f$, which we state in the following theorem.

**Theorem 3.6** (Directional subgradient lower bound, empirical objective). *There exist universal constants $C, c \geq 0$ so that the following holds: for all $\zeta_0 \in (0, 1)$, when $m \geq Cn^4 \theta^{-2} \zeta_0^{-2} \log^2 (n/\zeta_0)$, with probability at least $1 - \exp \left( -cm\theta^3 \zeta_0^2 n^{-3} \log^{-1} m \right)$, the following properties hold simultaneously for all the $2n$ subsets $\left\{ \mathcal{S}_{\zeta_0}^{(i+)}, \mathcal{S}_{\zeta_0}^{(i-)} : i \in [n] \right\}$: (stated only for $\mathcal{S}_{\zeta_0}^{(n+)}$)*

*(a) For all $q \in \mathcal{S}_{\zeta_0}^{(n+)}$ and all $j \in [n]$ with $q_j \neq 0$ and $q_n^2 / q_j^2 \leq 3$,*

$$\inf \left\langle \partial_R f (q), \frac{1}{q_j} e_j - \frac{1}{q_n} e_n \right\rangle \geq \frac{1}{4n} \theta (1 - \theta) \frac{\zeta_0}{1 + \zeta_0}. \tag{3.12}$$

*(b) For all $q \in \mathcal{S}_{\zeta_0}^{(n+)}$,*

$$\inf \langle \partial_R f (q), q_n q - e_n \rangle \geq \frac{\sqrt{2}}{16} \theta (1 - \theta) n^{-\frac{3}{2}} \zeta_0 \| q_{-n} \| \geq \frac{1}{16} \theta (1 - \theta) n^{-\frac{3}{2}} \zeta_0 \| q - e_n \|. \tag{3.13}$$

The consequence of Theorem 3.6 is two-fold. First, it guarantees that the only possible stationary point of $f$ in $\mathcal{S}_{\zeta_0}^{(n+)}$ is $e_n$: for every other point $q \neq e_n$, property (b) guarantees that $0 \notin \partial_R f(q)$, therefore $q$ is non-stationary. Second, the directional subgradient lower bounds allow us to establish convergence of the Riemannian subgradient descent algorithm, in a way similar to showing convergence of unconstrained gradient descent on star strongly convex functions.

We now present an upper bound on the norm of the subdifferential sets, which is needed for the convergence analysis.

**Proposition 3.7.** *There exist universal constants $C, c \geq 0$ such that*

$$\sup \| \partial f (q) \| \leq 2 \quad \forall q \in \mathbb{S}^{n-1} \tag{3.14}$$

*with probability at least $1 - \exp \left( -cm\theta \log^{-1} m \right)$, provided that $m \geq Cn \log n$. This particularly implies that*

$$\sup \| \partial_R f (q) \| \leq 2 \quad \forall q \in \mathbb{S}^{n-1}. \tag{3.15}$$

### 3.3 FINDING ONE BASIS VIA RIEMANNIAN SUBGRADIENT DESCENT

The benign geometry of the empirical objective allows a simple Riemannian subgradient descent algorithm to find one basis vector a time. The Riemannian subgradient descent algorithm with initialization $\boldsymbol{q}^{(0)}$ and step size $\left\{\eta^{(k)}\right\}_{k \geq 0}$ is as follows. For an arbitrary $\boldsymbol{v} \in \partial_R f\left(\boldsymbol{q}^{(k)}\right)$,

$$\boldsymbol{q}^{(k+1)} = \frac{\boldsymbol{q}^{(k)} - \eta^{(k)}\boldsymbol{v}}{\left\|\boldsymbol{q}^{(k)} - \eta^{(k)}\boldsymbol{v}\right\|}, \quad \text{for } k = 0, 1, 2, \dots . \tag{3.16}$$

Each iteration moves in an arbitrary Riemannian subgradient direction followed by a projection back onto the sphere. We show that the algorithm is guaranteed to find one basis as long as the initialization is in the "right" region. To give a concrete result, we set $\zeta_0 = 1/(5\log n)$.[2]

**Theorem 3.8** (One run of subgradient descent recovers one basis). *Let $m \geq C\theta^{-2}n^4\log^4 n$ and $\epsilon \in (0, 2\theta/25]$. With probability at least $1 - \exp\left(-cm\theta^3 n^{-3}\log^{-3} m\right)$ the following happens. If the initialization $\boldsymbol{q}^{(0)} \in \mathcal{S}_{1/(5\log n)}^{(n+)}$, and we run the projected Riemannian subgradient descent with step size $\eta^{(k)} = k^{-\alpha}/(100\sqrt{n})$ with $\alpha \in (0, 1/2)$, and keep track of the best function value so far until after iterate $K$ is performed, producing $\boldsymbol{q}^{\text{best}}$. Then, $\boldsymbol{q}^{\text{best}}$ obeys*

$$f\left(\boldsymbol{q}^{\text{best}}\right) - f\left(\boldsymbol{e}_n\right) \leq \epsilon, \quad \text{and} \quad \left\|\boldsymbol{q}^{\text{best}} - \boldsymbol{e}_n\right\| \leq \frac{16}{\theta(1-\theta)}\epsilon, \tag{3.17}$$

*provided that*

$$K \geq \max \left\{ \left(\frac{32000 n^{5/2}\log n\,(1-\alpha)}{\theta(1-\theta)\,\epsilon}\right)^{1/(1-\alpha)}, \left(\frac{64\frac{1-\alpha}{1-2\alpha}n^{3/2}\log n}{5\theta(1-\theta)\,\epsilon}\right)^{1/\alpha} \right\}. \tag{3.18}$$

*In particular, choosing $\alpha = 3/8 < 1/2$, it suffices to let*

$$K \geq K_{3/8} \doteq C'\theta^{-8/3}\epsilon^{-8/3}n^4\log^{8/3} n. \tag{3.19}$$

*Here $C, C', c \geq 0$ are universal constants.*

The above optimization result (Theorem 3.8) shows that Riemannian subgradient descent is able to find the basis vector $\boldsymbol{e}_n$ when initialized in the associated region $\mathcal{S}_{1/(5\log n)}^{(n+)}$. We now show that a simple uniformly random initialization on the sphere is guaranteed to be in one of these $2n$ regions with at least probability $1/2$.

**Lemma 3.9** (Random initialization falls in "good set"). *Let $\boldsymbol{q}^{(0)} \sim \text{Uniform}(\mathbb{S}^{n-1})$, then with probability at least $1/2$, $\boldsymbol{q}^{(0)}$ belongs to one of the $2n$ sets $\left\{\mathcal{S}_{1/(5\log n)}^{(i+)}, \mathcal{S}_{1/(5\log n)}^{(i-)} : i \in [n]\right\}$.*

### 3.4 RECOVERING ALL BASES FROM MULTIPLE RUNS

As long as the initialization belongs to $\mathcal{S}_{1/(5\log n)}^{(i+)}$ or $\mathcal{S}_{1/(5\log n)}^{(i-)}$, our finding-one-basis result in Theorem 3.8 guarantees that Riemannian subgradient descent will converge to $\boldsymbol{e}_i$ or $-\boldsymbol{e}_i$ respectively. Therefore if we run the algorithm with independent, uniformly random initializations on the sphere multiple times, by a coupon collector's argument, we will recover all the basis vectors. This is formalized in the following theorem.

**Theorem 3.10** (Recovering the identity dictionary from multiple random initializations). *Let $m \geq Cn^4\theta^{-2}\log^4 n$ and $\epsilon \in (0, 1)$, with probability at least $1 - \exp\left(-cm\theta^3 n^{-3}\log^{-3} m\right)$ the following happens. Suppose we run the Riemannian subgradient descent algorithm independently for $R$ times, each with a uniformly random initialization on $\mathbb{S}^{n-1}$, and choose the step size as $\eta^{(k)} = k^{-3/8}/(100\sqrt{n})$. Then, provided that $R \geq C'n\log n$, all standard basis vectors will be recovered up to $\epsilon$ accuracy with probability at least $1 - \exp\left(-cR/n\right)$ in $C'R\theta^{-16/3}\epsilon^{-8/3}n^4\log^{8/3} n$ iterations. Here $C, C', c \geq 0$ are universal constants.*

When the dictionary $\boldsymbol{A}$ is not the identity matrix, we can apply the rotation argument sketched in the beginning of this section to get the same result, which leads to our main result in Theorem 3.1.

---

[2] It is possible to set $\zeta_0$ to other values, inducing different combinations of the final sample complexity, iteration complexity, and repetition complexity in Theorem 3.10.

## 4 PROOF HIGHLIGHTS

A key technical challenge is establishing the uniform convergence of subdifferential sets in Proposition 3.5, which we now elaborate. Recall that the population and empirical subdifferentials are

$$\partial f(\boldsymbol{q}) = \sqrt{\frac{\pi}{2}} \cdot \frac{1}{m} \sum_{i=1}^{m} \operatorname{sign}(\boldsymbol{q}^\top \boldsymbol{x}_i) \boldsymbol{x}_i, \quad \mathbb{E}[\partial f](\boldsymbol{q}) = \sqrt{\frac{\pi}{2}} \cdot \mathbb{E}_{\boldsymbol{x} \sim \mathrm{BG}(\theta)}\left[\operatorname{sign}(\boldsymbol{q}^\top \boldsymbol{x}) \boldsymbol{x}\right], \quad (4.1)$$

and we wish to show that the difference between $\partial f(\boldsymbol{q})$ and $\mathbb{E}[\partial f](\boldsymbol{q})$ is small uniformly over $\boldsymbol{q} \in \mathcal{Q} = \mathbb{S}^{n-1}$. Two challenges stand out in showing such a uniform convergence:

1. The subdifferential is set-valued and random, and it is unclear a-priori how one could formulate and analyze the concentration of random sets.

2. The usual covering argument won't work here, as the Lipschitz gradient property does not hold: $\partial f(\boldsymbol{q})$ and $\mathbb{E}[\partial f](\boldsymbol{q})$ are not Lipschitz in $\boldsymbol{q}$. Therefore, no matter how fine we cover the sphere in Euclidean distance, points not in this covering can have radically different subdifferential sets.

### 4.1 CONCENTRATION OF RANDOM SETS

We state and analyze concentration of random sets in the Hausdorff distance (defined in Section 2). We now illustrate how the Hausdorff distance is the "right" distance to consider for concentration of subdifferentials—the reason is that the Hausdorff distance is closely related to the *support function* of sets, which for any set $S \in \mathbb{R}^n$ is defined as

$$h_S(\boldsymbol{u}) \doteq \sup_{\boldsymbol{x} \in S} \langle \boldsymbol{x}, \boldsymbol{u} \rangle. \quad (4.2)$$

For convex compact sets, the sup difference between their support functions is exactly the Hausdorff distance.

**Lemma 4.1** (Section 1.3.2, Molchanov (2013)). *For convex compact sets $X, Y \subset \mathbb{R}^n$, we have*

$$\mathrm{d}_\mathrm{H}(X, Y) = \sup_{\boldsymbol{u} \in \mathbb{S}^{n-1}} |h_X(\boldsymbol{u}) - h_Y(\boldsymbol{u})|. \quad (4.3)$$

Lemma 4.1 is convenient for us in the following sense. Suppose we wish to upper bound the difference of $\partial f(\boldsymbol{q})$ and $\mathbb{E}[\partial f](\boldsymbol{q})$ along some direction $\boldsymbol{u} \in \mathbb{S}^{n-1}$ (as we need in proving the key empirical geometry result Theorem 3.6). As both subdifferential sets are convex and compact, by Lemma 4.1 we immediately have

$$\left| \inf_{\boldsymbol{g} \in \partial f(\boldsymbol{q})} \langle \boldsymbol{g}, \boldsymbol{u} \rangle - \inf_{\boldsymbol{g} \in \mathbb{E}[\partial f](\boldsymbol{q})} \langle \boldsymbol{g}, \boldsymbol{u} \rangle \right| = \left| -h_{\partial f(\boldsymbol{q})}(-\boldsymbol{u}) + h_{\mathbb{E}[\partial f](\boldsymbol{q})}(-\boldsymbol{u}) \right| \le \mathrm{d}_\mathrm{H}(\partial f(\boldsymbol{q}), \mathbb{E}[\partial f](\boldsymbol{q})).$$
$$(4.4)$$

Therefore, as long as we are able to bound the Hausdorff distance, all directional differences between the subdifferentials are simultaneously bounded, which is exactly what we want to show to carry the benign geometry from the population to the empirical objective.

### 4.2 COVERING IN THE $\mathrm{d}_\mathbb{E}$ METRIC

We argue that the absence of gradient Lipschitzness is because the Euclidean distance is not the "right" metric in this problem. Think of the toy example $f(x) = |x|$, whose subdifferential set $\partial f(x) = \operatorname{sign}(x)$ is not Lipschitz across $x = 0$. However, once we partition $\mathbb{R}$ into $\mathbb{R}_{>0}$, $\mathbb{R}_{<0}$ and $\{0\}$ (i.e. according to the sign pattern), the subdifferential set is Lipschitz on each subset.

The situation with the dictionary learning objective is quie similar: we resolve the gradient non-Lipschitzness by proposing a stronger metric $\mathrm{d}_\mathbb{E}$ on the sphere which is sign-pattern aware and averages all "subset angles" between two points. Formally, we define $\mathrm{d}_\mathbb{E}$ as

$$\mathrm{d}_\mathbb{E}(\boldsymbol{p}, \boldsymbol{q}) \doteq \mathbb{P}_{\boldsymbol{x} \sim \mathrm{BG}(\theta)}\left[\operatorname{sign}(\boldsymbol{p}^\top \boldsymbol{x}) \ne \operatorname{sign}(\boldsymbol{q}^\top \boldsymbol{x})\right] = \frac{1}{\pi} \mathbb{E}_\Omega \angle(\boldsymbol{p}_\Omega, \boldsymbol{q}_\Omega), \quad (4.5)$$

(the second equality shown in Lemma C.1.) Our plan is to perform the covering argument in $\mathrm{d}_{\mathbb{E}}$, which requires showing gradient Lipschitzness in $\mathrm{d}_{\mathbb{E}}$ and bounding the covering number.

**Lipschitzness of** $\partial f$ **and** $\mathbb{E}\left[\partial f\right]$ **in** $\mathrm{d}_{\mathbb{E}}$ For the population subdifferential $\mathbb{E}\left[\partial f\right]$, note that $\mathbb{E}\left[\partial f\right](\boldsymbol{q}) = \mathbb{E}_{\boldsymbol{x}\sim\mathrm{BG}(\theta)}[\mathrm{sign}(\boldsymbol{q}^\top\boldsymbol{x})\boldsymbol{x}]$ (modulo rescaling). Therefore, to bound $\mathrm{d}_{\mathrm{H}}(\mathbb{E}\left[\partial f\right](\boldsymbol{p}), \mathbb{E}\left[\partial f\right](\boldsymbol{q}))$ by Lemma 4.1, we have the bound for all $\boldsymbol{u}\in\mathbb{S}^{n-1}$

$$\left|h_{\mathbb{E}[\partial f](\boldsymbol{p})}(\boldsymbol{u}) - h_{\mathbb{E}[\partial f](\boldsymbol{q})}(\boldsymbol{u})\right| = \mathbb{E}\left[\sup\left|\mathrm{sign}(\boldsymbol{p}^\top\boldsymbol{x}) - \mathrm{sign}(\boldsymbol{q}^\top\boldsymbol{x})\right|\cdot\left|\boldsymbol{x}^\top\boldsymbol{u}\right|\right] \quad (4.6)$$

$$\leq 2\mathbb{E}\left[\mathbb{1}\left\{\mathrm{sign}(\boldsymbol{p}^\top\boldsymbol{x})\neq\mathrm{sign}(\boldsymbol{q}^\top\boldsymbol{x})\right\}\left|\boldsymbol{x}^\top\boldsymbol{u}\right|\right]. \quad (4.7)$$

As long as $\mathrm{d}_{\mathbb{E}}(\boldsymbol{p},\boldsymbol{q})\leq\epsilon$, the indicator is non-zero with probability at most $\epsilon$, and thus the above expectation should also be small – we bound it by $O(\epsilon\sqrt{\log(1/\epsilon)})$ in Lemma F.5.

To show the same for the empirical subdifferential $\partial f$, one only needs to bound the observed proportion of sign differences for all $\boldsymbol{p},\boldsymbol{q}$ such that $\mathrm{d}_{\mathbb{E}}(\boldsymbol{p},\boldsymbol{q})\leq\epsilon$, which by a VC dimension argument is uniformly bounded by $2\epsilon$ with high probability (Lemma C.5).

**Bounding the covering number in** $\mathrm{d}_{\mathbb{E}}$ Our first step is to reduce $\mathrm{d}_{\mathbb{E}}$ to the **maximum length-2 angle** (the $\mathrm{d}_2$ metric) over any consistent support pattern. This is achieved through the following *vector angle inequality* (Lemma C.2): for any $\boldsymbol{p},\boldsymbol{q}\in\mathbb{R}^d$ ($d\geq 3$), we have

$$\angle(\boldsymbol{p},\boldsymbol{q}) \leq \sum_{\Omega\subset[d],|\Omega|=2}\angle(\boldsymbol{p}_\Omega,\boldsymbol{q}_\Omega) \quad\text{provided } \angle(\boldsymbol{p},\boldsymbol{q})\leq\pi/2. \quad (4.8)$$

Therefore, as long as $\mathrm{sign}(\boldsymbol{p}) = \mathrm{sign}(\boldsymbol{q})$ (coordinate-wise) and $\max_{|\Omega|=2}\angle(\boldsymbol{p}_\Omega,\boldsymbol{q}_\Omega)\leq\epsilon/n^2$, we would have for all $|\Omega|\geq 3$ that

$$\angle(\boldsymbol{p}_\Omega,\boldsymbol{q}_\Omega)\leq\pi/2 \quad\text{and}\quad \angle(\boldsymbol{p}_\Omega,\boldsymbol{q}_\Omega)\leq\sum_{\Omega'\subset\Omega,|\Omega'|=2}\angle(\boldsymbol{p}_{\Omega'},\boldsymbol{q}_{\Omega'})\leq\binom{|\Omega|}{2}\cdot\frac{\epsilon}{n^2}\leq\epsilon. \quad (4.9)$$

By Eq. (4.5), the above implies that $\mathrm{d}_{\mathbb{E}}(\boldsymbol{p},\boldsymbol{q})\leq\epsilon/\pi$, the desired result. Hence the task reduces to constructing an $\eta = \epsilon/n^2$ covering in $\mathrm{d}_2$ over any consistent sign pattern.

Our second step is a **tight bound on this covering number**: the $\eta$-covering number in $\mathrm{d}_2$ is bounded by $\exp(Cn\log(n/\eta))$ (Lemma C.3). For bounding this, a first thought would be to take the covering in all size-2 angles (there are $\binom{n}{2}$ of them) and take the common refinement of all their partitions, which gives covering number $(C/\eta)^{O(n^2)} = \exp(Cn^2\log(1/\eta))$. We improve upon this strategy by *sorting* the coordinates in $\boldsymbol{p}$ and restricting attentions in the consecutive size-2 angles after the sorting (there are $n-1$ of them). We show that a proper covering in these consecutive size-2 angles by $\eta/n$ will yield a covering for all size-2 angles by $\eta$. The corresponding covering number in this case is thus $(Cn/\eta)^{O(n)} = \exp(Cn\log(n/\eta))$, which modulo the $\log n$ factor is the tightest we can get.

## 5 Experiments

**Setup** We set the true dictionary $\boldsymbol{A}$ to be the identity and random orthogonal matrices, respectively. For each choice, we sweep the combinations of $(m,n)$ with $n\in\{30,50,70,100\}$ and $m = 10n^{\{0.5,1,1.5,2,2.5\}}$, and fix the sparsity level at $\theta = 0.1, 0.3, 0.5$, respectively. For each $(m,n)$ pair, we generate 10 problem instances, corresponding to re-sampling the coefficient matrix $\boldsymbol{X}$ for 10 times. Note that our theoretical guarantee applies for $m = \widetilde{\Omega}(n^4)$, and the sample complexity we experiment with here is lower than what our theory requires. To recover the dictionary, we run the Riemannian subgradient descent algorithm Eq. (3.16) with decaying step size $\eta^{(k)} = 1/\sqrt{k}$, corresponding to the boundary case $\alpha = 1/2$ in Theorem 3.8 with a much better base size.

**Metric** As Theorem 3.1 guarantees recovering the entire dictionary with $R\geq Cn\log n$ independent runs, we perform $R = \mathrm{round}\,(5n\log n)$ runs on each instance. For each run, a true dictionary element $\boldsymbol{a}_i$ is considered to be found if $\|\boldsymbol{a}_i - \boldsymbol{q}_{\mathrm{best}}\|\leq 10^{-3}$. For each instance, we regard it a successful recovery if the $R = \mathrm{round}\,(5n\log n)$ runs have found all the dictionary elements, and we report the empirical success rate over the 10 instances.

**Result** From our simulations, Riemannian subgradient descent succeeds in recovering the dictionary as long as $m \geq Cn^2$ (Fig. 2), across different sparsity level $\theta$. The dependency on $n$ is consistent with our theory and suggests that the actual sample complexity requirement for guaranteed recovery might be even lower than $\widetilde{O}(n^4)$ we established.[3] The $\widetilde{O}(n^2)$ rate we observe also matches the results based on the SOS method (Barak et al., 2015; Ma et al., 2016; Schramm & Steurer, 2017). Moreover, the problem seems to become harder when $\theta$ grows, evident from the observation that the success transition threshold being pushed to the right.

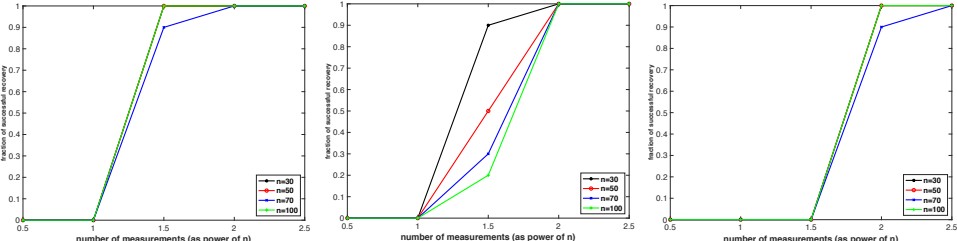

Figure 1: Empirical success rates of recovery of the Riemannian subgradient descent with $R = 5n \log n$ runs, averaged over 10 instances. Left to right: identitiy dictionaries with $\theta = 0.1, 0.3, 0.5$. See Appendix G for the results on orthogonal dictionaries, which have qualitatively the same behaviors.

**Additional experiments** A faster alternative algorithm for large-scale instances is tested in Appendix H. A complementary experiment on real images is included as Appendix I.

## 6 CONCLUSION AND FUTURE DIRECTIONS

This paper presents the first theoretical guarantee for orthogonal dictionary learning using subgradient descent on a natural $\ell_1$ minimization formulation. Along the way, we develop tools for analyzing the optimization landscape of nonconvex nonsmooth functions, which could be of broader interest.

For futute work, there is an $O(n^2)$ sample complexity gap between what we established in Theorem 3.1, and what we observed in the simulations alongside previous results based on the SOS method (Barak et al., 2015; Ma et al., 2016; Schramm & Steurer, 2017). As our main geometric result Theorem 3.6 already achieved tight bounds on the directional derivatives, further sample complexity improvement could potentially come out of utilizing second-order information such as the strong negative curvature (Lemma B.2), or careful algorithm-dependent analysis.

While our result applies only to (complete) orthogonal dictionaries, a natural question is whether we can generalize to overcomplete dictionaries. To date the only known provable algorithms for learning overcomplete dictionaries in the linear sparsity regime are based on the SOS method (Barak et al., 2015; Ma et al., 2016; Schramm & Steurer, 2017). We believe that our nonsmooth analysis has the potential of handling over-complete dictionaries, as for reasonably well-conditioned overcomplete dictionaries $\boldsymbol{A}$, each $\boldsymbol{a}_i$ (columns of $\boldsymbol{A}$) makes $\boldsymbol{a}^\top \boldsymbol{A}$ approximately 1-sparse and so $\boldsymbol{a}_i^\top \boldsymbol{A}\boldsymbol{X}$ gives noisy estimate of a certain row of $\boldsymbol{X}$. So the same formulation as Eq. (1.1) intuitively still works. We would like to leave that to future work.

Nonsmooth phase retrieval and deep networks with ReLU mentioned in Section 1.1 are examples of many nonsmooth, nonconvex problems encountered in practice. Most existing theoretical results on these problems tend to be technically vague about handling the nonsmooth points: they either prescribe a rule for choosing a subgradient element, which effectively disconnects theory and practice because numerical testing of nonsmooth points is often not reliable, or ignore the nonsmooth points altogether, assuming that practically numerical methods would never touch these points—this sounds intuitive but no formalism on this appears in the relevant literature yet. Besides our work, (Laurent & von Brecht, 2017; Kakade & Lee, 2018) also warns about potential problems of ignoring nonsmooth points when studying optimization of nonsmooth functions in machine learning.

---

[3]The $\widetilde{O}(\cdot)$ notation ignores the dependency on logarithmic terms and other factors.

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

## A TECHNICAL TOOLS

### A.1 HAUSDORFF DISTANCE

We need the Hausdorff metric to measure differences between nonempty sets. For any set $X$ and a point $\boldsymbol{p}$ in $\mathbb{R}^n$, the point-to-set distance is defined as

$$\mathrm{d}\left(\boldsymbol{q}, X\right) \doteq \inf_{\boldsymbol{x} \in X} \|\boldsymbol{x} - \boldsymbol{p}\|. \tag{A.1}$$

For any two sets $X_1, X_2 \in \mathbb{R}^n$, the Hausdorff distance is defined as

$$\mathrm{d}_{\mathrm{H}}\left(X_1, X_2\right) \doteq \sup\left\{ \sup_{\boldsymbol{x}_1 \in X_1} \mathrm{d}\left(\boldsymbol{x}_1, X_2\right), \sup_{\boldsymbol{x}_2 \in X_2} \mathrm{d}\left(\boldsymbol{x}_2, X_1\right) \right\}. \tag{A.2}$$

When $X_1$ is a singleton, say $X_1 = \{\boldsymbol{p}\}$. Then

$$\mathrm{d}_{\mathrm{H}}\left(\{\boldsymbol{p}\}, X_2\right) = \sup_{\boldsymbol{x}_2 \in X_2} \|\boldsymbol{x}_2 - \boldsymbol{p}\|. \tag{A.3}$$

Moreover, for any sets $X_1, X_2, Y_1, Y_2 \subset \mathbb{R}^n$,

$$\mathrm{d}_{\mathrm{H}}\left(X_1 + Y_1, X_2 + Y_2\right) \leq \mathrm{d}_{\mathrm{H}}\left(X_1, X_2\right) + \mathrm{d}_{\mathrm{H}}\left(Y_1, Y_2\right). \tag{A.4}$$

On the sets of nonempty, compact subsets of $\mathbb{R}^n$, the Hausdorff metric is a valid metric; particularly, it obeys the triangular inequality: for nonempty, compact subsets $X, Y, Z \subset \mathbb{R}^n$,

$$\mathrm{d}_{\mathrm{H}}\left(X, Z\right) \leq \mathrm{d}_{\mathrm{H}}\left(X, Y\right) + \mathrm{d}_{\mathrm{H}}\left(Y, Z\right). \tag{A.5}$$

See, e.g., Sec. 7.1 of Sternberg (2013) for a proof.

**Lemma A.1** (Restatement of Lemma A.1). *For convex compact sets $X, Y \subset \mathbb{R}^n$, we have*

$$\mathrm{d}_{\mathrm{H}}\left(X, Y\right) = \sup_{\boldsymbol{u} \in \mathbb{S}^{n-1}} |h_X(\boldsymbol{u}) - h_Y(\boldsymbol{u})|, \tag{A.6}$$

*where $h_S(\boldsymbol{u}) \doteq \sup_{\boldsymbol{x} \in S} \langle \boldsymbol{x}, \boldsymbol{u} \rangle$ is the support function associated with the set $S$.*

### A.2 SUB-GAUSSIAN RANDOM MATRICES AND PROCESSES

**Proposition A.2** (Talagrand's comparison inequality, Corollary 8.6.3 and Exercise 8.6.5 of Vershynin (2018)). *Let $\{X_x\}_{x \in T}$ be a zero-mean random process on a subset $T \subset \mathbb{R}^n$. Assume that for all $\boldsymbol{x}, \boldsymbol{y} \in T$ we have*

$$\|X_{\boldsymbol{x}} - X_{\boldsymbol{y}}\|_{\psi_2} \leq K \|\boldsymbol{x} - \boldsymbol{y}\|. \tag{A.7}$$

*Then, for any $t > 0$*

$$\sup_{\boldsymbol{x} \in T} |X_{\boldsymbol{x}}| \leq CK\left[w(T) + t \cdot \mathrm{rad}(T)\right] \tag{A.8}$$

*with probability at least $1 - 2\exp\left(-t^2\right)$. Here $w(T) \doteq \mathbb{E}_{\boldsymbol{g} \sim \mathcal{N}(\boldsymbol{0}, \boldsymbol{I})} \sup_{x \in T} \langle \boldsymbol{x}, \boldsymbol{g} \rangle$ is the Gaussian width of $T$ and $\mathrm{rad}(T) = \sup_{x \in T} \|\boldsymbol{x}\|$ is the radius of $T$.*

**Proposition A.3** (Deviation inequality for sub-Gaussian matrices, Theorem 9.1.1 and Exercise 9.1.8 of Vershynin (2018))**.** *Let $\boldsymbol{A}$ be an $n \times m$ matrix whose rows $\boldsymbol{A}^i$'s are independent, isotropic, and sub-Gaussian random vectors in $\mathbb{R}^m$. Then for any subset $T \subset \mathbb{R}^m$, we have*

$$\mathbb{P}\left[\sup_{\boldsymbol{x} \in T} \left|\|\boldsymbol{A}\boldsymbol{x}\| - \sqrt{n}\|\boldsymbol{x}\|\right| > CK^2 \left[w(T) + t \cdot \mathrm{rad}(T)\right]\right] \leq 2\exp\left(-t^2\right). \tag{A.9}$$

*Here $K = \max_i \left\|\boldsymbol{A}^i\right\|_{\psi_2}$.*

# B  PROOFS FOR SECTION 3.1

## B.1  PROOF OF PROPOSITION 3.2

We have

$$\mathbb{E}[f](\boldsymbol{q}) = \sqrt{\frac{\pi}{2}} \cdot \mathbb{E}\left[\left|\boldsymbol{q}^\top \boldsymbol{x}\right|\right] = \sqrt{\frac{\pi}{2}} \cdot \mathbb{E}\left[\left|\boldsymbol{q}_\Omega^\top \boldsymbol{x}_\Omega\right|\right] = \mathbb{E}\left[\|\boldsymbol{q}_\Omega\|_2\right], \tag{B.1}$$

where the last equality is obtained by conditioning on $\Omega$ and the fact that $\mathbb{E}_{Z \sim \mathsf{N}(0,\sigma^2)}[|Z|] = \sqrt{2/\pi}\sigma$.

The subdifferential expression comes from

$$\partial\|\boldsymbol{q}_\Omega\|_2 = \begin{cases} \dfrac{\boldsymbol{q}_\Omega}{\|\boldsymbol{q}_\Omega\|_2}, & \boldsymbol{q}_\Omega \neq 0; \\ \{\boldsymbol{v}_\Omega : \|\boldsymbol{v}_\Omega\|_2 \leq 1\}, & \boldsymbol{q}_\Omega = 0, \end{cases} \tag{B.2}$$

and the fact that $\partial\mathbb{E}[f](\boldsymbol{q}) = \partial\mathbb{E}[\|\boldsymbol{q}_\Omega\|_2] = \mathbb{E}[\partial\|\boldsymbol{q}_\Omega\|_2]$ as the sub-differential and expectation can be exchanged for convex functions (Hiriart-Urruty & Lemarchal, 2001). By the same exchangability result, we also have $\mathbb{E}[\partial f(\boldsymbol{q})] = \partial\mathbb{E}[f](\boldsymbol{q})$.

## B.2  PROOF OF PROPOSITION 3.3

We first show that points in the claimed set are indeed stationary points by taking the choice $\boldsymbol{v}_\Omega = \boldsymbol{0}$ in Eq. (3.5), giving the subgradient choice $\mathbb{E}[\partial f](\boldsymbol{q}) = \mathbb{E}[\boldsymbol{q}_\Omega/\|\boldsymbol{q}_\Omega\|_2 \, \mathbb{1}\{\boldsymbol{q}_\Omega \neq 0\}]$. Let $\boldsymbol{q} \in \mathcal{S}$ and such that $\|\boldsymbol{q}\|_0 = k$. For all $j \in \mathrm{supp}(\boldsymbol{q})$, we have

$$\boldsymbol{e}_j^T \mathbb{E}[\partial f](\boldsymbol{q}) = \theta q_j \cdot \mathbb{E}_\Omega \frac{1}{\|\boldsymbol{q}_\Omega\|} \mathbb{1}\{j \in \Omega\} \tag{B.3}$$

$$= \theta q_j \cdot \left[\sum_{i=1}^k \theta^{i-1}(1-\theta)^{k-i} \cdot \sqrt{\frac{k}{i}}\right] \tag{B.4}$$

$$= q_j \cdot \sum_{i=1}^k \theta^i(1-\theta)^{k-i} \cdot \frac{1}{\sqrt{i}} \doteq c(\theta,k)q_j \tag{B.5}$$

On the other hand, for all $j \notin \mathrm{supp}(\boldsymbol{q})$, we always have $[\boldsymbol{q}_\Omega]_j = 0$, so $\boldsymbol{e}_j^\top \mathbb{E}[\partial f](\boldsymbol{q}) = 0$. Therefore, we have that $\mathbb{E}[\partial f](\boldsymbol{q}) = c(\theta,k)\boldsymbol{q}$, and so

$$(\boldsymbol{I} - \boldsymbol{q}\boldsymbol{q}^\top)\mathbb{E}[\partial f](\boldsymbol{q}) = c(\theta,k)\boldsymbol{q} - c(\theta,k)\boldsymbol{q} = \boldsymbol{0}. \tag{B.6}$$

Therefore $\boldsymbol{q} \in \mathcal{S}$ is stationary. To see that $\{\pm\boldsymbol{e}_i : i \in [n]\}$ are the global minima, note that for all $\boldsymbol{q} \in \mathbb{S}^{n-1}$, we have

$$\mathbb{E}[f](\boldsymbol{q}) = \mathbb{E}[\|\boldsymbol{q}_\Omega\|_2] \geq \mathbb{E}\left[\|\boldsymbol{q}_\Omega\|_2^2\right] = \theta. \tag{B.7}$$

Equality holds if and only if $\|\boldsymbol{q}_\Omega\|_2 \in \{0,1\}$ almost surely, which is only satisfied at $\boldsymbol{q} \in \{\pm\boldsymbol{e}_i : i \in [n]\}$.

To see that the other $\boldsymbol{q}$'s are saddles, we only need to show that there exists a tangent direction along which $\boldsymbol{q}$ is local max. Indeed, for any other $\boldsymbol{q}$, there exists at least two non-zero entries (with equal absolute value): WLOG assume that $q_1 = q_n > 0$. Using the reparametrization in Appendix B.3 and applying Lemma B.2, we get that $\mathbb{E}[f](\boldsymbol{q})$ is directionally differentiable along $[-\boldsymbol{q}_{-n}; \frac{1-q_n^2}{q_n}]$, with derivative zero (necessarily, because $\boldsymbol{0} \in \mathbb{E}[\partial_R f](\boldsymbol{q})$) and strictly negative second derivative.

Therefore $\mathbb{E}[f](\boldsymbol{q})$ is locally maximized at $\boldsymbol{q}$ along this tangent direction, which shows that $\boldsymbol{q}$ is a saddle point.

The other direction (all other points are not stationary) is implied by Theorem 3.4, which guarantees that $\boldsymbol{0} \notin \mathbb{E}[\partial_R f](\boldsymbol{q})$ whenever $\boldsymbol{q} \notin \mathcal{S}$. Indeed, as long as $\boldsymbol{q} \notin \mathcal{S}$, $\boldsymbol{q}$ has a max absolute value coordinate (say $n$) and another non-zero coordinate with strictly smaller absolute value (say $j$). For this pair of indices, the proof of Theorem 3.4(a) goes through for index $j$ (even if $\boldsymbol{q} \in \mathcal{S}_{\zeta_0}^{(n+)}$ does not necessarily hold because the max index might not be unique), which implies that $\boldsymbol{0} \notin \mathbb{E}[\partial_R f](\boldsymbol{q})$.

### B.3 Reparametrization

For analysis purposes, we introduce the reparametrization $\boldsymbol{w} = \boldsymbol{q}_{1:(n-1)}$ in the region $\mathcal{S}_0^{(n+)}$, following (Sun et al., 2015) . With this reparametrization, the problem becomes

$$\text{minimize}_{\boldsymbol{w} \in \mathbb{R}^{n-1}} \ g(\boldsymbol{w}) \doteq \sqrt{\frac{\pi}{2}} \cdot \frac{1}{m} \left\| \left[ \boldsymbol{w}; \sqrt{1 - \|\boldsymbol{w}\|^2} \right]^\top \boldsymbol{X} \right\|_1 \quad \text{subject to } \|\boldsymbol{w}\| \leq \sqrt{\frac{n-1}{n}}. \tag{B.8}$$

The constraint comes from the fact that $q_n \geq 1/\sqrt{n}$ and thus $\|\boldsymbol{w}\| \leq \sqrt{(n-1)/n}$.

**Lemma B.1.** *We have*

$$\mathbb{E}_{\boldsymbol{x} \sim_{iid} \mathrm{BG}(\theta)} g(\boldsymbol{w}) = (1-\theta)\,\mathbb{E}_\Omega \|\boldsymbol{w}_\Omega\| + \theta \mathbb{E}_\Omega \sqrt{1 - \|\boldsymbol{w}_{\Omega^c}\|^2}. \tag{B.9}$$

**Proof.** Direct calculation gives

$$\mathbb{E}_{\boldsymbol{x} \sim_{iid} \mathrm{BG}(\theta)} g(\boldsymbol{w}) \tag{B.10}$$

$$= \mathbb{E}_{\Omega \sim_{iid} \mathrm{Ber}(\theta), \omega \sim \mathrm{Ber}(\theta)} \mathbb{E}_{\boldsymbol{z} \sim_{iid} \mathcal{N}(0,1), z \sim \mathcal{N}(0,1)} \left| \left[ \boldsymbol{w}; \sqrt{1 - \|\boldsymbol{w}\|^2} \right]^\top ([\Omega; \omega] \odot [\boldsymbol{z}; z]) \right| \tag{B.11}$$

$$= (1-\theta)\,\mathbb{E}_{\Omega \sim_{iid} \mathrm{Ber}(\theta)} \mathbb{E}_{\boldsymbol{z} \sim_{iid} \mathcal{N}(0,1)} \left| \boldsymbol{w}_\Omega^\top \boldsymbol{z} \right|$$
$$+ \theta \mathbb{E}_{\Omega \sim_{iid} \mathrm{Ber}(\theta)} \mathbb{E}_{\boldsymbol{z} \sim_{iid} \mathcal{N}(0,1), z \sim \mathcal{N}(0,1)} \left| \boldsymbol{w}_\Omega^\top \boldsymbol{z} + \sqrt{1 - \|\boldsymbol{w}\|^2} z \right| \tag{B.12}$$

$$= (1-\theta)\,\mathbb{E}_{\Omega \sim_{iid} \mathrm{Ber}(\theta)} \|\boldsymbol{w}_\Omega\| + \theta \mathbb{E}_{\Omega \sim_{iid} \mathrm{Ber}(\theta)} \sqrt{1 - \|\boldsymbol{w}_{\Omega^c}\|^2}, \tag{B.13}$$

as claimed. ∎

**Lemma B.2** (Negative-curvature region). *For all unit vector $\boldsymbol{v} \in \mathbb{S}^{n-1}$ and all $s \in (0,1)$, let*

$$h_{\boldsymbol{v}}(s) \doteq \mathbb{E}[g](s\boldsymbol{v}) \tag{B.14}$$

*it holds that*

$$\nabla^2 h_{\boldsymbol{v}}(s) \leq -\theta(1-\theta). \tag{B.15}$$

*In other words, for all $\boldsymbol{w} \neq \boldsymbol{0}$, $\pm \boldsymbol{w}/\|\boldsymbol{w}\|$ is a direction of negative curvature.*

**Proof.** By Lemma B.1,

$$h_{\boldsymbol{v}}(s) = (1-\theta)\,s\mathbb{E}_\Omega \|\boldsymbol{v}_\Omega\| + \theta \mathbb{E}_\Omega \sqrt{1 - s^2 \|\boldsymbol{v}_{\Omega^c}\|^2}. \tag{B.16}$$

For $s \in (0,1)$, $h_{\boldsymbol{v}}(s)$ is twice differentiable, and we have

$$\nabla^2 h_{\boldsymbol{v}}(s) = -\theta \mathbb{E}_\Omega \frac{\|\boldsymbol{v}_{\Omega^c}\|^2}{\left(1 - s^2 \|\boldsymbol{v}_{\Omega^c}\|^2\right)^{3/2}} \tag{B.17}$$

$$\leq -\theta \mathbb{E}_\Omega \|\boldsymbol{v}_{\Omega^c}\|^2 = -\theta(1-\theta), \tag{B.18}$$

completing the proof. ∎

**Lemma B.3** (Inward gradient). *For any $\boldsymbol{w}$ with $\|\boldsymbol{w}\|^2 + \|\boldsymbol{w}\|_\infty^2 \le 1$,*

$$D_{\boldsymbol{w}/\|\boldsymbol{w}\|}^c \mathbb{E}\left[g\right](\boldsymbol{w}) \ge \theta\left(1 - \theta\right)\left(1/\sqrt{1 + \|\boldsymbol{w}\|_\infty^2 /\|\boldsymbol{w}\|^2} - \|\boldsymbol{w}\|\right). \tag{B.19}$$

**Proof.** For any unit vector $\boldsymbol{v} \in \mathbb{R}^{n-1}$, define $h_{\boldsymbol{v}}(t) \doteq \mathbb{E}\left[g\right](t\boldsymbol{v})$ for $t \in (0, 1)$. We have from Lemma B.1

$$h_{\boldsymbol{v}}(t) = (1 - \theta)\, t\mathbb{E}_\Omega \|\boldsymbol{v}_\Omega\| + \theta\mathbb{E}_\Omega \sqrt{1 - t^2 \|\boldsymbol{v}_\Omega^c\|^2}. \tag{B.20}$$

Moreover,

$$\nabla_t h_{\boldsymbol{v}}(t) = (1 - \theta)\, \mathbb{E}_\Omega \|\boldsymbol{v}_\Omega\| - \theta\mathbb{E}_\Omega \frac{t \|\boldsymbol{v}_\Omega^c\|^2}{\sqrt{1 - t^2 \|\boldsymbol{v}_\Omega^c\|^2}} \tag{B.21}$$

$$= (1 - \theta)\, \mathbb{E}_\Omega \frac{\|\boldsymbol{v}_\Omega\|^2}{\|\boldsymbol{v}_\Omega\|} - \theta\mathbb{E}_\Omega \frac{t \|\boldsymbol{v}_\Omega^c\|^2}{\sqrt{1 - t^2 \|\boldsymbol{v}_\Omega^c\|^2}} \quad \left(\text{assuming } \frac{0}{0} \doteq 0\right) \tag{B.22}$$

$$= (1 - \theta)\sum_{i=1}^{n-1} \mathbb{E}_\Omega \frac{v_i^2 \mathbb{1}\left\{i \in \Omega\right\}}{\sqrt{v_i^2 \mathbb{1}\left\{i \in \Omega\right\} + \|\boldsymbol{v}_{\Omega\setminus i}\|^2}} - \theta\sum_{i=1}^{n-1} \mathbb{E}_\Omega \frac{t v_i^2 \mathbb{1}\left\{i \notin \Omega\right\}}{\sqrt{1 - t^2 v_i^2 \mathbb{1}\left\{i \notin \Omega\right\} - t^2 \|\boldsymbol{v}_{\Omega^c\setminus i}\|^2}} \tag{B.23}$$

$$= \theta\left(1 - \theta\right)\sum_{i=1}^{n-1} \mathbb{E}_\Omega \left[\frac{v_i^2}{\sqrt{v_i^2 + \|\boldsymbol{v}_{\Omega\setminus i}\|^2}} - \frac{t v_i^2}{\sqrt{1 - t^2 v_i^2 - t^2 \|\boldsymbol{v}_{\Omega^c\setminus i}\|^2}}\right] \tag{B.24}$$

$$= \theta\left(1 - \theta\right) t\sum_{i=1}^{n-1} v_i^2 \mathbb{E}_\Omega \left[\frac{1}{\sqrt{t^2 v_i^2 + t^2 \|\boldsymbol{v}_{\Omega\setminus i}\|^2}} - \frac{1}{\sqrt{1 - t^2 v_i^2 - t^2 \|\boldsymbol{v}_{\Omega^c\setminus i}\|^2}}\right] \tag{B.25}$$

$$= \theta\left(1 - \theta\right) t\sum_{i=1}^{n-1} v_i^2 \mathbb{E}_\Omega \left[\frac{1}{\sqrt{t^2 v_i^2 + t^2 \|\boldsymbol{v}_{\Omega\setminus i}\|^2}} - \frac{1}{\sqrt{1 - t^2 \|\boldsymbol{v}\|^2 + t^2 \|\boldsymbol{v}_{\Omega\setminus i}\|^2}}\right]. \tag{B.26}$$

We are interested in the regime of $t$ so that

$$1 - t^2 \|\boldsymbol{v}\|^2 \ge t^2 \|\boldsymbol{v}\|_\infty^2 \implies t \le 1/\sqrt{1 + \|\boldsymbol{v}\|_\infty^2}. \tag{B.27}$$

So $\nabla_t h_{\boldsymbol{v}}(t) \ge 0$ holds always for $t \le 1/\sqrt{1 + \|\boldsymbol{v}\|_\infty^2}$.

By Lemma B.2, $\nabla^2 h_{\boldsymbol{v}}(t) \le -\theta\left(1 - \theta\right)$ over $t \in (0, 1)$, which implies

$$\langle \nabla_t h_{\boldsymbol{v}}(t_1) - \nabla_t h_{\boldsymbol{v}}(t_2), t_1 - t_2 \rangle \le -\theta\left(1 - \theta\right)\left(t_1 - t_2\right)^2. \tag{B.28}$$

Taking $t_1 = 1/\sqrt{1 + \|\boldsymbol{v}\|_\infty^2}$ and considering $t_2 \in [0, t_1]$, we have

$$\nabla_t h_{\boldsymbol{v}}(t_2) \ge \nabla_t h_{\boldsymbol{v}}(t_1) + \theta\left(1 - \theta\right)\left(t_1 - t_2\right) \ge \theta\left(1 - \theta\right)\left(1/\sqrt{1 + \|\boldsymbol{v}\|_\infty^2} - t_2\right). \tag{B.29}$$

For any $\boldsymbol{w}$, applying the above result to the unit vector $\boldsymbol{w}/\|\boldsymbol{w}\|$ and recognizing that $\nabla_t h_{\boldsymbol{w}/\|\boldsymbol{w}\|}(t) = D_{\boldsymbol{w}/\|\boldsymbol{w}\|} g(\boldsymbol{w}) = D_{\boldsymbol{w}/\|\boldsymbol{w}\|}^c g(\boldsymbol{w})$, we complete the proof. ∎

## B.4 Proof of Theorem 3.4

We first show Eq. (3.9) using the reparametrization in Appendix B.3. We have

$$\langle \partial_R f(\boldsymbol{q}), \boldsymbol{q} - \boldsymbol{e}_n \rangle = \langle \partial f(\boldsymbol{q}), q_n \boldsymbol{q} - \boldsymbol{e}_n \rangle = q_n \langle \partial g(\boldsymbol{w}), \boldsymbol{w} \rangle, \tag{B.30}$$

where the second equality follows by differentiating $g$ via the chain rule. Now, by Lemma B.3,

$$q_n \langle \mathbb{E}[\partial g](\boldsymbol{w}), \boldsymbol{w} \rangle \geq \|\boldsymbol{w}\| \, \theta \, (1 - \theta) \cdot q_n \left( \frac{\|\boldsymbol{w}\|}{\sqrt{\|\boldsymbol{w}\|^2 + \|\boldsymbol{w}\|_\infty^2}} - \|\boldsymbol{w}\| \right). \tag{B.31}$$

For each radial direction $\boldsymbol{v} \doteq \boldsymbol{w}/\|\boldsymbol{w}\|$, consider points of the form $t\boldsymbol{v}$ with $t \leq 1/\sqrt{1 + \|\boldsymbol{v}\|_\infty^2}$. Obviously, the function

$$\hbar(t) \doteq q_n(t\boldsymbol{v}) \left( \frac{\|t\boldsymbol{v}\|}{\sqrt{\|t\boldsymbol{v}\|^2 + \|t\boldsymbol{v}\|_\infty^2}} - \|t\boldsymbol{v}\| \right) = q_n(t\boldsymbol{v}) \left( \frac{1}{\sqrt{1 + \|\boldsymbol{v}\|_\infty^2}} - t \right) \tag{B.32}$$

is monotonically decreasing wrt $t$. Thus, to derive a lower bound, it is enough to consider the largest $t$ allowed. In $\mathcal{S}_{\zeta_0}^{(n+)}$, the limit amounts to requiring $q_n^2/\|\boldsymbol{w}\|_\infty^2 = 1 + \zeta_0$,

$$1 - t_0^2 = t_0^2 \|\boldsymbol{v}\|_\infty^2 (1 + \zeta_0) \implies t_0 = \frac{1}{\sqrt{1 + (1 + \zeta_0) \|\boldsymbol{v}\|_\infty^2}}. \tag{B.33}$$

So for any fixed $\boldsymbol{v}$ and all allowed $t$ for points in $\mathcal{S}_{\zeta_0}^{(n+)}$, a uniform lower bound is

$$q_n(t_0 \boldsymbol{v}) \left( \frac{1}{\sqrt{1 + \|\boldsymbol{v}\|_\infty^2}} - t_0 \right) \tag{B.34}$$

$$\geq \frac{1}{\sqrt{n}} \left( \frac{1}{\sqrt{1 + \|\boldsymbol{v}\|_\infty^2}} - \frac{1}{\sqrt{1 + (1 + \zeta_0) \|\boldsymbol{v}\|_\infty^2}} \right) \geq \frac{1}{8\sqrt{n}} \zeta_0 \|\boldsymbol{v}\|_\infty^2 \geq \frac{1}{8} \zeta_0 n^{-3/2}. \tag{B.35}$$

So we conclude that for all $\boldsymbol{q} \in \mathcal{S}_{\zeta_0}^{(n+)}$,

$$\langle \mathbb{E}[\partial f](\boldsymbol{q}), q_n \boldsymbol{q} - \boldsymbol{e}_n \rangle \geq \frac{1}{8} \theta (1 - \theta) \zeta_0 n^{-3/2} \|\boldsymbol{w}\| = \frac{1}{8} \theta (1 - \theta) \zeta_0 n^{-3/2} \|\boldsymbol{q}_{-n}\|. \tag{B.36}$$

We now turn to showing Eq. (3.8). For $\boldsymbol{e}_j$ with $q_j \neq 0$,

$$\frac{1}{q_j} \boldsymbol{e}_j^\top \mathbb{E}[\partial f](\boldsymbol{q}) = \frac{1}{q_j} \boldsymbol{e}_j^\top \mathbb{E}_\Omega \left[ \frac{\boldsymbol{q}_\Omega}{\|\boldsymbol{q}_\Omega\|} \mathbb{1}\{\boldsymbol{q}_\Omega \neq \boldsymbol{0}\} + \{\boldsymbol{v}_\Omega : \|\boldsymbol{v}_\Omega\| \leq 1\} \mathbb{1}\{\boldsymbol{q}_\Omega = \boldsymbol{0}\} \right]$$

$$= \frac{1}{q_j} \mathbb{E}_\Omega \left[ \frac{\langle \boldsymbol{q}_\Omega, \boldsymbol{e}_j \rangle}{\|\boldsymbol{q}_\Omega\|} \mathbb{1}\{\boldsymbol{q}_\Omega \neq \boldsymbol{0}\} \right] = \frac{1}{q_j} \theta q_j \mathbb{E}_\Omega \left[ \frac{1}{\|\boldsymbol{q}_\Omega\|} \mathbb{1}\{j \in \Omega\} \right] = \theta \mathbb{E} \left[ \frac{1}{\sqrt{q_j^2 + \|\boldsymbol{q}_{\Omega \setminus \{j\}}\|^2}} \right] \geq \theta. \tag{B.37}$$

So for all $j$ with $q_j \neq 0$, we have

$$\left\langle \mathbb{E}[\partial_R f](\boldsymbol{q}), \frac{1}{q_j} \boldsymbol{e}_j - \frac{1}{q_n} \boldsymbol{e}_n \right\rangle$$

$$= \left\langle (\boldsymbol{I} - \boldsymbol{q}\boldsymbol{q}^\top) \mathbb{E}[\partial f](\boldsymbol{q}), \frac{1}{q_j} \boldsymbol{e}_j - \frac{1}{q_n} \boldsymbol{e}_n \right\rangle \tag{B.38}$$

$$= \left\langle \mathbb{E}[\partial f](\boldsymbol{q}), \frac{1}{q_j} \boldsymbol{e}_j - \frac{1}{q_n} \boldsymbol{e}_n \right\rangle \tag{B.39}$$

$$= \theta \mathbb{E}_\Omega \left[ \frac{1}{\sqrt{q_j^2 + \|\boldsymbol{q}_{\Omega \setminus \{j\}}\|^2}} \right] - \theta \mathbb{E}_\Omega \left[ \frac{1}{\sqrt{q_n^2 + \|\boldsymbol{q}_{\Omega \setminus \{n\}}\|^2}} \right] \tag{B.40}$$

$$= \theta^2 \mathbb{E}_\Omega \left[ \frac{1}{\sqrt{q_j^2 + q_n^2 + \|\boldsymbol{q}_{\Omega \setminus \{j,n\}}\|^2}} \right] + \theta (1 - \theta) \mathbb{E}_\Omega \left[ \frac{1}{\sqrt{q_j^2 + \|\boldsymbol{q}_{\Omega \setminus \{j,n\}}\|^2}} \right]$$

$$- \theta^2 \mathbb{E}_\Omega \left[ \frac{1}{\sqrt{q_j^2 + q_n^2 + \left\| \boldsymbol{q}_{\Omega \setminus \{j,n\}} \right\|^2}} \right] - \theta \left( 1 - \theta \right) \mathbb{E}_\Omega \left[ \frac{1}{\sqrt{q_n^2 + \left\| \boldsymbol{q}_{\Omega \setminus \{j,n\}} \right\|^2}} \right] \quad \text{(B.41)}$$

$$= \theta \left( 1 - \theta \right) \mathbb{E}_\Omega \left[ \frac{1}{\sqrt{q_j^2 + \left\| \boldsymbol{q}_{\Omega \setminus \{j,n\}} \right\|^2}} - \frac{1}{\sqrt{q_n^2 + \left\| \boldsymbol{q}_{\Omega \setminus \{j,n\}} \right\|^2}} \right] \quad \text{(B.42)}$$

$$= \theta \left( 1 - \theta \right) \mathbb{E}_\Omega \int_{q_j^2}^{q_n^2} \frac{1}{2 \left( t + \left\| \boldsymbol{q}_{\Omega \setminus \{j,n\}} \right\|^2 \right)^{3/2}} \, dt \quad \text{(B.43)}$$

$$\geq \theta \left( 1 - \theta \right) \frac{1}{2} \left( q_n^2 - q_j^2 \right) \geq \frac{1}{2} \theta \left( 1 - \theta \right) \left( q_n^2 - \left\| \boldsymbol{q}_{-n} \right\|_\infty^2 \right) \geq \frac{1}{2} \theta \left( 1 - \theta \right) \frac{\zeta_0}{1 + \zeta_0} q_n^2 \quad \text{(B.44)}$$

$$=\geq \frac{1}{2n} \theta \left( 1 - \theta \right) \frac{\zeta_0}{1 + \zeta_0} \quad \text{(B.45)}$$

completing the proof.

## C   PROOFS FOR SECTION 3.2

### C.1   COVERING IN THE $\mathrm{d}_\mathbb{E}$ METRIC

For any $\theta \in (0, 1)$, define

$$\mathrm{d}_{\mathbb{E},\theta} \left( \boldsymbol{p}, \boldsymbol{q} \right) \doteq \mathbb{E} \left[ \mathbb{1} \left\{ \text{sign} \left( \boldsymbol{p}^\top \boldsymbol{x} \right) \neq \text{sign} \left( \boldsymbol{q}^\top \boldsymbol{x} \right) \right\} \right] \quad \text{with } \boldsymbol{x} \sim_{iid} \text{BG}(\theta). \quad \text{(C.1)}$$

We stress that this notion always depend on $\theta$, and we will omit the subscript $\theta$ when no confusion is expected. This indeed defines a metric on subsets of $\mathbb{S}^{n-1}$.

**Lemma C.1.** *Over any subset of $\mathbb{S}^{n-1}$ with a consistent support pattern, $\mathrm{d}_\mathbb{E}$ is a valid metric.*

**Proof.** Recall that $\angle \left( \boldsymbol{x}, \boldsymbol{y} \right) \doteq \arccos \langle \boldsymbol{x}, \boldsymbol{y} \rangle$ defines a valid metric on $\mathbb{S}^{n-1}$.[4] In particular, the triangular inequality holds. For $\mathrm{d}_\mathbb{E}$ and $\boldsymbol{p}, \boldsymbol{q} \in \mathbb{S}^{n-1}$ with the same support pattern, we have

$$\mathrm{d}_\mathbb{E} \left( \boldsymbol{p}, \boldsymbol{q} \right) = \mathbb{E} \left[ \mathbb{1} \left\{ \text{sign} \left( \boldsymbol{p}^\top \boldsymbol{x} \right) \neq \text{sign} \left( \boldsymbol{q}^\top \boldsymbol{x} \right) \right\} \right] \quad \text{(C.2)}$$

$$= \mathbb{E}_\Omega \mathbb{E}_{\boldsymbol{z} \sim \mathsf{N}(\boldsymbol{0}, \boldsymbol{I})} \mathbb{1} \left\{ \text{sign} \left( \boldsymbol{p}_\Omega^\top \boldsymbol{z} \right) \neq \text{sign} \left( \boldsymbol{q}_\Omega^\top \boldsymbol{z} \right) \right\} \quad \text{(C.3)}$$

$$= \mathbb{E}_\Omega \left( \mathbb{E}_{\boldsymbol{z}} \mathbb{1} \left\{ \boldsymbol{p}_\Omega^\top \boldsymbol{z} \boldsymbol{q}_\Omega \boldsymbol{z} < 0 \right\} + \mathbb{E}_{\boldsymbol{z}} \mathbb{1} \left\{ \boldsymbol{p}_\Omega^\top \boldsymbol{z} = 0 \text{ or } \boldsymbol{q}_\Omega^\top \boldsymbol{z} = 0, \text{not both} \right\} \right) \quad \text{(C.4)}$$

$$= \mathbb{E}_\Omega \left( \mathbb{E}_{\boldsymbol{z}} \mathbb{1} \left\{ \boldsymbol{p}_\Omega^\top \boldsymbol{z} \boldsymbol{q}_\Omega \boldsymbol{z} < 0 \right\} \right) \quad \text{(C.5)}$$

$$= \frac{1}{\pi} \mathbb{E}_\Omega \angle \left( \boldsymbol{p}_\Omega, \boldsymbol{q}_\Omega \right), \quad \text{(C.6)}$$

where we have adopted the convention that $\angle \left( \boldsymbol{0}, \boldsymbol{v} \right) \doteq 0$ for any $\boldsymbol{v}$. It is easy to verify that $\mathrm{d}_\mathbb{E} \left( \boldsymbol{p}, \boldsymbol{q} \right) = 0 \iff \boldsymbol{p} = \boldsymbol{q}$, and $\mathrm{d}_\mathbb{E} \left( \boldsymbol{p}, \boldsymbol{q} \right) = \mathrm{d}_\mathbb{E} \left( \boldsymbol{q}, \boldsymbol{p} \right)$. To show the triangular inequality, note that for any $\boldsymbol{p}, \boldsymbol{q}$ and $\boldsymbol{r}$ with the same support pattern, $\boldsymbol{p}_\Omega, \boldsymbol{q}_\Omega$, and $\boldsymbol{r}_\Omega$ are either identically zero, or all nonzero. For the former case,

$$\angle \left( \boldsymbol{p}_\Omega, \boldsymbol{q}_\Omega \right) \leq \angle \left( \boldsymbol{p}_\Omega, \boldsymbol{r}_\Omega \right) + \angle \left( \boldsymbol{q}_\Omega, \boldsymbol{r}_\Omega \right) \quad \text{(C.7)}$$

holds trivially. For the latter, since $\angle \left( \cdot, \cdot \right)$ obeys the triangular inequality uniformly over the sphere,

$$\angle \left( \frac{\boldsymbol{p}_\Omega}{\| \boldsymbol{p}_\Omega \|}, \frac{\boldsymbol{q}_\Omega}{\| \boldsymbol{q}_\Omega \|} \right) \leq \angle \left( \frac{\boldsymbol{p}_\Omega}{\| \boldsymbol{p}_\Omega \|}, \frac{\boldsymbol{r}_\Omega}{\| \boldsymbol{r}_\Omega \|} \right) + \angle \left( \frac{\boldsymbol{q}_\Omega}{\| \boldsymbol{q}_\Omega \|}, \frac{\boldsymbol{r}_\Omega}{\| \boldsymbol{r}_\Omega \|} \right), \quad \text{(C.8)}$$

which implies

$$\angle \left( \boldsymbol{p}_\Omega, \boldsymbol{q}_\Omega \right) \leq \angle \left( \boldsymbol{p}_\Omega, \boldsymbol{r}_\Omega \right) + \angle \left( \boldsymbol{q}_\Omega, \boldsymbol{r}_\Omega \right). \quad \text{(C.9)}$$

---

[4]This fact can be proved either directly, see, e.g., page 12 of this online notes: http://www.math.mcgill.ca/drury/notes354.pdf, or by realizing that the angle equal to the geodesic length, which is the Riemmannian distance over the sphere; see, e.g., Riemannian Distance of Chapter 5 of the book O'neill (1983).

So

$$\mathbb{E}_\Omega \angle\left(\boldsymbol{p}_\Omega, \boldsymbol{q}_\Omega\right) \leq \mathbb{E}_\Omega \angle\left(\boldsymbol{p}_\Omega, \boldsymbol{r}_\Omega\right) + \mathbb{E}_\Omega \angle\left(\boldsymbol{q}_\Omega, \boldsymbol{r}_\Omega\right), \tag{C.10}$$

completing the proof. ∎

**Lemma C.2** (Vector angle inequality). *For $n \geq 2$, consider $\boldsymbol{u}, \boldsymbol{v} \in \mathbb{R}^n$ so that $\angle\left(\boldsymbol{u}, \boldsymbol{v}\right) \leq \pi/2$. It holds that*

$$\angle\left(\boldsymbol{u}, \boldsymbol{v}\right) \leq \sum_{\Omega \in \binom{[n]}{2}} \angle\left(\boldsymbol{u}_\Omega, \boldsymbol{v}_\Omega\right). \tag{C.11}$$

**Proof.** The inequality holds trivially when either of $\boldsymbol{u}, \boldsymbol{v}$ is zero. Suppose they are both nonzero and wlog assume both are normalized, i.e., $\|\boldsymbol{u}\| = \|\boldsymbol{v}\| = 1$. Then,

$$\sin^2 \angle\left(\boldsymbol{u}, \boldsymbol{v}\right) = 1 - \cos^2 \angle\left(\boldsymbol{u}, \boldsymbol{v}\right) \tag{C.12}$$

$$= \|\boldsymbol{u}\|^2 \|\boldsymbol{v}\|^2 - \langle \boldsymbol{u}, \boldsymbol{v}\rangle^2 \tag{C.13}$$

$$= \sum_{i,j:j>i} \left(u_i v_j - u_j v_i\right)^2 \quad \text{(Lagrange's identity)} \tag{C.14}$$

$$= \sum_{\Omega \in \binom{[n]}{2}} \|\boldsymbol{u}_\Omega\|^2 \|\boldsymbol{v}_\Omega\|^2 - \langle \boldsymbol{u}_\Omega, \boldsymbol{v}_\Omega\rangle^2 \tag{C.15}$$

$$= \sum_{\Omega \in \binom{[n]}{2}} \|\boldsymbol{u}_\Omega\|^2 \|\boldsymbol{v}_\Omega\|^2 \sin^2 \angle\left(\boldsymbol{u}_\Omega, \boldsymbol{v}_\Omega\right) \tag{C.16}$$

$$\leq \sum_{\Omega \in \binom{[n]}{2}} \sin^2 \angle\left(\boldsymbol{u}_\Omega, \boldsymbol{v}_\Omega\right). \tag{C.17}$$

If $\sum_{\Omega \in \binom{[n]}{2}} \angle\left(\boldsymbol{u}_\Omega, \boldsymbol{v}_\Omega\right) > \pi/2$, the claimed inequality holds trivially, as $\angle\left(\boldsymbol{u}, \boldsymbol{v}\right) \leq \pi/2$ by our assumption. Suppose $\sum_{\Omega \in \binom{[n]}{2}} \angle\left(\boldsymbol{u}_\Omega, \boldsymbol{v}_\Omega\right) \leq \pi/2$. Then,

$$\sum_{\Omega \in \binom{[n]}{2}} \sin^2 \angle\left(\boldsymbol{u}_\Omega, \boldsymbol{v}_\Omega\right) \leq \sin^2 \sum_{\Omega \in \binom{[n]}{2}} \angle\left(\boldsymbol{u}_\Omega, \boldsymbol{v}_\Omega\right) \tag{C.18}$$

by recursive application of the following inequality: $\forall \theta_1, \theta_2 \in [0, \pi/2]$ with $\theta_1 + \theta_2 \leq \pi/2$,

$$\sin^2\left(\theta_1 + \theta_2\right) = \sin^2 \theta_1 + \sin^2 \theta_2 + 2 \sin \theta_1 \sin \theta_2 \cos\left(\theta_1 + \theta_2\right) \geq \sin^2 \theta_1 + \sin^2 \theta_2. \tag{C.19}$$

So we have that when $\sum_{\Omega \in \binom{[n]}{2}} \angle\left(\boldsymbol{u}_\Omega, \boldsymbol{v}_\Omega\right) \leq \pi/2$,

$$\sin^2 \angle\left(\boldsymbol{u}, \boldsymbol{v}\right) \leq \sin^2 \sum_{\Omega \in \binom{[n]}{2}} \angle\left(\boldsymbol{u}_\Omega, \boldsymbol{v}_\Omega\right) \implies \angle\left(\boldsymbol{u}, \boldsymbol{v}\right) \leq \sum_{\Omega \in \binom{[n]}{2}} \angle\left(\boldsymbol{u}_\Omega, \boldsymbol{v}_\Omega\right), \tag{C.20}$$

as claimed. ∎

**Lemma C.3** (Covering in maximum length-2 angles). *For any $\eta \in (0, 1/3)$, there exists a subset $\mathcal{Q} \subset \mathbb{S}^{n-1}$ of size at most $(5n \log(1/\eta)/\eta)^{2n-1}$ satisfying the following: for any $\boldsymbol{p} \in \mathbb{S}^{n-1}$, there exists some $\boldsymbol{q} \in \mathcal{Q}$ such that $\angle\left(\boldsymbol{p}_\Omega, \boldsymbol{q}_\Omega\right) \leq \eta$ for all $\Omega \subset [n]$ with $|\Omega| \leq 2$.*

**Proof.** Define

$$\mathrm{d}_2(\boldsymbol{p}, \boldsymbol{q}) = \max_{|\Omega| \leq 2} \angle\left(\boldsymbol{p}_\Omega, \boldsymbol{q}_\Omega\right), \tag{C.21}$$

our goal is to give an $\eta$-covering of $\mathbb{S}^{n-1}$ in the $\mathrm{d}_2$ metric.

**Step 1** We partition $\mathbb{S}^{n-1}$ according to the support, the sign pattern, and the ordering of the non-zero elements. For each configuration, we are going to construct a covering with the same configuration of support, sign pattern, and ordering. There are no more than $3^n \cdot n!$ such configurations. Note that we only need to construct one such covering for each support size, and for each support size we can ignore the zero entries – the angle $\angle\left(\boldsymbol{p}_\Omega, \boldsymbol{q}_\Omega\right)$ is always zero when $\boldsymbol{p}, \boldsymbol{q}$ have matching support and $\Omega$ contains at least one zero index.

Therefore, the task reduces to bounding the covering number of

$$A_n = \left\{\boldsymbol{p} \in \mathbb{S}^{n-1} : \boldsymbol{p} \geq 0, \ 0 < p_1 \leq p_2 \leq \ldots \leq p_n\right\} \tag{C.22}$$

in $\mathrm{d}_2$ for all $n$.

**Step 2** We bound the covering number of $A_n$ by induction. Suppose that

$$N(A_{n'}) \leq \left( \frac{5 \log(1/\eta)}{\eta} \cdot n' \right)^{n'-1} = (C_\eta n')^{n'-1} \tag{C.23}$$

holds for all $n' \leq n - 1$. (The base case $m = 2$ clearly holds.) Let $\mathcal{C}_{n'} \subset \mathbb{S}^{n'-1}$ be the correpsonding covering sets.

We now construct a covering for $A_n$. Let $R \doteq 1/\eta = r^k$ for some $r \geq 1$ and $k$ to be determined. Consider the set

$$\mathcal{Q}_{r,k} = \left\{ \boldsymbol{q} \in \mathbb{S}^{n-1} : \frac{q_{i+1}}{q_i} \in \left\{1, r, r^2, \ldots, r^{k-1}\right\} \text{ for all } 1 \leq i \leq n-1 \right\}. \tag{C.24}$$

We claim that $\mathcal{Q}_{r,k}$ with properly chosen $(r, k)$ gives a covering of

$$A_{n,R} = \left\{ \boldsymbol{p} \in A_n : \frac{p_{i+1}}{p_i} \leq R, \, \forall i \right\} \subset A_n. \tag{C.25}$$

Indeed, we can decompose $[1, R)$ into $[1, r), [r, r^2), \ldots, [r^{k-1}, R)$. Each consecutive ratio $p_{i+1}/p_i$ falls in one of these intervals, and we choose $\boldsymbol{q}$ so that $q_{i+1}/q_i$ is the left endpoint of this interval. Such a $\boldsymbol{q}$ satisfies $\boldsymbol{q} \in \mathcal{Q}_{r,k}$ and

$$\frac{p_{i+1}/p_i}{q_{i+1}/q_i} \in [1, r) \text{ for all } i \in [n-1]. \tag{C.26}$$

By multiplying these bounds, we obtain that for all $1 \leq i < j \leq n$,

$$\frac{p_j/p_i}{q_j/q_i} \in [1, r^{n-1}). \tag{C.27}$$

Take $r = 1 + \eta/2n$, we have $r^{n-1} = (1 + \eta/2n)^{n-1} \leq \exp(\eta/2) \leq 1 + \eta$. Therefore, for all $i, j$, we have $\frac{p_j/p_i}{q_j/q_i} \in [1, 1 + \eta)$, which further implies that $\angle((p_i, p_j), (q_i, q_j)) \leq \eta$ by Lemma F.4. Thus we have for all $|\Omega| \leq 2$ that $\angle(\boldsymbol{p}_\Omega, \boldsymbol{q}_\Omega) \leq \eta$. (The size-1 angles are all zero as we have sign match.)

For this choice of $r$, we have $k = \log R / \log r$ and thus

$$|\mathcal{Q}_{r,k}| = k^{n-1} = \left( \frac{\log R}{\log r} \right)^{n-1} = \left( \frac{\log(1/\eta)}{\log(1 + \eta/(2n))} \right)^{n-1} \leq \left( \frac{4n \log(1/\eta)}{\eta} \right)^{n-1} \doteq \widetilde{N}_n, \tag{C.28}$$

and we have $N(A_{n,R}) \leq \widetilde{N}_n$.

**Step 3** We now construct the covering of $A_n \setminus A_{n,R}$. For any $\boldsymbol{p} \in A_n \setminus A_{n,R}$, there exists some $i$ such that $p_{i+1}/p_i \in [R, \infty)$, which means that the angle of the ray $(p_i, p_{i+1})$ is in between $[\arctan(R), \pi/2) = [\pi/2 - \eta, \pi/2)$. As $\boldsymbol{p}$ is sorted, we have that

$$\frac{p_{i+j}}{p_{i-l}} \geq R \text{ for all } j \geq 1, \, l \geq 0, \tag{C.29}$$

So if we take $\boldsymbol{q}$ such that $q_{i+1}/q_i \in [R, \infty)$, $\boldsymbol{q}$ also has the above property, which gives that

$$\angle((p_{i-l}, p_{i+j}), (q_{i-l}, q_{i+j})) \leq \pi/2 - (\pi/2 - \eta) = \eta \text{ for all } j \geq 1, \, l \geq 0. \tag{C.30}$$

Therefore to obtain the cover in $\mathrm{d}_2$, we only need to consider the angles for $\Omega \subset \{1, \ldots, i\}$ and $\Omega \subset \{i + 1, \ldots, n\}$, which can be done by taking the product of the covers in $A_i$ and $A_{n-i}$.

By considering all $i \in \{1, \ldots, n-1\}$, we obtain the bound

$$N(A_n \setminus A_{n,R}) \leq \sum_{i=1}^{n-1} N(A_i) N(A_{n-i}). \tag{C.31}$$

**Step 4** Putting together Step 2 and Step 3 and using the inductive assumption, we get that

$$N(A_n) \leq N(A_{n,R}) + N(A_n \setminus A_{n,R}) \leq \widetilde{N}_n + \sum_{i=1}^{n-1} N(A_i)N(A_{n-i}) \tag{C.32}$$

$$\leq \left( \frac{4n \log(1/\eta)}{\eta} \right)^{n-1} + \sum_{i=1}^{n-1} (C_\eta i)^{i-1} (C_\eta (n-i))^{n-i-1} \tag{C.33}$$

$$\leq \left( \frac{4}{5} \right)^{n-1} (C_\eta n)^{n-1} + (n-1) \cdot C_\eta^{n-2} n^{n-2} \tag{C.34}$$

$$\leq \left( \left( \frac{4}{5} \right)^{n-1} + \frac{1}{C_\eta} \right) (C_\eta n)^{n-1} \leq (C_\eta n)^{n-1}. \tag{C.35}$$

This shows the case for $m = n$ and completes the induction.

**Step 5** Considering all configurations of {support, sign pattern, ordering}, we have

$$N(\mathbb{S}^{n-1}) \leq 3^n \cdot n! \cdot N(A_n) \leq (3n)^n \left( \frac{5 \log(1/\eta)}{\eta} n \right)^{n-1} \leq \left( \frac{5 \log(1/\eta)}{\eta} n \right)^{2n-1}. \tag{C.36}$$

$\blacksquare$

**Lemma C.4** (Covering number in the $d_\mathbb{E}$ metric). *Assume $n \geq 3$. There exists a numerical constant $C > 0$ such that for any $\epsilon \in (0,1)$, $\mathbb{S}^{n-1}$ admits an $\epsilon$-net of size $\exp(Cn \log \frac{n}{\epsilon})$ w.r.t. $d_\mathbb{E}$ defined in Eq. (C.1): for any $p \in \mathbb{S}^{n-1}$, there exists a $q$ in the net with $\mathrm{supp}\,(q) = \mathrm{supp}\,(p)$ and $d_\mathbb{E}\,(p,q) \leq \epsilon$. We say such $\epsilon$ nets are admissible for $\mathbb{S}^{n-1}$ wrt $d_\mathbb{E}$.*

**Proof.** Let $\eta = \epsilon/n^2$. By Lemma C.3, there exists a subset $\mathcal{Q} \subset \mathbb{S}^{n-1}$ of size at most

$$\left( \frac{5n \log(1/\eta)}{\eta} \right)^{2n-1} = \left( \frac{5n^3 \log(n^2/\epsilon)}{\epsilon} \right)^{2n-1} \leq \exp \left( Cn \log \frac{n}{\epsilon} \right) \tag{C.37}$$

such that for any $p \in \mathbb{S}^{n-1}$, there exists $q \in \mathbb{S}^{n-1}$ such that $\mathrm{supp}(p) = \mathrm{supp}(q)$ and $\angle\,(p_\Omega, q_\Omega) \leq \eta$ for all $|\Omega| \leq 2$. In particular, the $|\Omega| = 1$ case says that $\mathrm{sign}(p) = \mathrm{sign}(q)$, which implies that

$$\angle\,(p_\Omega, q_\Omega) \leq \pi/2 \quad \forall \, \Omega \in \{0,1\}^n. \tag{C.38}$$

Thus, applying the vector angle inequality (Lemma C.2), for any $p \in \mathbb{S}^{n-1}$ and the corresponding $q \in \mathcal{Q}$, we have

$$\angle\,(p_\Omega, q_\Omega) \leq \sum_{|\Omega'|=2, \Omega' \subset \Omega} \angle\,(p_{\Omega'}, q_{\Omega'}) \leq 2 \binom{|\Omega|}{2} \eta \leq |\Omega|^2 \eta \quad \forall \, \Omega \text{ with } 3 \leq |\Omega| \leq n. \tag{C.39}$$

Summing up, we get

$$\angle\,(p_\Omega, q_\Omega) \leq \max\left(2, |\Omega|^2\right) \eta \leq n^2 \eta = \epsilon \quad \forall \, \Omega. \tag{C.40}$$

Therefore $d_\mathbb{E}\,(p, q) \leq \epsilon$. $\blacksquare$

Below we establish the "Lipschitz" property in terms of $d_\mathbb{E}$ distance.

**Lemma C.5.** *Fix $\theta \in (0,1)$. For any $\epsilon \in (0,1)$, let $N_\epsilon$ be an admissible $\epsilon$-net for $\mathbb{S}^{n-1}$ wrt $d_\mathbb{E}$. Let $x_1, \ldots, x_m$ be iid copies of $x \sim_{iid} \mathrm{BG}(\theta)$ in $\mathbb{R}^n$. When $m \geq C\epsilon^{-2}n$, the inequality*

$$\sup_{\substack{p \in \mathbb{S}^{n-1}, q \in N_\epsilon \\ \mathrm{supp}(p) = \mathrm{supp}(q), \, d_\mathbb{E}(p,q) \leq \epsilon}} R\,(p, q) \doteq \frac{1}{m} \sum_{i=1}^m \mathbb{1}\left\{ \mathrm{sign}\,(p^\top x_i) \neq \mathrm{sign}\,(q^\top x_i) \right\} \leq 2\epsilon \tag{C.41}$$

*holds with probability at least $1 - \exp\left(-c\epsilon^2 m\right)$. Here $C$ and $c$ are universal constants independent of $\epsilon$.*

**Proof.** We call any pair of $\boldsymbol{p}, \boldsymbol{q} \in \mathbb{S}^{n-1}$ with $\boldsymbol{q} \in N_\epsilon$, $\mathrm{supp}\,(\boldsymbol{p}) = \mathrm{supp}\,(\boldsymbol{q})$, and $\mathrm{d}_{\mathbb{E}}\,(\boldsymbol{p}, \boldsymbol{q}) \le \epsilon$ an admissible pair. Over any admissible pair $(\boldsymbol{p}, \boldsymbol{q})$, $\mathbb{E}\,[R] = \mathrm{d}_{\mathbb{E}}\,(\boldsymbol{p}, \boldsymbol{q})$. We next bound the deviation $R - \mathbb{E}\,[R]$ uniformly over all admissible $(\boldsymbol{p}, \boldsymbol{q})$ pairs. Observe that the process $R$ is the sample average of $m$ indicator functions. Define the hypothesis class

$$\mathcal{H} = \left\{ \boldsymbol{x} \mapsto \mathbb{1}\left\{ \mathrm{sign}\,\left(\boldsymbol{p}^\top \boldsymbol{x}\right) \ne \mathrm{sign}\,\left(\boldsymbol{q}^\top \boldsymbol{x}\right) \right\} : (\boldsymbol{p}, \boldsymbol{q}) \text{ is an admissible pair} \right\}. \tag{C.42}$$

and let $d_{\mathrm{vc}}(\mathcal{H})$ be the VC-dimension of $\mathcal{H}$. From concentration results for VC-classes (see, e.g., Eq (3) and Theorem 3.4 of Boucheron et al. (2005)), we have

$$\mathbb{P}\left[ \sup_{(\boldsymbol{p}, \boldsymbol{q}) \text{ admissible}} \left\{ R(\boldsymbol{p}, \boldsymbol{q}) - \mathbb{E}\,[R]\,(\boldsymbol{p}, \boldsymbol{q}) \right\} \ge C_0 \sqrt{\frac{d_{\mathrm{vc}}(\mathcal{H})}{m}} + t \right] \le \exp(-mt^2) \tag{C.43}$$

for any $t > 0$. It remains to bound the VC-dimension $d_{\mathrm{vc}}(\mathcal{H})$. First, we have

$$d_{\mathrm{vc}}\,(\mathcal{H}) \le d_{\mathrm{vc}}\left\{ \boldsymbol{x} \mapsto \mathbb{1}\left\{ \mathrm{sign}\,\left(\boldsymbol{p}^\top \boldsymbol{x}\right) \ne \mathrm{sign}\,\left(\boldsymbol{q}^\top \boldsymbol{x}\right) \right\} : \boldsymbol{p}, \boldsymbol{q} \in \mathbb{S}^{n-1} \right\}. \tag{C.44}$$

Observe that each set in the latter hypothesis class can be written as

$$\left\{ \boldsymbol{x} \mapsto \mathbb{1}\left\{ \mathrm{sign}\,\left(\boldsymbol{p}^\top \boldsymbol{x}\right) \ne \mathrm{sign}\,\left(\boldsymbol{q}^\top \boldsymbol{x}\right) \right\} : \boldsymbol{p}, \boldsymbol{q} \in \mathbb{S}^{n-1} \right\}$$
$$= \left\{ \boldsymbol{x} \mapsto \mathbb{1}\left\{ \boldsymbol{p}^\top \boldsymbol{x} > 0, \boldsymbol{q}^\top \boldsymbol{x} \le 0 \right\} : \boldsymbol{p}, \boldsymbol{q} \in \mathbb{S}^{n-1} \right\} \cup \left\{ \boldsymbol{x} \mapsto \mathbb{1}\left\{ \boldsymbol{p}^\top \boldsymbol{x} \ge 0, \boldsymbol{q}^\top \boldsymbol{x} < 0 \right\} : \boldsymbol{p}, \boldsymbol{q} \in \mathbb{S}^{n-1} \right\} \tag{C.45}$$
$$\cup \left\{ \boldsymbol{x} \mapsto \mathbb{1}\left\{ \boldsymbol{p}^\top \boldsymbol{x} < 0, \boldsymbol{q}^\top \boldsymbol{x} \ge 0 \right\} : \boldsymbol{p}, \boldsymbol{q} \in \mathbb{S}^{n-1} \right\} \cup \left\{ \boldsymbol{x} \mapsto \mathbb{1}\left\{ \boldsymbol{p}^\top \boldsymbol{x} \le 0, \boldsymbol{q}^\top \boldsymbol{x} > 0 \right\} : \boldsymbol{p}, \boldsymbol{q} \in \mathbb{S}^{n-1} \right\}. \tag{C.46}$$

the union of intersections of two halfspaces. Thus, letting

$$\mathcal{H}_0 = \left\{ \boldsymbol{x} \mapsto \mathbb{1}\left\{ \boldsymbol{x}^\top \boldsymbol{z} \ge 0 \right\} : \boldsymbol{z} \in \mathbb{R}^n \right\} \tag{C.47}$$

be the class of halfspaces, we have

$$\mathcal{H} \subset (\mathcal{H}_0 \sqcap \mathcal{H}_0) \sqcup (\mathcal{H}_0 \sqcap \mathcal{H}_0) \sqcup (\mathcal{H}_0 \sqcap \mathcal{H}_0) \sqcup (\mathcal{H}_0 \sqcap \mathcal{H}_0). \tag{C.48}$$

Note that $\mathcal{H}_0$ has VC-dimension $n + 1$. Applying bounds on the VC-dimension of unions and intersections (Theorem 1.1, Van Der Vaart & Wellner (2009)), we get that

$$d_{\mathrm{vc}}(\mathcal{H}) \le C_1 d_{\mathrm{vc}}(\mathcal{H}_0 \sqcap \mathcal{H}_0) \le C_2 d_{\mathrm{vc}}(\mathcal{H}_0) \le C_3 n. \tag{C.49}$$

Plugging this bound into Eq. (C.43), we can set $t = \epsilon/2$ and make $m$ large enough so that $C_0\sqrt{C_3}\sqrt{n/m} \le \epsilon/2$, completing the proof. ∎

## C.2  POINTWISE CONVERGENCE OF SUB-DIFFERENTIAL

**Proposition C.6** (Pointwise convergence). *For any fixed $\boldsymbol{q} \in \mathbb{S}^{n-1}$,*

$$\mathbb{P}\left[ \mathrm{d}_{\mathrm{H}}\,(\partial f\,(\boldsymbol{q})\,, \mathbb{E}\,[\partial f]\,(\boldsymbol{q})) > C_a\sqrt{n/m} + C_b t/\sqrt{m} \right] \le 2\exp\left(-t^2\right) \quad \forall\, t > 0. \tag{C.50}$$

*Here $C_a, C_b \ge 0$ are universal constants.*

**Proof.** Recall that

$$\mathrm{d}_{\mathrm{H}}\,(\partial f\,(\boldsymbol{q})\,, \mathbb{E}\,[\partial f\,(\boldsymbol{q})]) = \sup_{\boldsymbol{u} \in \mathbb{S}^{n-1}} \left| h_{\partial f(\boldsymbol{q})}\,(\boldsymbol{u}) - h_{\mathbb{E}\partial f(\boldsymbol{q})}\,(\boldsymbol{u}) \right| = \sup_{\boldsymbol{u} \in \mathbb{S}^{n-1}} \left| h_{\partial f(\boldsymbol{q})}\,(\boldsymbol{u}) - \mathbb{E} h_{\partial f(\boldsymbol{q})}\,(\boldsymbol{u}) \right|. \tag{C.51}$$

Write $X_{\boldsymbol{u}} \doteq h_{\partial f(\boldsymbol{q})}\,(\boldsymbol{u}) - \mathbb{E} h_{\partial f(\boldsymbol{q})}\,(\boldsymbol{u})$ and consider the zero-mean random process $\{X_{\boldsymbol{u}}\}$ defined on $\mathbb{S}^{n-1}$. For any $\boldsymbol{u}, \boldsymbol{v} \in \mathbb{S}^{n-1}$, we have

$$\|X_{\boldsymbol{u}} - X_{\boldsymbol{v}}\|_{\psi_2} = \left\| h_{\partial f(\boldsymbol{q})}\,(\boldsymbol{u}) - \mathbb{E} h_{\partial f(\boldsymbol{q})}\,(\boldsymbol{u}) - h_{\partial f(\boldsymbol{q})}\,(\boldsymbol{v}) + \mathbb{E} h_{\partial f(\boldsymbol{q})}\,(\boldsymbol{v}) \right\|_{\psi_2} \tag{C.52}$$

$$= C_0 \left\| \frac{1}{m} \sum_{i \in [m]} \left( h_{Q_i}\,(\boldsymbol{u}) - \mathbb{E} h_{Q_i}\,(\boldsymbol{u}) - h_{Q_i}\,(\boldsymbol{v}) + \mathbb{E} h_{Q_i}\,(\boldsymbol{v}) \right) \right\|_{\psi_2} \tag{C.53}$$

$$\leq C_1 \frac{1}{m} \left( \sum_{i \in [m]} \| h_{Q_i}(\boldsymbol{u}) - \mathbb{E} h_{Q_i}(\boldsymbol{u}) - h_{Q_i}(\boldsymbol{v}) + \mathbb{E} h_{Q_i}(\boldsymbol{v}) \|_{\psi_2}^2 \right)^{1/2} \quad \text{(C.54)}$$

$$\leq C_2 \frac{1}{m} \left( \sum_{i \in [m]} \| h_{Q_i}(\boldsymbol{u}) - h_{Q_i}(\boldsymbol{v}) \|_{\psi_2}^2 \right)^{1/2} \quad \text{(centering)}, \quad \text{(C.55)}$$

where we write $Q_i \doteq \mathrm{sign}\left(\boldsymbol{q}^\top \boldsymbol{x}_i\right) \boldsymbol{x}_i$ for all $i \in [m]$. Next we estimate $\| h_{Q_i}(\boldsymbol{u}) - h_{Q_i}(\boldsymbol{v}) \|_{\psi_2}$. By definition,

$$h_{Q_i}(\boldsymbol{u}) - h_{Q_i}(\boldsymbol{v}) = \sup_{\boldsymbol{z} \in Q_i} \langle \boldsymbol{z}, \boldsymbol{u} \rangle - \sup_{\boldsymbol{z}' \in Q_i} \langle \boldsymbol{z}', \boldsymbol{v} \rangle. \quad \text{(C.56)}$$

If $h_{Q_i}(\boldsymbol{u}) - h_{Q_i}(\boldsymbol{v}) \geq 0$ and let $\boldsymbol{z}_* \doteq \arg\max_{\boldsymbol{z} \in Q_i} \langle \boldsymbol{z}, \boldsymbol{u} \rangle$, we have

$$h_{Q_i}(\boldsymbol{u}) - h_{Q_i}(\boldsymbol{v}) \leq \langle \boldsymbol{z}_*, \boldsymbol{u} \rangle - \langle \boldsymbol{z}_*, \boldsymbol{v} \rangle = \langle \boldsymbol{z}_*, \boldsymbol{u} - \boldsymbol{v} \rangle, \quad \text{(C.57)}$$

and

$$\| h_{Q_i}(\boldsymbol{u}) - h_{Q_i}(\boldsymbol{v}) \|_{\psi_2} \leq \| \langle \boldsymbol{z}_*, \boldsymbol{u} - \boldsymbol{v} \rangle \|_{\psi_2} \leq \| \boldsymbol{x}_i^\top (\boldsymbol{u} - \boldsymbol{v}) \|_{\psi_2} \leq C_3 \| \boldsymbol{u} - \boldsymbol{v} \|, \quad \text{(C.58)}$$

where we have used Lemma F.1 to obtain the last upper bound. If $h_{Q_i}(\boldsymbol{u}) - h_{Q_i}(\boldsymbol{v}) \leq 0$, $h_{Q_i}(\boldsymbol{v}) - h_{Q_i}(\boldsymbol{u}) \geq 0$ and we can use similar argument to conclude that

$$\| h_{Q_i}(\boldsymbol{u}) - h_{Q_i}(\boldsymbol{v}) \|_{\psi_2} \leq C_3 \| \boldsymbol{u} - \boldsymbol{v} \|. \quad \text{(C.59)}$$

So

$$\| X_{\boldsymbol{u}} - X_{\boldsymbol{v}} \|_{\psi_2} \leq \frac{C_4}{\sqrt{m}} \| \boldsymbol{u} - \boldsymbol{v} \|. \quad \text{(C.60)}$$

Thus, $\{X_{\boldsymbol{u}}\}$ is a centered random process with sub-Gaussian increments with a parameter $C_4/\sqrt{m}$. We can apply Proposition A.2 to conclude that

$$\mathbb{P} \left[ \sup_{\boldsymbol{u} \in \mathbb{S}^{n-1}} \left| h_{\partial f(\boldsymbol{q})}(\boldsymbol{u}) - \mathbb{E} h_{\partial f(\boldsymbol{q})}(\boldsymbol{u}) \right| > C_5 \sqrt{n/m} + C_6 t/\sqrt{m} \right] \leq 2 \exp\left(-t^2\right) \quad \forall\, t > 0, \quad \text{(C.61)}$$

which implies the claimed result. ∎

## C.3 PROOF OF PROPOSITION 3.5 (UNIFORM CONVERGENCE)

Throughout the proof, we let $c, C$ denote universal constants that could change from step to step.

Fix an $\epsilon \in (0, 1/2)$ to be decided later. Let $N_\epsilon$ be an admissible $\epsilon$ net for $\mathbb{S}^{n-1}$ wrt $\mathrm{d}_{\mathbb{E}}$, with $|N_\epsilon| \leq \exp(Cn \log(n/\epsilon))$ (Lemma C.4). By Proposition C.6 and the union bound,

$$\mathbb{P} \left[ \exists\, \boldsymbol{q} \in N_\epsilon, \mathrm{d}_{\mathrm{H}}\left(\partial f(\boldsymbol{q}), \mathbb{E}[\partial f](\boldsymbol{q})\right) > t/3 \right] \leq \exp\left(-cmt^2 + Cn \log \frac{n}{\epsilon}\right) \quad \text{(C.62)}$$

provided that $m \geq Ct^{-2}n$.

For any $\boldsymbol{p} \in \mathbb{S}^{n-1}$, let $\boldsymbol{q} \in N_\epsilon$ satisfy $\mathrm{supp}(\boldsymbol{q}) = \mathrm{supp}(\boldsymbol{p})$ and $\mathrm{d}_{\mathbb{E}}(\boldsymbol{p}, \boldsymbol{q}) \leq \epsilon$. Then we have

$$\mathrm{d}_{\mathrm{H}}\left(\partial f(\boldsymbol{p}), \mathbb{E}[\partial f](\boldsymbol{p})\right) \leq \underbrace{\mathrm{d}_{\mathrm{H}}\left(\partial f(\boldsymbol{q}), \mathbb{E}[\partial f](\boldsymbol{q})\right)}_{\mathrm{I}} + \underbrace{\mathrm{d}_{\mathrm{H}}\left(\mathbb{E}[\partial f](\boldsymbol{p}), \mathbb{E}[\partial f](\boldsymbol{q})\right)}_{\mathrm{II}} + \underbrace{\mathrm{d}_{\mathrm{H}}\left(\partial f(\boldsymbol{p}), \partial f(\boldsymbol{q})\right)}_{\mathrm{III}}$$
$$\text{(C.63)}$$

by the triangular inequality for the Hausdorff metric. By the preceding union bound, term I is bounded by $t/3$ as long as the bad event does not happen. For term II, we have

$$\mathrm{d}_{\mathrm{H}}\left(\mathbb{E}[\partial f](\boldsymbol{p}), \mathbb{E}[\partial f](\boldsymbol{q})\right)$$
$$= \sup_{\boldsymbol{u} \in \mathbb{S}^{n-1}} \left| h_{\mathbb{E}[\partial f](\boldsymbol{p})}(\boldsymbol{u}) - h_{\mathbb{E}[\partial f](\boldsymbol{q})}(\boldsymbol{u}) \right| \quad \text{(C.64)}$$

$$= \sup_{\boldsymbol{u} \in \mathbb{S}^{n-1}} \left| \mathbb{E} \left[ h_{\partial f(\boldsymbol{p})} (\boldsymbol{u}) - h_{\partial f(\boldsymbol{q})} (\boldsymbol{u}) \right] \right| \tag{C.65}$$

$$= \sqrt{\frac{\pi}{2}} \cdot \sup_{\boldsymbol{u} \in \mathbb{S}^{n-1}} \left| \mathbb{E} \left[ \sup \ \mathrm{sign} \left( \boldsymbol{p}^\top \boldsymbol{x}_i \right) \boldsymbol{x}_i^\top \boldsymbol{u} - \sup \ \mathrm{sign} \left( \boldsymbol{q}^\top \boldsymbol{x}_i \right) \boldsymbol{x}_i^\top \boldsymbol{u} \right] \right| \tag{C.66}$$

$$\leq \sqrt{\frac{\pi}{2}} \cdot \sup_{\boldsymbol{u} \in \mathbb{S}^{n-1}} \left| \mathbb{E} \left[ |\boldsymbol{x}_i^\top \boldsymbol{u}| \mathbb{1} \left\{ \mathrm{sign} \left( \boldsymbol{p}^\top \boldsymbol{x}_i \right) \neq \mathrm{sign} \left( \boldsymbol{q}^\top \boldsymbol{x}_i \right) \right\} \right] \right| \tag{C.67}$$

$$\leq \sqrt{\frac{\pi}{2}} \cdot 3\epsilon \sqrt{\log \frac{1}{\epsilon}}. \tag{C.68}$$

where the last line follows from Lemma F.5. As long as $\epsilon \leq ct/\sqrt{\log(1/t)}$, the above term is upper bounded by $t/3$. For term III, we have

$$\mathrm{d}_{\mathrm{H}} \left( \partial f \left( \boldsymbol{p} \right), \partial f \left( \boldsymbol{q} \right) \right)$$

$$= \sup_{\boldsymbol{u} \in \mathbb{S}^{n-1}} \left| h_{\partial f(\boldsymbol{p})} (\boldsymbol{u}) - h_{\partial f(\boldsymbol{q})} (\boldsymbol{u}) \right| \tag{C.69}$$

$$= \sqrt{\frac{\pi}{2}} \cdot \frac{1}{m} \sup_{\boldsymbol{u} \in \mathbb{S}^{n-1}} \left| \sum_{i \in [m]: \mathrm{sign}(\boldsymbol{p}^\top \boldsymbol{x}_i) \neq \mathrm{sign}(\boldsymbol{q}^\top \boldsymbol{x}_i)} \sup \ \mathrm{sign} \left( \boldsymbol{p}^\top \boldsymbol{x}_i \right) \boldsymbol{x}_i^\top \boldsymbol{u} - \sup \ \mathrm{sign} \left( \boldsymbol{q}^\top \boldsymbol{x}_i \right) \boldsymbol{x}_i^\top \boldsymbol{u} \right| \tag{C.70}$$

$$= \sqrt{\frac{\pi}{2}} \cdot \frac{2}{m} \sup_{\boldsymbol{u} \in \mathbb{S}^{n-1}} \left| \sum_{i \in [m]: \mathrm{sign}(\boldsymbol{p}^\top \boldsymbol{x}_i) \neq \mathrm{sign}(\boldsymbol{q}^\top \boldsymbol{x}_i)} s_i \boldsymbol{x}_i^\top \boldsymbol{u} \right| \quad (s_i \in \{+1, -1\} \text{ dependent on } \boldsymbol{p}, \boldsymbol{q}, \boldsymbol{x}_i \text{ and } \boldsymbol{u}) \tag{C.71}$$

$$= \sqrt{\frac{\pi}{2}} \cdot \frac{2}{m} \left\| \sum_{i \in [m]: \mathrm{sign}(\boldsymbol{p}^\top \boldsymbol{x}_i) \neq \mathrm{sign}(\boldsymbol{q}^\top \boldsymbol{x}_i)} s_i \boldsymbol{x}_i \right\|. \tag{C.72}$$

By Lemma C.5, with probability at least $1 - \exp(-c\epsilon^2 m)$, the number of different signs is upper bounded by $2m\epsilon$ for all $\boldsymbol{p}, \boldsymbol{q}$ such that $\mathrm{d}_{\mathbb{E}}(\boldsymbol{p}, \boldsymbol{q}) \leq \epsilon$. On this good event, the above quantity can be upper bounded as follows. Define a set $T \doteq \left\{ \boldsymbol{s} \in \mathbb{R}^m : s_i \in \{+1, -1, 0\}, \|\boldsymbol{s}\|_0 \leq 2m\epsilon \right\}$ and consider the quantity $\sup_{\boldsymbol{s} \in T} \|\boldsymbol{X}\boldsymbol{s}\|$, where $\boldsymbol{X} = [\boldsymbol{x}_1, \ldots, \boldsymbol{x}_m]$. Then,

$$\left\| \sum_{i \in [m]: \mathrm{sign}(\boldsymbol{p}^\top \boldsymbol{x}_i) \neq \mathrm{sign}(\boldsymbol{q}^\top \boldsymbol{x}_i)} s_i \boldsymbol{x}_i \right\| \leq \sup_{\boldsymbol{s} \in T} \|\boldsymbol{X}\boldsymbol{s}\| \tag{C.73}$$

uniformly (i.e., indepdent of $\boldsymbol{p}, \boldsymbol{q}$ and $\boldsymbol{u}$). We have

$$w \left( T \right) = \mathbb{E} \sup_{\boldsymbol{s} \in T} \boldsymbol{s}^\top \boldsymbol{g} = \mathbb{E} \sup_{K \subset [m], |K| \leq 2m\epsilon} \sum_{i \in K} |g_i| \tag{C.74}$$

$$\leq 2m\epsilon \mathbb{E} \|\boldsymbol{g}\|_\infty \leq 4m\sqrt{\log m}\epsilon, \quad (\text{here } \boldsymbol{g} \sim \mathcal{N} \left( \boldsymbol{0}, \boldsymbol{I}_m \right)) \tag{C.75}$$

$$\mathrm{rad} \left( T \right) = \sqrt{2m\epsilon}. \tag{C.76}$$

Noting that $1/\sqrt{\theta} \cdot \boldsymbol{X}$ has independent, isotropic, and sub-Gaussian rows with a parameter $C/\sqrt{\theta}$, we apply Proposition A.3 and obtain that

$$\sup_{\boldsymbol{s} \in T} \|\boldsymbol{X}\boldsymbol{s}\| \leq \sqrt{\theta n}\sqrt{2m\epsilon} + \frac{C}{\sqrt{\theta}} \left( 4m\sqrt{\log m}\,\epsilon + t_0 \sqrt{2m\epsilon} \right) \tag{C.77}$$

with probability at least $1 - 2\exp\left( -t_0^2 \right)$. So we have over all admissible $(\boldsymbol{p}, \boldsymbol{q})$ pairs,

$$\mathrm{d}_{\mathrm{H}} \left( \partial f \left( \boldsymbol{p} \right), \partial f \left( \boldsymbol{q} \right) \right) \leq \sqrt{\frac{\pi}{2}} \cdot \frac{2}{m} \left[ \sqrt{\theta n}\sqrt{2m\epsilon} + \frac{C}{\sqrt{\theta}} \left( 4m\sqrt{\log m}\,\epsilon + t_0 \sqrt{2m\epsilon} \right) \right] \tag{C.78}$$

$$= \sqrt{\frac{\pi}{2}} \cdot \left( \sqrt{\frac{8\theta n\epsilon}{m}} + C\epsilon \sqrt{\frac{\log m}{\theta}} + Ct_0 \sqrt{\frac{8\epsilon}{m}} \right). \tag{C.79}$$

Setting $t_0 = ct\sqrt{m}$ and $\epsilon = ct\sqrt{\theta/\log m}$, we have that

$$\mathrm{d}_{\mathrm{H}}\left(\partial f\left(\boldsymbol{p}\right), \partial f\left(\boldsymbol{q}\right)\right) \leq \frac{t}{3}, \tag{C.80}$$

provided that $m \geq C\epsilon t^{-2}n = Ct^{-1}n\sqrt{\theta/\log m}$, which is subsumed by the earlier requirement $m \geq Ct^{-2}n$.

Putting together the three bounds Eq. (C.62), Eq. (C.67), Eq. (C.80), we can choose

$$\epsilon = ct\sqrt{\frac{\theta}{\log(m/t)}} \leq ct \cdot \min\left\{\sqrt{\frac{\theta}{\log m}}, \frac{1}{\sqrt{\log(1/t)}}\right\} \tag{C.81}$$

and get that $\mathrm{d}_{\mathrm{H}}\left(\partial f(\boldsymbol{p}), \mathbb{E}\left[\partial f\right](\boldsymbol{p})\right) \leq t$ with probability at least

$$1 - 2\exp\left(-cmt^2\right) - \exp(-cm\epsilon^2) - \exp\left(-cmt^2 + Cn\log\frac{n}{\epsilon}\right) \tag{C.82}$$

$$\geq 1 - 2\exp(-cmt^2) - \exp\left(-\frac{cm\theta t^2}{\log(m/t)}\right) - \exp\left(-cmt^2 + Cn\log\frac{n\log(m/t)}{\theta t}\right) \tag{C.83}$$

$$\geq 1 - \exp\left(-\frac{cm\theta t^2}{\log(m/t)}\right) \tag{C.84}$$

provided that $m \geq Cnt^{-2}\log\frac{n\log(m/t)}{\theta t}$. A sufficient condition is that $m \geq Cnt^{-2}\log^2(n/t)$ for sufficiently large $C$. When this is satisfied, the probability is further lower bounded by $1 - \exp(-cm\theta t^2/\log m)$.

## C.4 Proof of Theorem 3.6

Define

$$t = \frac{1}{32n^{3/2}}\theta(1-\theta)\zeta_0 \leq \min\left\{\frac{1}{8n^{3/2}}\theta(1-\theta)\frac{\zeta_0}{1+\zeta_0}, \frac{2-\sqrt{2}}{16n^{3/2}}\theta(1-\theta)\zeta_0\right\}. \tag{C.85}$$

By Proposition 3.5, with probability at least $1 - \exp\left(-cm\theta^3\zeta_0^2 n^{-3}\log^{-1}m\right)$ we have

$$\mathrm{d}_{\mathrm{H}}\left(\mathbb{E}\left[\partial f\right](\boldsymbol{q}), \partial f\left(\boldsymbol{q}\right)\right) \leq t, \tag{C.86}$$

provided that $m \geq Cn^4\theta^{-2}\zeta_0^{-2}\log(n/\zeta_0)$. We now show the properties Eq. (3.12) and Eq. (3.13) on this good event, focusing on $\mathcal{S}_{\zeta_0}^{(n+)}$ but obtaining the same results for all other $2n-1$ subsets by the same arguments.

For Eq. (3.12), we have

$$\langle\partial_R f\left(\boldsymbol{q}\right), \boldsymbol{e}_j/q_j - \boldsymbol{e}_n/q_n\rangle = \langle\partial f\left(\boldsymbol{q}\right), \left(\boldsymbol{I} - \boldsymbol{q}\boldsymbol{q}^\top\right)\left(\boldsymbol{e}_j/q_j - \boldsymbol{e}_n/q_n\right)\rangle = \langle\partial f\left(\boldsymbol{q}\right), \boldsymbol{e}_j/q_j - \boldsymbol{e}_n/q_n\rangle. \tag{C.87}$$

Now

$$\sup\langle\partial f\left(\boldsymbol{q}\right), \boldsymbol{e}_n/q_n - \boldsymbol{e}_j/q_j\rangle$$
$$= h_{\partial f(\boldsymbol{q})}\left(\boldsymbol{e}_n/q_n - \boldsymbol{e}_j/q_j\right) \tag{C.88}$$
$$= \mathbb{E}h_{\partial f(\boldsymbol{q})}\left(\boldsymbol{e}_n/q_n - \boldsymbol{e}_j/q_j\right) - \mathbb{E}h_{\partial f(\boldsymbol{q})}\left(\boldsymbol{e}_n/q_n - \boldsymbol{e}_j/q_j\right) + h_{\partial f(\boldsymbol{q})}\left(\boldsymbol{e}_n/q_n - \boldsymbol{e}_j/q_j\right) \tag{C.89}$$
$$\leq \mathbb{E}h_{\partial f(\boldsymbol{q})}\left(\boldsymbol{e}_n/q_n - \boldsymbol{e}_j/q_j\right) + \|\boldsymbol{e}_n/q_n - \boldsymbol{e}_j/q_j\|\sup_{\boldsymbol{u}\in\mathbb{S}^{n-1}}\left|\mathbb{E}h_{\partial f(\boldsymbol{q})}\left(\boldsymbol{u}\right) - h_{\partial f(\boldsymbol{q})}\left(\boldsymbol{u}\right)\right| \tag{C.90}$$
$$= \sup\langle\mathbb{E}\left[\partial f\right]\left(\boldsymbol{q}\right), \boldsymbol{e}_n/q_n - \boldsymbol{e}_j/q_j\rangle + \|\boldsymbol{e}_n/q_n - \boldsymbol{e}_j/q_j\|\,\mathrm{d}_{\mathrm{H}}\left(\mathbb{E}\left[\partial f\right]\left(\boldsymbol{q}\right), \partial f\left(\boldsymbol{q}\right)\right). \tag{C.91}$$

By Theorem 3.4(a),

$$\sup\langle\mathbb{E}\left[\partial f\right]\left(\boldsymbol{q}\right), \boldsymbol{e}_n - q_n\boldsymbol{q}\rangle \leq -\frac{1}{2n}\theta\left(1-\theta\right)\frac{\zeta_0}{1+\zeta_0}. \tag{C.92}$$

Moreover, $\|\boldsymbol{e}_n/q_n - \boldsymbol{e}_j/q_j\| = \sqrt{1/q_n^2 + 1/q_j^2} \le \sqrt{1/q_n^2 + 3/q_n^2} \le 2\sqrt{n}$. Meanwhile, we have

$$\mathrm{d}_{\mathrm{H}}\left(\mathbb{E}\left[\partial f\right]\left(\boldsymbol{q}\right), \partial f\left(\boldsymbol{q}\right)\right) \le t \le \frac{1}{8n^{3/2}}\theta\left(1-\theta\right)\frac{\zeta_0}{1+\zeta_0}. \tag{C.93}$$

We conclude that

$$\inf\left\langle \partial f\left(\boldsymbol{q}\right), \boldsymbol{e}_j/q_j - \boldsymbol{e}_n/q_n \right\rangle = -\sup\left\langle \partial f\left(\boldsymbol{q}\right), \boldsymbol{e}_n/q_n - \boldsymbol{e}_j/q_j \right\rangle \tag{C.94}$$

$$\ge \frac{1}{2n}\theta\left(1-\theta\right)\frac{\zeta_0}{1+\zeta_0} - 2\sqrt{n}\cdot\frac{1}{4n}\theta\left(1-\theta\right)\frac{\zeta_0}{1+\zeta_0}\cdot\frac{1}{2\sqrt{n}} \tag{C.95}$$

$$\ge \frac{1}{4n}\theta\left(1-\theta\right)\frac{\zeta_0}{1+\zeta_0}, \tag{C.96}$$

as claimed.

For Eq. (3.13), we have by Theorem 3.4(b) that

$$\sup\left\langle \partial f\left(\boldsymbol{q}\right), \boldsymbol{e}_n - q_n\boldsymbol{q} \right\rangle = h_{\partial f(\boldsymbol{q})}\left(\boldsymbol{e}_n - q_n\boldsymbol{q}\right) \tag{C.97}$$

$$= \mathbb{E}h_{\partial f(\boldsymbol{q})}\left(\boldsymbol{e}_n - q_n\boldsymbol{q}\right) - \mathbb{E}h_{\partial f(\boldsymbol{q})}\left(\boldsymbol{e}_n - q_n\boldsymbol{q}\right) + h_{\partial f(\boldsymbol{q})}\left(\boldsymbol{e}_n - q_n\boldsymbol{q}\right) \tag{C.98}$$

$$\le \mathbb{E}h_{\partial f(\boldsymbol{q})}\left(\boldsymbol{e}_n - q_n\boldsymbol{q}\right) + \|\boldsymbol{e}_n - q_n\boldsymbol{q}\|\sup_{\boldsymbol{u}\in\mathbb{S}^{n-1}}\left|\mathbb{E}h_{\partial f(\boldsymbol{q})}\left(\boldsymbol{u}\right) - h_{\partial f(\boldsymbol{q})}\left(\boldsymbol{u}\right)\right| \tag{C.99}$$

$$= \sup\left\langle \mathbb{E}\left[\partial f\right]\left(\boldsymbol{q}\right), \boldsymbol{e}_n - q_n\boldsymbol{q} \right\rangle + \|\boldsymbol{q}_{-n}\|\,\mathrm{d}_{\mathrm{H}}\left(\mathbb{E}\left[\partial f\right]\left(\boldsymbol{q}\right), \partial f\left(\boldsymbol{q}\right)\right). \tag{C.100}$$

As we are on the good event

$$\mathrm{d}_{\mathrm{H}}\left(\mathbb{E}\left[\partial f\right]\left(\boldsymbol{q}\right), \partial f\left(\boldsymbol{q}\right)\right) \le t \le \frac{2-\sqrt{2}}{16n^{3/2}}\cdot\theta\left(1-\theta\right)\zeta_0, \tag{C.101}$$

we have

$$\inf\left\langle \partial f\left(\boldsymbol{q}\right), q_n\boldsymbol{q} - \boldsymbol{e}_n \right\rangle = -\sup\left\langle \partial f\left(\boldsymbol{q}\right), \boldsymbol{e}_n - q_n\boldsymbol{q} \right\rangle \tag{C.102}$$

$$\ge \frac{1}{8}\theta\left(1-\theta\right)\zeta_0 n^{-3/2}\|\boldsymbol{w}\| - \|\boldsymbol{q}_{-n}\|\frac{2-\sqrt{2}}{16}\cdot\theta\left(1-\theta\right)\zeta_0 n^{-3/2} \tag{C.103}$$

$$\ge \frac{\sqrt{2}}{16}\theta\left(1-\theta\right)\zeta_0 n^{-3/2}\|\boldsymbol{q}_{-n}\|. \tag{C.104}$$

Noting that $\|\boldsymbol{q}_{-n}\| \ge \frac{1}{\sqrt{2}}\|\boldsymbol{q} - \boldsymbol{e}_n\|$ for all $\boldsymbol{q}$ with $q_n \ge 0$ completes the proof.

## C.5 PROOF OF PROPOSITION 3.7

For any $\boldsymbol{q} \in \mathbb{S}^{n-1}$,

$$\sup\|\partial f\left(\boldsymbol{q}\right)\| = \mathrm{d}_{\mathrm{H}}\left(\{\boldsymbol{0}\}, \partial f\left(\boldsymbol{q}\right)\right) \le \mathrm{d}_{\mathrm{H}}\left(\{\boldsymbol{0}\}, \mathbb{E}\left[\partial f\right]\left(\boldsymbol{q}\right)\right) + \mathrm{d}_{\mathrm{H}}\left(\partial f\left(\boldsymbol{q}\right), \mathbb{E}\left[\partial f\right]\left(\boldsymbol{q}\right)\right) \tag{C.105}$$

by the metric property of the Hausdorff metric. On one hand, we have

$$\sup\|\mathbb{E}\left[\partial f\right]\left(\boldsymbol{q}\right)\| = \sup\left\|\mathbb{E}_\Omega\left[\frac{\boldsymbol{q}_\Omega}{\|\boldsymbol{q}_\Omega\|}\mathbb{1}\left\{\boldsymbol{q}_\Omega \ne \boldsymbol{0}\right\} + \left\{\boldsymbol{v}_\Omega : \|\boldsymbol{v}_\Omega\| \le 1\right\}\mathbb{1}\left\{\boldsymbol{q}_\Omega = \boldsymbol{0}\right\}\right]\right\| \le 1. \tag{C.106}$$

On the other hand, by Proposition 3.5,

$$\mathrm{d}_{\mathrm{H}}\left(\partial f\left(\boldsymbol{q}\right), \mathbb{E}\left[\partial f\right]\left(\boldsymbol{q}\right)\right) \le 1 \quad \forall\, \boldsymbol{q} \in \mathbb{S}^{n-1} \tag{C.107}$$

with probability at least $1 - \exp\left(-c_1 m\theta \log^{-1} m\right)$, provided that $m \ge C_2 n^2 \log n$ (simplified using $\theta \ge 1/n$). Combining the two results complete the proof.

## C.6 ADDITIONAL GEOMETRIES ON THE EMPIRICAL OBJECTIVE

**Proposition C.7.** *On the good event in [Proposition 3.7](#), for all $\boldsymbol{q} \in \mathcal{S}_{\zeta_0}^{(n+)}$, we have*

$$f(\boldsymbol{q}) - f(\boldsymbol{e}_n) \leq 2\sqrt{n} \|\boldsymbol{q} - \boldsymbol{e}_n\|. \tag{C.108}$$

**Proof.** We use Lebourg's mean value theorem for locally Lipschitz functions[5], i.e., Theorem 2.3.7 of ([Clarke, 1990](#)). It is convenient to work in the $\boldsymbol{w}$ space here. By subdifferential chain rules, $g(w)$ is locally Lipschitz over $\left\{\boldsymbol{w} : \|\boldsymbol{w}\| < \sqrt{\frac{n-1}{n}}\right\}$. Thus, we have

$$f(\boldsymbol{q}) - f(\boldsymbol{e}_n) = g(\boldsymbol{w}) - g(\boldsymbol{0}) = \langle \boldsymbol{v}, \boldsymbol{w} \rangle \tag{C.109}$$

for a certain $t_0 \in (0,1)$ and a certain $\boldsymbol{v} \in \partial g(t_0 \boldsymbol{w})$. Now for any $\boldsymbol{q}$ and the corresponding $\boldsymbol{w}$,

$$\langle \partial g(\boldsymbol{w}), \boldsymbol{w} \rangle = \frac{1}{q_n} \langle \partial_R f(\boldsymbol{q}), q_n \boldsymbol{q} - \boldsymbol{e}_n \rangle. \tag{C.110}$$

It follows

$$\langle \boldsymbol{v}, \boldsymbol{w} \rangle \leq \frac{1}{t_0} \sup \langle \partial g(t_0 \boldsymbol{w}), t_0 \boldsymbol{w} \rangle = \frac{1}{t_0} \frac{1}{q_n(t_0 \boldsymbol{w})} \sup \langle \partial_R f(\boldsymbol{q}(t_0 \boldsymbol{w})), q_n(t_0 \boldsymbol{w}) \boldsymbol{q}(t_0 \boldsymbol{w}) - \boldsymbol{e}_n \rangle$$

$$\leq \sup \|\partial_R f(\boldsymbol{q}(t_0 \boldsymbol{w}))\| \cdot \frac{\|q_n(t_0 \boldsymbol{w}) \boldsymbol{q}(t_0 \boldsymbol{w}) - \boldsymbol{e}_n\|}{t_0 q_n(t_0 \boldsymbol{w})} \leq 2 \frac{\|q_n(t_0 \boldsymbol{w}) \boldsymbol{q}(t_0 \boldsymbol{w}) - \boldsymbol{e}_n\|}{t_0 q_n(t_0 \boldsymbol{w})}, \tag{C.111}$$

where at the last inequality we have used [Proposition 3.7](#). Continuing the calculation, we further have

$$\frac{\|q_n(t_0 \boldsymbol{w}) \boldsymbol{q}(t_0 \boldsymbol{w}) - \boldsymbol{e}_n\|}{t_0 q_n(t_0 \boldsymbol{w})} = \frac{t_0 \|\boldsymbol{w}\|}{t_0 q_n(t_0 \boldsymbol{w})} \leq \sqrt{n} \|\boldsymbol{w}\| \leq \sqrt{n} \|\boldsymbol{q} - \boldsymbol{e}_n\|, \tag{C.112}$$

completing the proof. ∎

**Proposition C.8.** *Assume $\theta \in [1/n, 1/2]$. When $m \geq C\theta^{-2} n \log n$, with probability at least $1 - \exp\left(-cm\theta^3 \log^{-1} m\right)$, the following holds: for all $\boldsymbol{q} \in \mathcal{S}_{\zeta_0}^{(n+)}$ satisfying $f(\boldsymbol{q}) - f(\boldsymbol{e}_n) \leq \frac{2}{25}\theta$,*

$$f(\boldsymbol{q}) - f(\boldsymbol{e}_n) \geq \frac{\sqrt{2}}{16} \theta(1-\theta) \|\boldsymbol{q}_{-n}\| \geq \frac{1}{16} \theta(1-\theta) \|\boldsymbol{q} - \boldsymbol{e}_n\|. \tag{C.113}$$

*Here $C, c > 0$ are universal constants.*

**Proof.** We first establish uniform convergence of $f(\boldsymbol{p})$ to $\mathbb{E}[f](\boldsymbol{p})$. Consider the zero-centered random process $X_{\boldsymbol{p}} \doteq f(\boldsymbol{p}) - \mathbb{E}[f](\boldsymbol{p})$ on $\mathbb{S}^{n-1}$. Similar to proof of [Proposition C.6](#), we can show that for all $\boldsymbol{p}, \boldsymbol{q} \in \mathbb{S}^{n-1}$

$$\|X_{\boldsymbol{p}} - X_{\boldsymbol{q}}\|_{\psi_2} \leq \frac{C}{\sqrt{m}} \|\boldsymbol{p} - \boldsymbol{q}\|. \tag{C.114}$$

Applying [Proposition A.2](#) gives that

$$\|f(\boldsymbol{p}) - \mathbb{E}[f](\boldsymbol{q})\| \leq \frac{1}{100} \theta \quad \forall \boldsymbol{q} \in \mathbb{S}^{n-1} \tag{C.115}$$

with probability at least $1 - \exp\left(-cm\theta^2\right)$, provided that $m \geq C\theta^{-2} n$.

Now we consider $\mathbb{E}[f](\boldsymbol{q}) - \mathbb{E}[f](\boldsymbol{e}_n)$. For convenience, we first work in the $\boldsymbol{w}$ space and note that $\mathbb{E}[f](\boldsymbol{q}) - \mathbb{E}[f](\boldsymbol{e}_n) = \mathbb{E}[g](\boldsymbol{w}(\boldsymbol{q})) - \mathbb{E}[g](\boldsymbol{0})$. By [Lemma B.3](#), $\mathbb{E}[g]$ is monotonically increasing in every radial direction $\boldsymbol{v}$ until $\|\boldsymbol{w}\|^2 + \|\boldsymbol{w}\|_\infty^2 \leq 1$, which implies that

$$\inf_{\|\boldsymbol{w}\| \geq 1/2} \mathbb{E}[g](\boldsymbol{w}(\boldsymbol{q})) - \mathbb{E}[g](\boldsymbol{0}) = \inf_{\|\boldsymbol{w}\| = 1/2} \mathbb{E}[g](\boldsymbol{w}(\boldsymbol{q})) - \mathbb{E}[g](\boldsymbol{0}). \tag{C.116}$$

---

[5] It is possible to directly apply the manifold version of Lebourg's mean value theorem, i.e., Theorem 3.3 of [Hosseini & Pouryayevali (2011)](#). We avoid this technicality by working with the Euclidean version in $\boldsymbol{w}$ space.

For $\boldsymbol{w}$ with $\|\boldsymbol{w}\| = 1/2$,

$$\mathbb{E}\left[g\right]\left(\boldsymbol{w}\right) - \mathbb{E}\left[g\right]\left(\boldsymbol{0}\right) = \left(1-\theta\right)\mathbb{E}_{\Omega}\left\|\boldsymbol{w}_{\Omega}\right\| + \theta\mathbb{E}_{\Omega}\sqrt{1-\left\|\boldsymbol{w}_{\Omega^{c}}\right\|^{2}} - \theta \quad \text{(Lemma B.1)} \quad \text{(C.117)}$$

$$\geq \left(1-\theta\right)\theta\left\|\boldsymbol{w}\right\| + \theta\mathbb{E}_{\Omega}\sqrt{1-\left\|\boldsymbol{w}\right\|^{2}} - \theta \quad \text{(C.118)}$$

$$\geq \frac{1}{4}\theta + \frac{\sqrt{3}}{2}\theta - \theta \quad \text{(using } \theta \leq 1/2 \text{ and } \|\boldsymbol{w}\| = 1/2\text{)} \quad \text{(C.119)}$$

$$\geq \frac{1}{10}\theta. \quad \text{(C.120)}$$

So, back to the $\boldsymbol{q}$ space,

$$\inf_{\boldsymbol{q}\in\mathcal{S}_{\zeta_{0}}^{(n+)}:\,\|\boldsymbol{q}_{-n}\|\geq 1/2} \mathbb{E}\left[f\right]\left(\boldsymbol{q}\right) - \mathbb{E}\left[f\right]\left(\boldsymbol{0}\right) \geq \frac{1}{10}\theta. \quad \text{(C.121)}$$

Combining the results in Eq. (C.115) and Eq. (C.121), we conclude that with high probability

$$\inf_{\boldsymbol{q}\in\mathcal{S}_{\zeta_{0}}^{(n+)}:\,\|\boldsymbol{q}_{-n}\|\geq 1/2} f\left(\boldsymbol{q}\right) - f\left(\boldsymbol{0}\right) \geq \frac{2}{25}\theta. \quad \text{(C.122)}$$

So when $f\left(\boldsymbol{q}\right) - f\left(\boldsymbol{0}\right) \leq 2/25 \cdot \theta$, $\|\boldsymbol{q}_{-n}\| \leq 1/2$, which is equivalent to $\|\boldsymbol{w}\| \leq 1/2$ in the $\boldsymbol{w}$ space. Under this constraint, by Lemma B.3,

$$D_{-\boldsymbol{w}/\|\boldsymbol{w}\|}^{c}\mathbb{E}\left[g\right]\left(\boldsymbol{w}\right) \leq -\theta\left(1-\theta\right)\left(1/\sqrt{1+\left\|\boldsymbol{w}\right\|_{\infty}^{2}/\left\|\boldsymbol{w}\right\|^{2}} - \left\|\boldsymbol{w}\right\|\right) \quad \text{(C.123)}$$

$$\leq -\theta\left(1-\theta\right)\left(\frac{1}{\sqrt{2}} - \frac{1}{2}\right) \leq -\frac{1}{5}\theta\left(1-\theta\right). \quad \text{(C.124)}$$

So, emulating the proof of Eq. (3.9) in Theorem 3.4, we have that for $\boldsymbol{q}\in\mathcal{S}_{\zeta_{0}}^{(n+)}$ with $\|\boldsymbol{q}_{-n}\| \leq 1/2$,

$$\left\langle\mathbb{E}\left[\partial_{R}f\right]\left(\boldsymbol{q}\right), q_{n}\boldsymbol{q} - \boldsymbol{e}_{n}\right\rangle = q_{n}\left\langle\mathbb{E}\left[\partial g\right]\left(\boldsymbol{w}\right),\boldsymbol{w}\right\rangle \geq q_{n}\left\|\boldsymbol{w}\right\| \cdot \frac{1}{5}\theta\left(1-\theta\right) \geq \frac{\sqrt{3}}{10}\theta\left(1-\theta\right)\left\|\boldsymbol{w}\right\|, \quad \text{(C.125)}$$

where at the last inequality we use $q_{n} = \sqrt{1-\left\|\boldsymbol{w}\right\|^{2}} \geq \sqrt{3}/2$ when $\|\boldsymbol{w}\| \leq 1/2$. Moreover, we emulate the proof of Eq. (3.13) in Theorem 3.6 to obtain that

$$\inf\left\langle\partial_{R}f\left(\boldsymbol{q}\right), q_{n}\boldsymbol{q} - \boldsymbol{e}_{n}\right\rangle \geq \frac{\sqrt{2}}{16}\theta\left(1-\theta\right)\left\|\boldsymbol{q}_{-n}\right\| \geq \frac{1}{16}\theta\left(1-\theta\right)\left\|\boldsymbol{q} - \boldsymbol{e}_{n}\right\| \quad \text{(C.126)}$$

with probability at least $1 - \exp\left(-cm\theta^{3}\log^{-1}m\right)$, provided that $m \geq C\theta^{-2}n\log n$.

The last step of our proof is invoking the mean value theorem, similar to the proof of Proposition C.7. For any $\boldsymbol{q}$, we have

$$f\left(\boldsymbol{q}\right) - f\left(\boldsymbol{e}_{n}\right) = g\left(\boldsymbol{w}\right) - g\left(\boldsymbol{0}\right) = \left\langle\boldsymbol{v},\boldsymbol{w}\right\rangle \quad \text{(C.127)}$$

for a certain $t \in \left(0,1\right)$ and a certain $\boldsymbol{v}\in\partial g\left(t\boldsymbol{w}\right)$. We have

$$\left\langle\boldsymbol{v},\boldsymbol{w}\right\rangle \geq \frac{1}{t_{0}}\inf\left\langle\partial g\left(t_{0}\boldsymbol{w}\right), t_{0}\boldsymbol{w}\right\rangle = \frac{1}{t_{0}}\frac{1}{q_{n}\left(t_{0}\boldsymbol{w}\right)}\inf\left\langle\partial_{R}f\left(\boldsymbol{q}\left(t_{0}\boldsymbol{w}\right)\right), q_{n}\left(t_{0}\boldsymbol{w}\right)\boldsymbol{q}\left(t_{0}\boldsymbol{w}\right) - \boldsymbol{e}_{n}\right\rangle \quad \text{(C.128)}$$

$$\geq \frac{1}{t_{0}}\frac{1}{q_{n}\left(t_{0}\boldsymbol{w}\right)}\frac{\sqrt{2}}{16}\theta\left(1-\theta\right)\left\|t_{0}\boldsymbol{w}\right\| \quad \text{(C.129)}$$

$$\geq \frac{\sqrt{2}}{16}\theta\left(1-\theta\right)\left\|\boldsymbol{w}\right\| \quad \text{(C.130)}$$

$$\geq \frac{1}{16}\theta\left(1-\theta\right)\left\|\boldsymbol{q} - \boldsymbol{e}_{n}\right\|, \quad \text{(C.131)}$$

completing the proof. ∎

## D    PROOFS FOR SECTION 3.3

### D.1    STAYING IN THE REGION $\mathcal{S}_{\zeta_0}^{(n+)}$

**Lemma D.1** (Progress in $\mathcal{S}_{\zeta_0}^{(n+)} \setminus S_1^{(n+)}$). *Set $\eta = t_0/(100\sqrt{n})$ for $t_0 \in (0,1)$. For any $\zeta_0 \in (0,1)$, on the good events stated in Proposition 3.7 and Theorem 3.6, we have for all $\boldsymbol{q} \in \mathcal{S}_{\zeta_0}^{(n+)} \setminus \mathcal{S}_1^{(n+)}$ and $\boldsymbol{q}_+$ being the next step of Riemannian subgradient descent that*

$$\frac{q_{+,n}^2}{\|\boldsymbol{q}_{+,-n}\|_\infty^2} \geq \frac{q_n^2}{\|\boldsymbol{q}_{-n}\|_\infty^2} \left(1 + t\frac{\theta(1-\theta)\zeta_0}{400n^{3/2}(1+\zeta_0)}\right)^2. \tag{D.1}$$

*In particular, we have $\boldsymbol{q}_+ \in \mathcal{S}_{\zeta_0}^{(n+)}$.*

**Proof.** We divide the index set $[n-1]$ into three sets

$$\mathcal{I}_0 \doteq \{j \in [n-1] : q_j = 0\}, \tag{D.2}$$

$$\mathcal{I}_1 \doteq \{j \in [n-1] : q_n^2/q_j^2 > 1 + 2\zeta_1 = 3, q_j \neq 0\} \tag{D.3}$$

$$\mathcal{I}_2 \doteq \{j \in [n-1] : q_n^2/q_j^2 \leq 1 + 2\zeta_1 = 3\}. \tag{D.4}$$

We perform different arguments on different sets. We let $\boldsymbol{g}(\boldsymbol{q}) \in \partial_R f(\boldsymbol{q})$ be the subgradient taken at $\boldsymbol{q}$ and note by Proposition 3.7 that $\|\boldsymbol{g}\| \leq 2$, and so $|g_i| \leq 2$ for all $i \in [n]$. We have

$$\frac{q_{+,n}^2}{q_{+,j}^2} = \frac{(q_n - \eta g_n)^2/\|\boldsymbol{q} - \eta \boldsymbol{g}\|^2}{(q_j - \eta g_j)^2/\|\boldsymbol{q} - \eta \boldsymbol{g}\|^2} = \frac{(q_n - \eta g_n)^2}{(q_j - \eta g_j)^2}. \tag{D.5}$$

For any $j \in \mathcal{I}_0$,

$$\frac{q_{+,n}^2}{q_{+,j}^2} = \frac{(q_n - \eta g_n)^2}{\eta^2 g_j^2} = q_n^2 \frac{(1 - \eta g_n/q_n)^2}{\eta^2 g_j^2} \geq \frac{(1 - 2\eta\sqrt{n})^2}{4n\eta^2}. \tag{D.6}$$

Provided that $\eta \leq 1/(4\sqrt{n})$, $1 - 2\eta\sqrt{n} \geq 1/2$, and so

$$\frac{(1 - 2\eta\sqrt{n})^2}{4n\eta^2} \geq \frac{1}{16n\eta^2} \geq \frac{5}{2}, \tag{D.7}$$

where the last inequality holds when $\eta \leq 1/\sqrt{40n}$.

For any $j \in \mathcal{I}_1$,

$$\frac{q_{+,n}^2}{q_{+,j}^2} \geq \frac{q_n^2(1 - \eta g_n/q_n)^2}{q_j^2 + \eta^2 g_j^2} \geq \frac{q_n^2(1 - \eta g_n/q_n)^2}{q_n^2/3 + 4\eta^2} = \frac{3(1 - \eta g_n/q_n)^2}{1 + 12\eta^2/q_n^2} \geq \frac{3(1 - 2\eta\sqrt{n})^2}{1 + 12n\eta^2} \geq \frac{5}{2}, \tag{D.8}$$

where the very last inequality holds when $\eta \leq 1/(26\sqrt{n})$.

Since $\boldsymbol{q} \in \mathcal{S}_{\zeta_0}^{(n+)} \setminus \mathcal{S}_1^{(n+)}$, $\mathcal{I}_2$ is nonempty. For any $j \in \mathcal{I}_2$,

$$\frac{q_{+,n}^2}{q_{+,j}^2} = \frac{q_n^2}{q_j^2}\left(1 + \eta\frac{g_j/q_j - g_n/q_n}{1 - \eta g_j/q_j}\right)^2. \tag{D.9}$$

Since $g_j/q_j \leq 2\sqrt{3n}$, $1 - \eta g_j/q_j \geq 1/2$ when $\eta \leq 1/(4\sqrt{3n})$. Conditioned on this and due to that $g_j/q_j - g_n/q_n \geq 0$, it follows

$$\left(1 + \eta\frac{g_j/q_j - g_n/q_n}{1 - \eta g_j/q_j}\right)^2 \leq [1 + 2\eta(g_j/q_j - g_n/q_n)]^2 \leq \left[1 + 2\eta\left(2\sqrt{3n} + 2\sqrt{n}\right)\right]^2 \leq (1 + 11\eta\sqrt{n})^2 \tag{D.10}$$

If $q_n^2/q_j^2 \leq 2$, $q_{+,n}^2/q_{+,j}^2 \leq 5/2$ provided that

$$\left(1 + 11\eta\sqrt{n}\right)^2 \leq \frac{5/2}{2} = \frac{5}{4} \impliedby \eta \leq \frac{1}{100\sqrt{n}}. \tag{D.11}$$

As $q \notin S_1^{(n+)}$, we have $q_n^2/\|q_{-n}\|_\infty^2 \leq 2$, so there must be a certain $j \in \mathcal{I}_2$ satisfying $q_n^2/q_j^2 \leq 2$. We conclude that when

$$\eta \leq \min\left\{\frac{1}{\sqrt{40n}}, \frac{1}{26\sqrt{n}}, \frac{1}{100\sqrt{n}}\right\} = \frac{1}{100\sqrt{n}}, \tag{D.12}$$

the index of largest entries of $q_{+,-n}$ remains in $\mathcal{I}_2$.

On the other hand, when $\eta \leq 1/(100\sqrt{n})$, for all $j \in \mathcal{I}_2$,

$$\left(1 + \eta\frac{g_j/q_j - g_n/q_n}{1 - \eta g_j/q_j}\right)^2 \geq [1 + \eta(g_j/q_j - g_n/q_n)]^2 \geq \left(1 + \frac{\eta}{4n}\theta(1-\theta)\frac{\zeta_0}{1+\zeta_0}\right)^2. \tag{D.13}$$

So when $\eta = t/(100\sqrt{n})$ for any $t \in (0,1)$,

$$\frac{q_{+,n}^2}{\|q_{+,-n}\|_\infty^2} \geq \frac{q_n^2}{\|q_{-n}\|_\infty^2}\left(1 + t\frac{\theta(1-\theta)\zeta_0}{400n^{3/2}(1+\zeta_0)}\right)^2, \tag{D.14}$$

completing the proof. ∎

**Proposition D.2.** *For any $\zeta_0 \in (0,1)$, on the good events stated in [Proposition 3.7] and [Theorem 3.6], if the step sizes satisfy*

$$\eta^{(k)} \leq \min\left\{\frac{1}{100\sqrt{n}}, \frac{1-\zeta_0}{9\sqrt{n}}\right\} \quad \text{for all } k, \tag{D.15}$$

*the iteration sequence will stay in $\mathcal{S}_{\zeta_0}^{(n+)}$ provided that our initialization $q^{(0)} \in \mathcal{S}_{\zeta_0}^{(n+)}$.*

**Proof.** By [Lemma D.1], if the current iterate $q \in \mathcal{S}_{\zeta_0}^{(n+)} \setminus \mathcal{S}_1^{(n+)}$, the next iterate $q_+ \in \mathcal{S}_{\zeta_0}^{(n+)}$, provided that $\eta \leq 1/(100\sqrt{n})$. Now if the current $q \in \mathcal{S}_1^{(n+)}$, i.e., $q_n^2/q_j^2 \geq 2$ for all $j \in [n-1]$, we can emulate the analysis of the set $\mathcal{I}_1$ in proof of [Lemma D.1]. Indeed, for any $j \in [n-1]$,

$$\frac{q_{+,n}^2}{q_{+,j}^2} \geq \frac{q_n^2(1-\eta g_n/q_n)^2}{q_j^2 + \eta^2 g_j^2} \geq \frac{q_n^2(1-2\eta\sqrt{n})^2}{q_n^2/2 + 4\eta^2} \geq \frac{2(1-2\eta\sqrt{n})^2}{1+8n\eta^2} \geq 1 + \zeta_0, \tag{D.16}$$

where the last inequality holds provided that $\eta \leq (1-\zeta_0)/(9\sqrt{n})$. Combining the two cases finishes the proof. ∎

## D.2 Proof of [Theorem 3.8]

As we have $\eta^{(k)} \leq \frac{1}{100\sqrt{n}}$ and $q^{(0)} \in \mathcal{S}_{\zeta_0}^{(n+)}$, the entire sequence $\{q^{(k)}\}_{k \geq 0}$ will stay in $\mathcal{S}_{\zeta_0}^{(n+)}$ by [Proposition D.2].

For any $q$ and any $v \in \partial_R f(q)$, we have $\langle v, q \rangle = 0$ and therefore

$$\|q - \eta v\|^2 = \|q\|^2 + \eta^2\|v\|^2 \geq 1. \tag{D.17}$$

So $q - \eta v$ is not inside $\mathbb{B}^n$. Since projection onto $\mathbb{B}^n$ is a contraction, we have

$$\|q_+ - e_n\|^2 = \left\|\frac{q - \eta v}{\|q - \eta v\|} - e_n\right\|^2 \leq \|q - \eta v - e_n\|^2$$

$$\leq \|q - e_n\|^2 + \eta^2\|v\|^2 - 2\eta\langle v, q - e_n\rangle \leq \|q - e_n\|^2 + 4\eta^2 - \frac{1}{8}\eta\theta(1-\theta)n^{-3/2}\zeta_0\|q - e_n\|, \tag{D.18}$$

where we have used the bounds in Proposition 3.7 and Theorem 3.6 to obtain the last inequality. Further applying Proposition C.7, we have

$$\left\| \boldsymbol{q}_+ - \boldsymbol{e}_n \right\|^2 \le \left\| \boldsymbol{q} - \boldsymbol{e}_n \right\|^2 + 4\eta^2 - \frac{1}{16}\eta\theta\left(1-\theta\right)n^{-2}\zeta_0\left(f\left(\boldsymbol{q}\right) - f\left(\boldsymbol{e}_n\right)\right). \tag{D.19}$$

Summing up the inequalities until step $K$ (assumed $\ge 5$), we have

$$0 \le \left\| \boldsymbol{q}^{(K)} - \boldsymbol{e}_n \right\|^2 + 4\sum_{j=0}^{K}\left(\eta^{(j)}\right)^2 - \frac{1}{16}\theta\left(1-\theta\right)n^{-2}\zeta_0\sum_{j=0}^{K}\eta^{(j)}\left(f\left(\boldsymbol{q}^{(j)}\right) - f\left(\boldsymbol{e}_n\right)\right) \tag{D.20}$$

$$\Longrightarrow \sum_{j=0}^{K}\eta^{(j)}\left(f\left(\boldsymbol{q}^{(j)}\right) - f\left(\boldsymbol{e}_n\right)\right) \le \frac{16\left\| \boldsymbol{q}^{(K)} - \boldsymbol{e}_n \right\|^2 + 64\sum_{j=0}^{K}\left(\eta^{(j)}\right)^2}{\theta\left(1-\theta\right)n^{-2}\zeta_0} \tag{D.21}$$

$$\Longrightarrow f\left(\boldsymbol{q}^{\text{best}}\right) - f\left(\boldsymbol{e}_n\right) \le \frac{16\left\| \boldsymbol{q}^{(K)} - \boldsymbol{e}_n \right\|^2 + 64\sum_{j=0}^{K}\left(\eta^{(j)}\right)^2}{\theta\left(1-\theta\right)n^{-2}\zeta_0\sum_{j=0}^{K}\eta^{(j)}}. \tag{D.22}$$

Substituting the following estimates

$$\sum_{j=0}^{K}\left(\eta^{(j)}\right)^2 \le \frac{1}{10^4 n}\left(1 + \int_0^K t^{-2\alpha}\,dt\right) \le \frac{1}{10^4 n}\frac{1}{1-2\alpha}\left(K^{1-2\alpha} + 1\right), \tag{D.23}$$

$$\sum_{j=0}^{K'}\eta^{(j)} \ge \frac{1}{10^2\sqrt{n}}\int_0^K t^{-\alpha}\,dt \ge \frac{1}{10^2\sqrt{n}}\frac{K^{1-\alpha}}{1-\alpha}, \tag{D.24}$$

and noting $16\left\| \boldsymbol{q}^{(K)} - \boldsymbol{e}_n \right\|^2 \le 32$, we have

$$f\left(\boldsymbol{q}^{\text{best}}\right) - f\left(\boldsymbol{e}_n\right) \le \frac{3200 n^{5/2}\left(1-\alpha\right) + 16/25 \cdot n^{3/2}\left(\frac{1-\alpha}{1-2\alpha}K^{1-2\alpha} + 1 - \alpha\right)}{\theta\left(1-\theta\right)\zeta_0 K^{1-\alpha}}. \tag{D.25}$$

Noting that

$$K \ge \left(\frac{6400 n^{5/2}\left(1-\alpha\right)}{\theta\left(1-\theta\right)\zeta_0\epsilon}\right)^{\frac{1}{1-\alpha}} \Longrightarrow \frac{3200 n^{5/2}\left(1-\alpha\right)}{\theta\left(1-\theta\right)\zeta_0 K^{1-\alpha}} \le \frac{\epsilon}{2}, \tag{D.26}$$

and when $K \ge 1$, $K^{1-2\alpha} \ge 1$, yielding that

$$K \ge \left(\frac{64 n^{3/2}\frac{1-\alpha}{1-2\alpha}}{25\epsilon\theta\left(1-\theta\right)\zeta_0}\right)^{\frac{1}{\alpha}} \Longrightarrow \frac{32 n^{3/2}\frac{1-\alpha}{1-2\alpha}K^{-\alpha}}{25\theta\left(1-\theta\right)\zeta_0} \le \frac{\epsilon}{2} \Longrightarrow \frac{16 n^{3/2}\cdot\left(1-\alpha\right)\left(\frac{1}{1-2\alpha}K^{1-2\alpha} + 1\right)}{25\theta\left(1-\theta\right)\zeta_0 K^{1-\alpha}} \le \frac{\epsilon}{2}. \tag{D.27}$$

So we conclude that when

$$K \ge \max\left\{\left(\frac{6400 n^{5/2}\left(1-\alpha\right)}{\theta\left(1-\theta\right)\zeta_0\epsilon}\right)^{\frac{1}{1-\alpha}}, \left(\frac{64 n^{3/2}\frac{1-\alpha}{1-2\alpha}}{25\epsilon\theta\left(1-\theta\right)\zeta_0}\right)^{\frac{1}{\alpha}}\right\}, \tag{D.28}$$

$f\left(\boldsymbol{q}^{\text{best}}\right) - f\left(\boldsymbol{e}_n\right) \le \epsilon$. When this happens, by Proposition C.8,

$$\left\| \boldsymbol{q}^{\text{best}} - \boldsymbol{e}_n \right\| \le \frac{16}{\theta\left(1-\theta\right)}\epsilon. \tag{D.29}$$

Plugging in the choice $\zeta_0 = 1/(5\log n)$ in Eq. (D.28) gives the desired bound on the number of iterations.

# E    PROOFS FOR SECTION 3.4

## E.1    PROOF OF LEMMA 3.9

**Lemma E.1.** *For all $n \ge 3$ and $\zeta \ge 0$, it holds that*

$$\frac{\text{vol}\left(\mathcal{S}_\zeta^{(n+)}\right)}{\text{vol}\left(\mathbb{S}^{n-1}\right)} \ge \frac{1}{2n} - \frac{9}{8}\frac{\log n}{n}\zeta. \tag{E.1}$$

We note that a similar result appears in (Gilboa et al., 2018) but our definitions of the region $\mathcal{S}_\zeta$ are slightly different. For completeness we provide a proof in Lemma F.3.

We now prove Lemma 3.9. Taking $\zeta = 1/(5 \log n)$ in Lemma E.1, we obtain

$$\frac{\text{vol}\left(\mathcal{S}_{1/(5 \log n)}^{(n+)}\right)}{\text{vol}\left(\mathbb{S}^{n-1}\right)} \geq \frac{1}{2n} - \frac{9}{8}\frac{\log n}{n} \cdot \frac{1}{5 \log n} \geq \frac{1}{4n}. \tag{E.2}$$

By symmetry, all the $2n$ sets $\left\{\mathcal{S}_{1/(5 \log n)}^{(i+)}, \mathcal{S}_{1/(5 \log n)}^{(i-)} : i \in [n]\right\}$ have the same volume which is at least $1/(4n)$. As $\boldsymbol{q}^{(0)} \sim \text{Uniform}(\mathbb{S}^{n-1})$, it falls into their union with probability at least $2n \cdot 1/(4n) = 1/2$, on which it belongs to a uniformly random one of these $2n$ sets.

### E.2 Proof of Theorem 3.10

Assume that the good event in Proposition 3.7 happens and that in Theorem 3.6 happens to all the $2n$ sets $\left\{\mathcal{S}_{1/(5 \log n)}^{(i+)}, \mathcal{S}_{1/(5 \log n)}^{(i-)} : i \in [n]\right\}$, which by setting $\zeta_0 = 1/(5 \log n)$ has probability at least

$$1 - \exp(-cm\theta^3\zeta_0^2 n^{-3} \log^{-1} m) - \exp(-cm\theta \log^{-1} m) = 1 - \exp(-c'm\theta^3 n^{-3} \log m^{-3}). \tag{E.3}$$

By Lemma 3.9, random initialization will fall these $2n$ sets with probability at least $1/2$. When it falls in one of these $2n$ sets, by Theorem 3.8, one run of the algorithm will find a signed standard basis vector up to $\epsilon$ accuracy. With $R$ independent runs, at least $S \doteq \frac{1}{4}R$ of them are effective with probability at least $1 - \exp\left(-(R/4)^2/(R/4 \cdot 2)\right) = 1 - \exp(-R/8)$, due to Bernstein's inequality. After these effective runs, the probability any standard basis vector is missed (up to sign) is bounded by

$$n\left(1 - \frac{1}{n}\right)^S \leq \exp\left(-\frac{S}{n} + \log n\right) \leq \exp\left(-\frac{S}{2n}\right), \tag{E.4}$$

where the second inequality holds whenever $S \geq 2n \log n$.

## F Auxiliary Calculations

**Lemma F.1.** *For $x \sim \text{BG}(\theta)$, $\|x\|_{\psi_2} \leq C_a$. For any vector $\boldsymbol{u} \in \mathbb{R}^n$ and $\boldsymbol{x} \sim_{iid} \text{BG}(\theta)$, $\left\|\boldsymbol{x}^\top \boldsymbol{u}\right\|_{\psi_2} \leq C_b \|\boldsymbol{u}\|$. Here $C_a, C_b \geq 0$ are universal constants.*

**Proof.** For any $\lambda \in \mathbb{R}$,

$$\exp(\lambda x) = \theta \exp(\lambda x) \leq \exp(\lambda x). \tag{F.1}$$

So $\|x\|_{\psi_2}$ is bounded by a universal constant. Moreover,

$$\left\|\boldsymbol{u}^\top \boldsymbol{x}\right\|_{\psi_2} = \left\|\sum_i u_i x_i\right\|_{\psi_2} \leq C_1\left(\sum_i u_i^2 \|x_i\|_{\psi_2}^2\right)^{1/2} \leq C_2 \|\boldsymbol{u}\|, \tag{F.2}$$

as claimed. ∎

**Lemma F.2.** *Let $\boldsymbol{a}_1, \ldots, \boldsymbol{a}_m$ be iid copies of $\boldsymbol{a} \sim_{iid} \text{BG}(\theta)$. Then,*

$$\mathbb{P}\left[\sup_{\boldsymbol{q} \in \mathbb{S}^{n-1}}\left|\sum_{i \in [m]}\left|\boldsymbol{q}^\top \boldsymbol{x}_i\right| - \mathbb{E}\left[\left|\boldsymbol{q}^\top \boldsymbol{x}\right|\right]\right| > C_a\sqrt{mn} + C_b\sqrt{m}t\right] \leq 2\exp\left(-t^2\right). \tag{F.3}$$

*for any $t \geq 0$. Here $C_a, C_b \geq 0$ are universal constants.*

**Proof.** Consider the zero-centered random process defined on $\mathbb{S}^{n-1}$: $X_{\boldsymbol{q}} \doteq \sum_{i \in [m]}\left(\left|\boldsymbol{q}^\top \boldsymbol{x}_i\right| - \mathbb{E}\left|\boldsymbol{q}^\top \boldsymbol{x}\right|\right)$. Then, for any $\boldsymbol{p}, \boldsymbol{q} \in \mathbb{S}^{n-1}$,

$$\|X_{\boldsymbol{p}} - X_{\boldsymbol{q}}\|_{\psi_2} = \left\|\sum_{i \in [m]}\left(\left|\boldsymbol{p}^\top \boldsymbol{x}_i\right| - \left|\boldsymbol{q}^\top \boldsymbol{x}_i\right| - \mathbb{E}\left|\boldsymbol{p}^\top \boldsymbol{x}\right| + \mathbb{E}\left|\boldsymbol{q}^\top \boldsymbol{x}\right|\right)\right\|_{\psi_2} \tag{F.4}$$

$$\leq C_1 \left( \sum_{i \in [m]} \left\| \left| \boldsymbol{p}^\top \boldsymbol{x}_i \right| - \left| \boldsymbol{q}^\top \boldsymbol{x}_i \right| - \mathbb{E} \left| \boldsymbol{p}^\top \boldsymbol{x} \right| + \mathbb{E} \left| \boldsymbol{q}^\top \boldsymbol{x} \right| \right\|_{\psi_2}^2 \right)^{1/2} \tag{F.5}$$

$$\leq C_2 \left( \sum_{i \in [m]} \left\| \left| \boldsymbol{p}^\top \boldsymbol{x}_i \right| - \left| \boldsymbol{q}^\top \boldsymbol{x}_i \right| \right\|_{\psi_2}^2 \right)^{1/2} \quad \text{(centering)} \tag{F.6}$$

$$\leq C_2 \left( \sum_{i \in [m]} \left\| (\boldsymbol{p} - \boldsymbol{q})^\top \boldsymbol{x}_i \right\|_{\psi_2}^2 \right)^{1/2} \tag{F.7}$$

$$= C_3 \sqrt{m} \left\| \boldsymbol{p} - \boldsymbol{q} \right\|, \tag{F.8}$$

where we use the estimate in Lemma F.1 to obtain the last inequality. Note that $X_{\boldsymbol{q}}$ is a mean-zero random process, and we can invoke Proposition A.2 with $w(\mathbb{S}^{n-1}) = C_4 \sqrt{n}$ and $\text{rad}\left(\mathbb{S}^{n-1}\right) = 2$ to get the claimed result. ∎

**Lemma F.3.** *For all $n \geq 3$ and $\zeta \geq 0$, it holds that*

$$\frac{\text{vol}\left(\mathcal{S}_\zeta^{(n+)}\right)}{\text{vol}\left(\mathbb{S}^{n-1}\right)} \geq \frac{1}{2n} - \frac{9}{8} \frac{\log n}{n} \zeta. \tag{F.9}$$

**Proof.** We have

$$\frac{\text{vol}\left(\mathcal{S}_\zeta^{(n+)}\right)}{\text{vol}\left(\mathbb{S}^{n-1}\right)} = \mathbb{P}_{\boldsymbol{q} \sim \text{uniform}(\mathbb{S}^{n-1})} \left[ q_n^2 \geq (1 + \zeta) \left\| \boldsymbol{q}_{-n} \right\|_\infty^2, q_n \geq 0 \right] \tag{F.10}$$

$$= \mathbb{P}_{\boldsymbol{x} \sim \mathcal{N}(\boldsymbol{0}, \boldsymbol{I}_n)} \left[ x_n \geq 0, x_n^2 \geq (1 + \zeta) x_i^2 \ \forall \ i \neq n \right] \tag{F.11}$$

$$= (2\pi)^{n/2} \int_0^\infty e^{-x_n^2/2} \left( \prod_{j=1}^{n-1} \int_{-x_n/\sqrt{1+\zeta}}^{x_n/\sqrt{1+\zeta}} e^{-x_j^2/2} \ dx_j \right) \ dx_n \tag{F.12}$$

$$= (2\pi)^{1/2} \int_0^\infty e^{-x_n^2/2} \psi^{n-1} \left( x_n / \sqrt{1+\zeta} \right) \ dx_n \tag{F.13}$$

$$= \frac{\sqrt{1+\zeta}}{\sqrt{2\pi}} \int_0^\infty e^{-(1+\zeta)x^2/2} \psi^{n-1}(x) \ dx \doteq \hbar(\zeta) > 0, \tag{F.14}$$

where we write $\psi(t) \doteq \frac{1}{\sqrt{2\pi}} \int_{-t}^t \exp\left(-s^2/2\right) \ ds$. Now we derive a lower bound of the volume ratio by considering a first-order Taylor expansion of the last equation around $\zeta = 0$ (as we are mostly interested in small $\zeta$). By symmetry, $\hbar(0) = 1/(2n)$. Moreover, we have

$$\left. \frac{\partial \hbar(\zeta)}{\partial \zeta} \right|_{\zeta=0} = \frac{1}{2} \frac{1}{\sqrt{2\pi}} \int_0^\infty e^{-x^2/2} \psi^{n-1}(x) \ dx - \frac{1}{\sqrt{2\pi}} \int_0^\infty e^{-x^2/2} x^2 \psi^{n-1}(x) \ dx \tag{F.15}$$

$$= \frac{1}{4n} - \frac{1}{2\sqrt{2\pi}} \int_0^\infty e^{-x^2/2} x^2 \psi^{n-1}(x) \ dx. \tag{F.16}$$

Now we provide an upper bound for the second term of the last equation. Note that

$$\frac{1}{\sqrt{2\pi}} \int_0^\infty e^{-x^2/2} x^2 \psi^{n-1}(x) \ dx = \mathbb{E}_{\boldsymbol{x} \sim \mathcal{N}(\boldsymbol{0}, \boldsymbol{I}_n)} \left[ x_n^2 \mathbb{1}\left\{ x_n^2 \geq \left\| \boldsymbol{x}_{-n} \right\|_\infty^2 \right\} \mathbb{1}\left\{ x_n \geq 0 \right\} \right] \tag{F.17}$$

$$= \frac{1}{2n} \mathbb{E}_{\boldsymbol{x} \sim \mathcal{N}(\boldsymbol{0}, \boldsymbol{I}_n)} \left\| \boldsymbol{x} \right\|_\infty^2. \tag{F.18}$$

Now for any $\lambda \in (0, 1/2)$,

$$\exp\left( \lambda \mathbb{E} \left\| \boldsymbol{x} \right\|_\infty^2 \right) \leq \mathbb{E} \exp\left( \lambda \left\| \boldsymbol{x} \right\|_\infty^2 \right) \leq \sum_{j=1}^n \mathbb{E} \exp\left( \lambda x_j^2 \right) = n \mathbb{E}_{x \sim \mathcal{N}(0,1)} \exp\left( \lambda x^2 \right) \leq \frac{n}{\sqrt{1-2\lambda}}. \tag{F.19}$$

Taking logarithm on both sides, rearranging the terms, and setting $\lambda = 1/4$, we obtain

$$\mathbb{E} \|\boldsymbol{x}\|_\infty^2 \leq \inf_{\lambda \in (0,1/2)} \frac{\log n + \frac{1}{2} \log (1 - 2\lambda)^{-1}}{\lambda} \leq 4 \log n + 2 \log 2. \tag{F.20}$$

So

$$\left. \frac{\partial \hbar (\zeta)}{\partial \zeta} \right|_{\zeta=0} \geq \frac{1}{4n} - \frac{1}{4n} (4 \log n + 2 \log 2) \geq -\frac{9}{8} \frac{\log n}{n}, \tag{F.21}$$

provided that $n \geq 3$. Now we show that $\hbar (\zeta) \geq \hbar (0) + \hbar'(0)\zeta$ by showing that $\hbar''(\zeta) \geq 0$. We have

$$\frac{\partial^2 \hbar (\zeta)}{\partial \zeta^2} = \frac{\sqrt{1+\zeta}}{4\sqrt{2\pi}} \int_0^\infty \left[ x^4 - \frac{2x^2}{1+\zeta} - \frac{1}{(1+\zeta)^2} \right] e^{-\frac{1+\zeta}{2} x^2} \psi^{n-1} (x) \ dx. \tag{F.22}$$

Using integration by part, we have

$$\int_0^\infty \left[ x^4 - \frac{3x^2}{1+\zeta} \right] e^{-\frac{1+\zeta}{2} x^2} \psi^{n-1} (x) \ dx \tag{F.23}$$

$$= -\frac{1}{1+\zeta} e^{-\frac{1+\zeta}{2} x^2} x^3 \cdot \psi^{n-1} (x) \Big|_0^\infty + \int_0^\infty \frac{1}{1+\zeta} e^{-\frac{1+\zeta}{2} x^2} x^3 (n-1) \psi^{n-2} (x) \sqrt{\frac{2}{\pi}} e^{-\frac{x^2}{2}} \ dx \tag{F.24}$$

$$= \int_0^\infty \frac{1}{1+\zeta} e^{-\frac{1+\zeta}{2} x^2} x^3 (n-1) \psi^{n-2} (x) \sqrt{\frac{2}{\pi}} e^{-\frac{x^2}{2}} \ dx \geq 0, \tag{F.25}$$

and similarly

$$\int_0^\infty \left[ x^2 - \frac{1}{1+\zeta} \right] e^{-\frac{1+\zeta}{2} x^2} \psi^{n-1} (x) \ dx \tag{F.26}$$

$$= -\frac{1}{1+\zeta} e^{-\frac{1+\zeta}{2} x^2} x \cdot \psi^{n-1} (x) \Big|_0^\infty + \int_0^\infty \frac{1}{1+\zeta} e^{-\frac{1+\zeta}{2} x^2} x (n-1) \psi^{n-2} (x) \sqrt{\frac{2}{\pi}} e^{-\frac{x^2}{2}} \ dx \tag{F.27}$$

$$= \int_0^\infty \frac{1}{1+\zeta} e^{-\frac{1+\zeta}{2} x^2} x (n-1) \psi^{n-2} (x) \sqrt{\frac{2}{\pi}} e^{-\frac{x^2}{2}} \ dx \geq 0. \tag{F.28}$$

Noting that

$$x^4 - \frac{2x^2}{1+\zeta} - \frac{1}{(1+\zeta)^2} = x^4 - \frac{3x^2}{1+\zeta} + \frac{1}{1+\zeta} \left( x^2 - \frac{1}{1+\zeta} \right) \tag{F.29}$$

and combining the above integral results, we conclude that $\hbar''(\zeta) \geq 0$ and complete the proof. $\blacksquare$

**Lemma F.4.** *Let* $(x_1, y_1), (x_2, y_2) \in \mathbb{R}_{>0}^2$ *be two points in the first quadrant satisfying* $y_1 \geq x_1$ *and* $y_2 \geq x_2$, *and* $\frac{y_2/x_2}{y_1/x_1} \in [1, 1+\eta]$ *for some* $\eta \leq 1$, *then we have* $\angle ((x_1, y_1), (x_2, y_2)) \leq \eta$.

**Proof.** For $i = 1, 2$, let $\theta_i$ be the angle between the ray $(x_i, y_i)$ and the $x$-axis. Our assumption implies that $\theta_i \in [\pi/4, \pi/2)$ and $\theta_2 \geq \theta_1$, thus $\angle ((x_1, y_1), (x_2, y_2)) = \theta_2 - \theta_1$, so we have

$$\begin{aligned}
&\tan \angle ((x_1, y_1), (x_2, y_2)) \\
&= \frac{\tan \theta_2 - \tan \theta_1}{1 + \tan \theta_2 \tan \theta_1} \\
&= \frac{y_2/x_2 - y_1/x_1}{1 + y_2 y_1/(x_2 x_1)} \\
&= \frac{\frac{y_2/x_2}{y_1/x_1} - 1}{y_2/x_2 + x_1/y_1} \\
&\leq \frac{y_2/x_2}{y_1/x_1} - 1 \\
&\leq \eta.
\end{aligned} \tag{F.30}$$

Therefore $\angle ((x_1, y_1), (x_2, y_2)) \leq \arctan(\eta) \leq \eta$. $\blacksquare$

**Lemma F.5.** *For any $\boldsymbol{p}, \boldsymbol{q} \in \mathbb{S}^{n-1}$ with the same support pattern such that $\mathrm{d}_{\mathbb{E}}(\boldsymbol{p}, \boldsymbol{q}) \leq \epsilon \leq \frac{1}{2}$, we have for all $\boldsymbol{u} \in \mathbb{S}^{n-1}$ that*

$$\mathbb{E}_{\boldsymbol{x} \sim \mathrm{BG}(\theta)} \left[ |\boldsymbol{u}^\top \boldsymbol{x}| \mathbb{1} \left\{ \mathrm{sign}(\boldsymbol{p}^\top x) \neq \mathrm{sign}(\boldsymbol{q}^\top \boldsymbol{x}) \right\} \right] \leq 3\epsilon \sqrt{\log \frac{1}{\epsilon}}. \tag{F.31}$$

**Proof.** Fix some threshold $t > 0$ to be determined. We have

$$\mathbb{E} \left[ |\boldsymbol{u}^\top \boldsymbol{x}| \mathbb{1} \left\{ \mathrm{sign}(\boldsymbol{p}^\top x) \neq \mathrm{sign}(\boldsymbol{q}^\top \boldsymbol{x}) \right\} \right] \tag{F.32}$$

$$\leq \mathbb{E} \left[ |\boldsymbol{u}^\top \boldsymbol{x}| \mathbb{1} \left\{ |\boldsymbol{u}^\top \boldsymbol{x}| > t \right\} \right] + \mathbb{E} \left[ |\boldsymbol{u}^\top \boldsymbol{x}| \mathbb{1} \left\{ |\boldsymbol{u}^\top \boldsymbol{x}| \leq t, \mathrm{sign}(\boldsymbol{p}^\top \boldsymbol{x}) \neq \mathrm{sign}(\boldsymbol{q}^\top \boldsymbol{x}) \right\} \right] \tag{F.33}$$

$$\leq \left( \mathbb{E} \left[ (\boldsymbol{u}^\top \boldsymbol{x})^2 \right] \cdot \mathbb{P} \left[ |\boldsymbol{u}^\top \boldsymbol{x}| > t \right] \right)^{1/2} + t\mathbb{E} \left[ \mathbb{1} \left\{ \mathrm{sign}(\boldsymbol{p}^\top \boldsymbol{x}) \neq \mathrm{sign}(\boldsymbol{q}^\top \boldsymbol{x}) \right\} \right] \tag{F.34}$$

$$\leq \left( \theta \cdot 2 \exp(-t^2/2) \right)^{1/2} + \epsilon t. \tag{F.35}$$

The second to last inequality uses Cauchy-Schwarz, and the last inequality uses the fact that $\boldsymbol{u}^\top \boldsymbol{x} = \boldsymbol{u}_\Omega^\top \boldsymbol{x}_\Omega$ is $\|\boldsymbol{u}_\Omega\|_2^2$-sub-Gaussian conditioned on $\Omega$ and thus 1-sub-Gaussian marginally. Taking $t = \sqrt{2 \log \frac{1}{\epsilon^2}}$, the above bound simplifies to

$$\sqrt{2\theta \exp \left( -\log \frac{1}{\epsilon^2} \right)} + \epsilon \sqrt{2 \log \frac{1}{\epsilon^2}} = \sqrt{2\theta} \epsilon \left( 1 + \log \frac{1}{\epsilon^2} \right) = \epsilon \left( \sqrt{2\theta} + \sqrt{2 \log \frac{1}{\epsilon^2}} \right) \leq 3\epsilon \sqrt{\log \frac{1}{\epsilon}} \tag{F.36}$$

where we have used $\theta \leq 1/2$ and $\epsilon \leq 1/2$. ∎

# G  RESULTS ON ORTHOGONAL DICTIONARIES

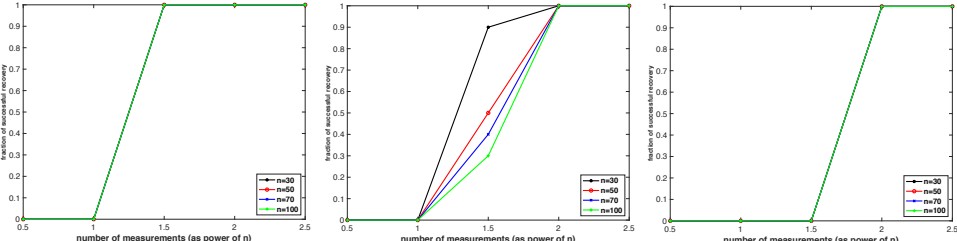

Figure 2: Empirical success rates of recovery of the Riemannian subgradient descent with $R = 5n \log n$ runs, averaged over 10 instances. Left to right: orthogonal dictionaries with $\theta = 0.1, 0.3, 0.5$.

# H  FASTER ALTERNATIVE ALGORITHM FOR LARGE-SCALE INSTANCES

The Riemannian subgradient descent is cheap per iteration but slow in overall convergence, similar to many other first-order methods. We also test a faster quasi-Newton type method, GRANSO,[6] that employs BFGS for solving constrained nonsmooth problems based on sequential quadratic optimization (Curtis et al., 2017). For a large dictionary of dimension $n = 400$ and sample complexity $m = 10n^2$ (i.e., $1.6 \times 10^6$), GRANSO successfully identifies a basis after 1500 iterations with CPU time 4 hours on a two-socket Intel Xeon E5-2640v4 processor (10-core Broadwell, 2.40 GHz)—this is approximately $10\times$ faster than the Riemannian subgradient descent method, showing the potential of quasi-Newton type methods for solving large-scale problems.

# I  EXPERIMENT WITH IMAGES

To experiment with images, we follow a typical setup for dictionary learning as used in image processing (Mairal et al., 2014). We focus on testing if complete (i.e., square and invertible) dictionaries are reasonable sparsification bases for real images, instead on any particular image processing or vision tasks.

---

[6]Available online: http://www.timmitchell.com/software/GRANSO/.

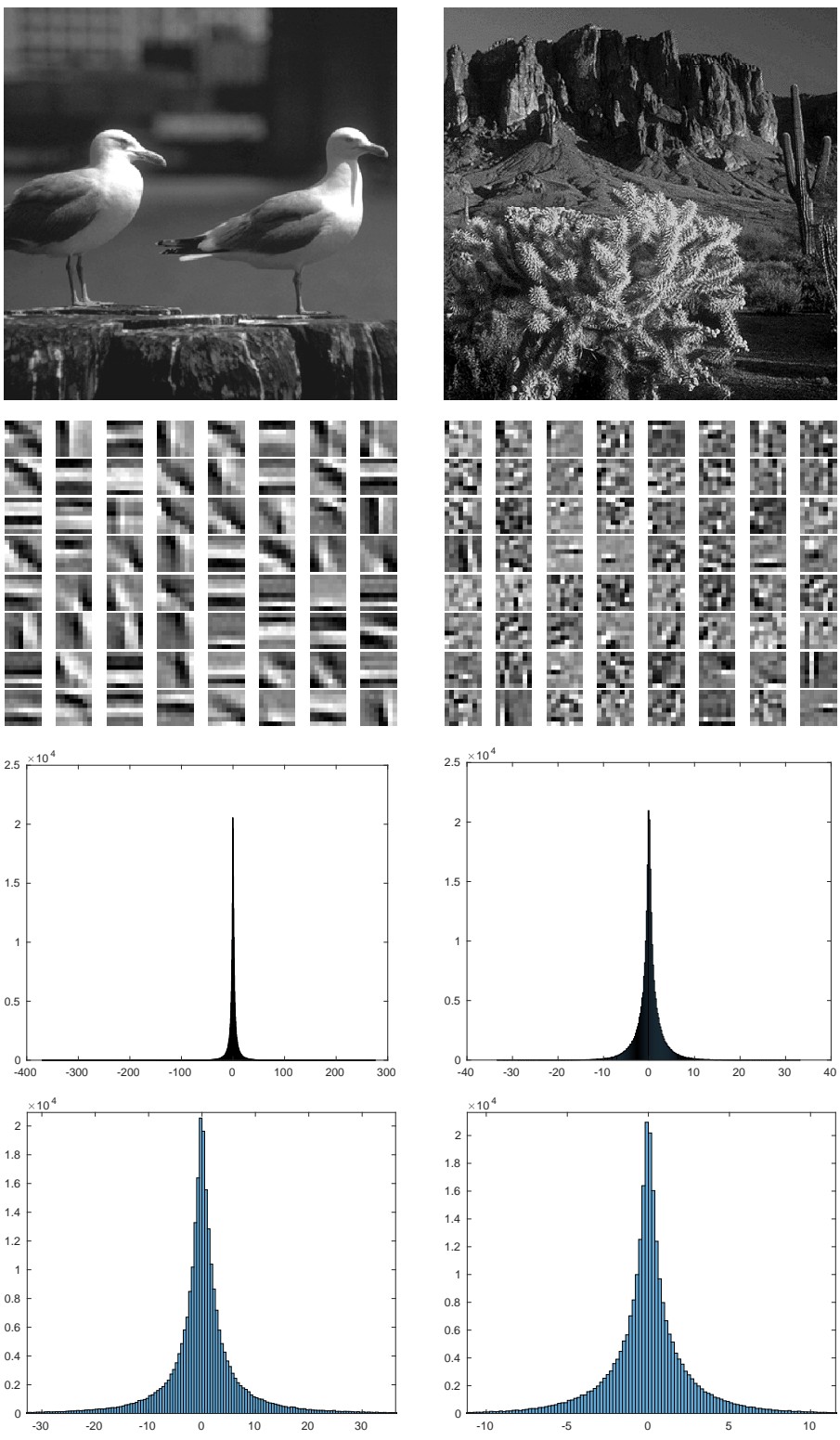

Figure 3: Results on two images. First row: the images; Second row: learned dictionaries; Third row: histograms of the representation coefficients; Fourth row: zoomed-in versions of the histograms around zero.

**Setup** Two natural images are picked for this experiment, as shown in the first row of Fig. 3, each of resolution $512 \times 512$. Each image is divided into $8 \times 8$ non-overlapping blocks, resulting in $64 \times 64 = 4096$ blocks. The blocks are then vectorized, and stacked columnwise into a data matrix $\boldsymbol{Y} \in \mathbb{R}^{64 \times 4096}$. We precondition the data to obtain

$$\overline{\boldsymbol{Y}} = \left(\boldsymbol{Y}\boldsymbol{Y}^\top\right)^{-1/2} \boldsymbol{Y}, \tag{I.1}$$

so that nonvanishing singular values of $\overline{\boldsymbol{Y}}$ are identically one. We then solve formulation (1.1) round $(5n \log n)$ times with $n = 64$ using the BFGS solver based on GRANSO, obtaining round $(5n \log n)$ vectors. Negative equivalent copies are pruned and vectors with large correlations with other remaining vectors are sequentially removed until only $64$ vectors are left. This forms the final complete dictionary.

**Results** The learned complete dictionaries for the two test images are displayed in the second row of Fig. 3. Visually, the dictionaries seem reasonably adaptive to the image contents: for the left image with prevalent sharp edges, the learned dictionary consists of almost exclusively oriented sharp corners and edges, while for the right image with blurred textures and occasional sharp features, the learned dictionary does seem to be composed of the two kinds of elements. Let the learned dictionary be $\boldsymbol{A}$. We estimate the representation coefficients as $\boldsymbol{A}^{-1}\overline{\boldsymbol{Y}}$. The third row of Fig. 3 contains the histograms of the coefficients. For both images, the coefficients are sharply concentrated around zero (see also the fourth row for zoomed versions of the portions around zero), and the distribution resembles a typical zero-centered Laplace distribution—which is a good indication of sparsity. Quantitatively, we calculate the mean sparsity level of the coefficient vectors (i.e., columns of $\boldsymbol{A}^{-1}\boldsymbol{Y}$) by the metric $\left\|\cdot\right\|_1 / \left\|\cdot\right\|_2$: for a vector $\boldsymbol{v} \in \mathbb{R}^n$, $\left\|\boldsymbol{v}\right\|_1 / \left\|\boldsymbol{v}\right\|_2$ ranges from $1$ (when $\boldsymbol{v}$ is one-sparse) to $\sqrt{n}$ (when $\boldsymbol{v}$ is fully dense with elements of equal magnitudes), which serves as a good measure of sparsity level for $\boldsymbol{v}$. For our two images, the sparsity levels by the norm-ratio metric are $5.9135$ and $6.4339$, respectively, while the fully dense extreme would have a value $\sqrt{64} = 8$, suggesting the complete dictionaries we learned are reasonable sparsification bases for the two natural images, respectively.

