# OpenReview forum: "Subgradient Descent Learns Orthogonal Dictionaries"
_ICLR.cc/2019/Conference_

### Official Review · AnonReviewer1 · 2018-11-02
**Relevant problem,  incomplete paper**

**Rating:** 6
**Confidence:** 1

**Review:**

The paper proposes a subgradient descent method to learn orthogonal, squared /complete n x n  dictionaries under l1 norm regularization. The problem is interesting and relevant, and the paper, or at least the first part, is clear.

The most interesting property is that the solution does not depend on the dictionary initialization, unlike many other competing methods.

The experiments sections in disappointingly short. Could the authors play with real data? How does sparsity affect the results? How does it change with different sample complexities? Also, it would be nice to have a final conclusion section. I think the paper contains interesting material but, overall, it gives the impression that the authors rushed to submit the paper before the deadline!

---

> ### Author Response · Authors · 2018-11-11
> **Will expand experiments and add conclusion/discussion**
>
> Thank you for the thoughtful feedback!
>
> Our preliminary experiments do show the effect of sample complexity -- in particular, empirically the subgradient descent algorithm almost always succeed as long as m = O(n^2), which is even better than the O(n^4) suggested by our theory.
>
> We are working on additional experiments comparing different sparsity, and real data experiments. (The experiments are indeed a bit time-consuming and would require days.)
>
> We are also working on adding a conclusion section and revising the paper a bit. Please stay tuned and we will let you know when it’s done.

---

> > ### Author Response · Authors · 2018-11-14
> > **Update: paper revised**
> >
> > We have expanded our synthetic experiment section, added an experiment with real data, and added a conclusion section which discusses some connections to shallow neural nets. Please feel free to take a look at our revision.

---

### Official Review · AnonReviewer2 · 2018-11-05
**Non-smooth non-convex optimization approach to complete dictionary learning**

**Rating:** 7
**Confidence:** 3

**Review:**

This paper is a direct follow-up on the Sun-Qu-Wright non-convex optimization view on the Spielman-Wang-Wright complete dictionary learning approach. In the latter paper the idea is to simply realize that with Y=AX, X being nxm sparse and A a nxn rotation, one has the property that for m large enough, the rows of X will be the sparsest element of the subspace in R^m generated by the rows of Y. This leads to a natural non-convex optimization problem, whose local optimum are hopefully the rows of X. This was proved in SWW for *very* sparse X, and then later improved in SQW to the linear sparsity scenario. The present paper refines this approach, and obtain slightly better sample complexity by studying the most natural non-convex problem (ell_1 regularization on the sphere).


I am not an expert on SQW so it is hard to evaluate how difficult it was to extend their approach to the non-smooth case (which seems to be the main issue with ell_1 regularization compared to the surrogate loss of SQW).


Overall I think this is a solid theoretical contribution, at least from the point of view of non-smooth non-convex optimization. I have some concerns about the model itself. Indeed *complete* dictionary learning seemed like an important first step in 2012 towards more general and realistic scenario. It is unclear to this reviewer whether the insights gained for this complete scenario are actually useful more generally.

---

> ### Author Response · Authors · 2018-11-11
> **Response on SQW and potential generalizations**
>
> Thank you for the positive feedback! We respond to the specific questions in turn.
>
> “Challenge of extending SQW to non-smooth case” --- The high-level ideas of obtaining the two results are the same: characterizing the nice global landscape of the respective objectives on the sphere, and then designing specific optimization algorithms taking advantage of the particular landscapes. Characterization of the landscape is through the use of first-order (and second-order) derivatives. For our nonsmooth setting, we have to use the subdifferential to describe the first-order geometry, which involves dealing with set-valued functions and random sets (due to the randomness in the data assumption)---very different than dealing with the gradient and Hessian in the smooth calculus, as in SQW. Moreover, traditional argument of uniform convergence of random quantities to their expectation often relies on Lipschitz property of the quantities of interest. For random sets, the notion of concentration is unconventional, and the desired Lipschitz property also fails to hold. We introduce tools from random set theory and construct a novel concentration argument getting around the Lipschitz requirement. This in turn implies that the first-order geometry of the sample objective is close to the benign population objective, from which the algorithmic guarantee follows.
>
> “Potential generalizations” ---  We believe that our theory has the potential to generalize into the overcomplete case. There, a natural generalization of the orthogonality assumption is that the dictionary A is a well-conditioned tight frame (n x L “fat” matrix with orthonormal rows and suitably widespread columns in the n-dim space). Although the "sparse vectors in a linear subspace" intuition fails there, we would still expect the columns a_i of A minimize the population objective ||a^T Y||_1 = ||a^T A X||_1: due to the widespread nature of columns of A, a_i^T A would be an “approximately 1-sparse” vector (i.e., with one dominant entry and others having small magnitudes) and so vectors a_i^T AX are expected to be noisy versions of rows of X, which are the sparest (in a soft sense) vectors among all vectors of the form a^T AX. Figuring out the precise optimization landscape in that case would be of great interest.

---

### Official Review · AnonReviewer3 · 2018-11-09
**solid analysis and new insights**

**Rating:** 7
**Confidence:** 4

**Review:**

This paper studies nonsmooth and nonconvex optimization and provides a global analysis for orthogonal dictionary learning. The analysis is highly nontrivial compared with existing work. Also for dictionary learning nonconvex $\ell_1$ minimization is very important due to its robustness properties.

I am wondering how extendable is this approach to overcomplete dictionary learning. It seems that overcomplete dictionary would break the key observation of "sparsest vector in the subspace".

Is it possible to circumvent the difficulty of nonsmoothness using (randomized) smoothing, and then apply the existing theory to the transformed objective? My knowledge is limited but this seems to be a more natural thing to try first. Could the authors compare this naive approach with the one proposed in the paper?

Another minor question is about the connection with training deep neural networks. It seems that in practical training algorithms we often ignore the fact that ReLU is nonsmooth since it only has one nonsmooth point — only with diminishing probability, it affects the dynamics of SGD, which makes subgradient descent seemingly unnecessary. Could the authors elaborate more on this connection?

---

> ### Author Response · Authors · 2018-11-11
> **Extension to overcomplete case; role of nonsmoothness**
>
> Thank you for the positive feedback! We respond to the questions in the following.
>
> “Extending to overcomplete DL” --- We believe that our theory has the potential to generalize into the overcomplete case. There, a natural generalization of the orthogonality assumption is that the dictionary A is a well-conditioned tight frame (n x L “fat” matrix with orthonormal rows and suitably widespread columns in the n-dim space). Although the "sparse vectors in a linear subspace" intuition fails there, we would still expect the columns a_i of A minimize the population objective ||a^T Y||_1 = ||a^T A X||_1: due to the widespread nature of columns of A, a_i^T A would be an “approximately 1-sparse” vector (i.e., with one dominant entry and others having small magnitudes) and so vectors a_i^T AX are expected to be noisy estimates of rows of X, which are the sparest (in a soft sense) vectors among all vectors of the form a^T AX. Figuring out the precise optimization landscape in that case would be of great interest.
>
> “Nonsmooth approach vs. (randomized) smoothing” --- We wonder whether you’re referring to the smoothed *objective*, or applying smoothing *algorithms* on our non-smooth objective. We will discuss both as follows.
>
> A smoothed objective was analyzed in Sun et al.‘15. Smoothing therein helped to make conventional calculus tools and expectation-concentration style argument readily applicable conceptually, but the smoothed objective and its low-order derivatives led to involved technical analysis---the smoothed objective loses the simplicity of the L1 function. This tends to be the case for several natural smoothing schemes. Also, L1 function is the regularizer people use in practical dictionary learning. This paper directly works with the non-smooth L1 objective and is able to obtain stronger results with a substantially cleaner argument, using unconventional yet highly accessible tools from nonsmooth analysis, set-valued analysis, and random set theory.
>
> Smoothing algorithms on non-smooth objective is an active area of ongoing research. For example, Jin et al. ‘18 showed that randomized smoothing algorithms succeed on minimizing non-smooth objectives as long as it is point-wise close to a smooth objective, which is often chosen to be its expected version. However, in our case, even the expected objective is non-smooth (see e.g. Section 3.1), so it is not readily applicable. Moreover, the result there is based on a zero-th order method, which is a conservative algorithmic choice when the (sub)gradient information is readily available---this is the case for us. In this paper, we are able to show the convergence of subgradient descent (i.e., a first-order method) directly on the non-smooth objective. It would be of interest to see whether first-order smoothing algorithms work as well.
>
> “Nonsmoothness in neural networks” --- It depends on what perspective we take.
>
> If we are interested in the landscape (i.e. the global geometry of the loss function), then the nonsmoothness matters a lot as the nonsmooth points are scattered everywhere in the space, and if one initializes the model adversarially near the highly nonsmooth parts, intuitively the performance can be hurt by the nonsmoothness.
>
> However, if we are more interested in the trajectory of some particular algorithms (say, SGD), then maybe the non-smoothness won’t hurt a lot --- as long as nice properties on the trajectory can be established. Such a trajectory-specific analysis has been done recently in, e.g., Du et al. ‘18. Even in this kind of results, there is no formal theory or statement saying that the nonsmooth points won’t be encountered.
>
> Besides our work, there are other recent papers showing why nonsmoothness should and can be handled on a rigorous basis, e.g., Laurent & von Brecht ’17, Kakade & Lee ’18.
>
> Reference:
> Sun, J., Qu, Q., & Wright, J. (2015). Complete Dictionary Recovery over the Sphere I: Overview and the Geometric Picture. arXiv preprint arXiv:1511.03607.
>
> Jin, C., Liu, L. T., Ge, R., & Jordan, M. I. (2018). Minimizing Nonconvex Population Risk from Rough Empirical Risk. arXiv preprint arXiv:1803.09357.
>
> Du, S. S., Zhai, X., Poczos, B., & Singh, A. (2018). Gradient Descent Provably Optimizes Over-parameterized Neural Networks. arXiv preprint arXiv:1810.02054.
>
> Laurent, T., & von Brecht, J. (2017). The Multilinear Structure of ReLU Networks. arXiv preprint arXiv:1712.10132.
>
> Kakade, S., & Lee, J. D. (2018). Provably Correct Automatic Subdifferentiation for Qualified Programs. arXiv preprint arXiv:1809.08530.

---

### Official Review · AnonReviewer4 · 2018-11-10
**A good paper**

**Rating:** 7
**Confidence:** 3

**Review:**

This paper studies dictionary learning problem by a non-convex constrained l1 minimization. By using subgradient descent algorithm with random initialization, they provide a non-trivial global convergence analysis for problem. The result is interesting, which does not depend on the complicated initializations used in other methods.

The paper could be better, if the authors could provide more details and results on numerical experiments.   This could be used to confirm the proved theoretical properties in practical algorithms.

---

> ### Author Response · Authors · 2018-11-11
> **Thanks & will have more detailed experiments**
>
> Thank you for the positive feedback!
>
> We are performing some more experiments as well as expanding the experiments section in more details. Please stay tuned and we will let you know when it’s done.

---

> > ### Author Response · Authors · 2018-11-14
> > **Update: paper revised**
> >
> > We have expanded the synthetic experiments in Section 5 and added a real data experiments in Appendix H. Please feel free to take a look at our revision.

---

### Official Review · AnonReviewer5 · 2018-11-12
**Nice work on nonconvex nonsmooth theory, needs more work on experiments and relation to loss landscape of neural networks mentioned in abstract**

**Rating:** 7
**Confidence:** 2

**Review:**

The paper provides a very nice analysis for the nonsmooth (l1) dictionary learning minimization in the case of orthogonal complete dictionaries and linearly sparse signals. They utilize a subgradient method and prove a non-trivial convergence result.

The theory provided is solid and expands on the earlier works of sun et al. for the nonsmooth case. Also interesting is the use a covering number argument with the d_E metric.

A big plus of the method presented is that unlike previous methods the subgradient descent based scheme presented is independent of the initialization.

Despite a solid theory developed, lack of numerical experiments reduces the quality of the paper. Additional experiments with random data to illustrate the theory would be beneficial and it would also be nice to find applications with real data.

In addition as mentioned in the abstract the authors suggest that the methods used in the paper may also aid in the analysis of shallow non-smooth neural networks but they need to continue and elaborate with more explicit connections.

Minor typos near the end of the paper and perhaps missing few definitions and notation are also a small concern

The paper is a very nice work and still seems significant! Nonetheless, fixing the above will elevate the quality of the paper.

---

> ### Author Response · Authors · 2018-11-14
> **Response**
>
> Thank you for your valuable feedback!
>
> We have expanded our synthetic experiment section, added an experiment with real data, and added a conclusion section which discusses some connections to shallow neural nets. Please feel free to take a look at our revision.

---

### Author Response · Authors · 2018-11-14
**Revision: expanded synthetic experiments + real data experiments + conclusion**

We have made a revision of our paper. The major changes are summarized as follows:

(1) The synthetic experiment (Section 5) is slightly expanded with results on different sparsity (\theta = 0.1, 0.3, 0.5). Recovery is easier when the sparsity is higher (i.e. \theta is lower), but in all cases we get successful recovery when m >= O(n^2).

(2) We added an experiment on real images (Appendix H), which shows that complete dictionaries offer a reasonable sparsifying basis for real image patches.

(3) We have added a conclusion section (Section 6) with discussions of our contributions and future directions.

---

### Meta-Review · Area_Chair1 · 2018-12-16
**Good ratings, strong theory**

**Confidence:** 4
**Recommendation:** Accept (Poster)

**Metareview:**

This paper studies non smooth and non convex optimization and provides a global analysis for orthogonal dictionary learning. The referees indicate that the analysis is highly nontrivial compared with existing work.

The experiments fall a bit short and the relation to the loss landscape of neural networks could be described more clearly.

The reviewers pointed out that the experiments section was too short. The revision included a few more experiments. The paper has a theoretical focus, and scores high ratings there.

The confidence levels of the reviewers is relatively moderate, with only one confident reviewer. However, all five reviewers regard this paper positively, in particular the confident reviewer.